# CAPSUL: A Comprehensive Human Protein Benchmark for Subcellular Localization

**Yicheng Hu**[1], **Xinyu Lin**[2], **Shulin Li**[3][*] **Wenjie Wang**[1], **Fengbin Zhu**[2][†] **Fuli Feng**[1]
[1] University of Science and Technology of China, [2] National University of Singapore,
[3] Tsinghua University

## ABSTRACT

Subcellular localization is a crucial biological task for drug target identification and function annotation. Although it has been biologically realized that subcellular localization is closely associated with protein structure, no existing dataset offers comprehensive 3D structural information with detailed subcellular localization annotations, thus severely hindering the application of promising structure-based models on this task. To address this gap, we introduce a new benchmark called **CAPSUL**, a **C**omprehensive hum**A**n **P**rotein benchmark for **SU**bcellular **L**ocalization. It features a dataset that integrates diverse 3D structural representations with fine-grained subcellular localization annotations carefully curated by domain experts. We evaluate this benchmark using a variety of state-of-the-art sequence-based and structure-based models, showcasing the importance of involving structural features in this task. Furthermore, we explore reweighting and single-label classification strategies to facilitate future investigation on structure-based methods for this task. Lastly, we showcase the powerful interpretability of structure-based methods through a case study on the Golgi apparatus, where we discover a decisive localization pattern $\alpha$-helix from attention mechanisms, demonstrating the potential for bridging the gap with intuitive biological interpretability and paving the way for data-driven discoveries in cell biology.

## 1 INTRODUCTION

Understanding the subcellular localization of proteins is a fundamental question in cell biology, as a protein's function is often tightly coupled to its spatial context within the cell (Scott et al., 2005). Localization information is essential for elucidating molecular mechanisms such as signal transduction, metabolic regulation, and organelle-specific functions (Hung et al., 2017). It also provides a foundation for translational applications such as drug design (Hung et al., 2017; Rajendran et al., 2010). Recently, the data-driven AI approaches have emerged as a powerful paradigm for predicting whether or not a protein will be localized to a specific subcellular location. These methods substantially reduce the time and cost associated with traditional experimental techniques while holding promise for revealing novel biological patterns, thereby showcasing promising performance and attracting extensive research attention (Thumuluri et al., 2022; Stärk et al., 2021; Almagro Armenteros et al., 2017; Kobayashi et al., 2022; Elnaggar et al., 2021).

However, there remains a significant scarcity of high-quality datasets designed for this task. To the best of our knowledge, the only widely accepted dataset targeting this problem in the AI field is DeepLoc (Thumuluri et al., 2022; Almagro Armenteros et al., 2017), which contains the amino acid sequence information for each protein. DeepLoc has spurred the development of numerous sequence-based models for subcellular localization that infer localization solely from amino acid sequences. Nevertheless, several studies have shown that ***spatial conformations*** play a critical role in determining subcellular localization patterns. For example, the nuclear localization signals of transcription factor NF-$\kappa$B are conditionally exposed only under specific structural conformations (Lusk et al., 2007). This demonstrates that the 3D structures of proteins, as dynamic regulatory elements, are the key to governing their subcellular localization.

---

[*]Corresponding author. Email: lsl19@tsinghua.org.cn
[†]Corresponding author. Email: zhfengbin@gmail.com

To fully leverage protein structural data, recent research has developed structure-based protein representation models. Benefiting from the emergence of AlphaFold2 (Jumper et al., 2021), which offers reliable structural predictions for a vast number of proteins, the structure-based methods learn representations directly from the spatial geometry of proteins. Such approaches have demonstrated impressive performance across a range of tasks, including protein classification (Jing et al., 2020; Zhang et al., 2022; Fan et al., 2022) and protein generation (Dauparas et al., 2022; Watson et al., 2023), showcasing their ability to capture complex structural patterns beyond what sequence alone can provide. These successful implementations underscore the substantial potential of incorporating structural information into subcellular localization prediction frameworks.

However, the existing subcellular localization datasets, such as DeepLoc, suffer from several limitations, which hinder the investigation of structure-based methods. Most notably, 1) they lack explicit protein 3D information, which is the key input to structure-based methods. Furthermore, 2) the current dataset typically uses coarse-grained compartment classifications, grouping subcellular areas into broad categories (*e.g.,* do not distinguish nuclear membrane and nucleoli in nucleus), which overlooks the unique localization characteristics and mechanisms associated with different organelles. Therefore, it leads to poor interpretability and great difficulty in discovering distinct patterns and underlying biological principles.

To address these limitations, we aim to construct a human protein subcellular localization dataset that can facilitate research on structure-based methods for localization prediction and enable the discovery of more specific and biologically relevant localization patterns. Specifically, we have two considerations for the dataset: 1) **Comprehensive 3D information**, which seeks to enhance the comprehensiveness of the dataset by recording detailed localization data from different databases and integrating 3D structural information of proteins, thereby bringing convenience and providing a unified evaluation benchmark for structure-based prediction models within the community; 2) **Fine-grained subcellular categorization**, which aims to incorporate finer-grained localization labels with annotations based on biological empirical evidence. As such, researchers are allowed to investigate protein localization patterns at a more detailed and functionally meaningful level.

To this end, we take the initiative of building a dataset called **CAPSUL** that simultaneously fulfills the two considerations. Specifically, to obtain the 3D information, we leverage AlphaFold2 to extract the Cartesian coordinates of the C$\alpha$ (alpha carbon) and utilize the FoldSeek to derive corresponding 3Di structural tokens for each protein, promoting structure understanding such as backbone conformation, folding patterns, and local structure. Moreover, to obtain comprehensive subcellular localization labels, we cross-reference each protein with annotation data from both the UniProt (Consortium, 2019) and Human Protein Atlas (HPA) (Thul et al., 2017) databases. Building upon the categories in the existing dataset DeepLoc, we further refine the subcellular area space by introducing 20 aggregated subcellular compartments, carefully curated and validated by domain experts. We extend several state-of-the-art (SOTA) protein representation models to this downstream task and evaluate their performance on CAPSUL. To facilitate future research, we investigate several potential optimization strategies for structure-based model training and make innovative use of the attention mechanism to enhance the interpretability of protein subcellular localization patterns by integrating Transformer modules into existing models. Empirical results on CAPSUL validate the necessity of 3D information incorporation and the potential of leveraging structure-based methods for causal biology pattern discovery on the subcellular localization task.

In summary, the contributions of this paper are threefold:

- We represent the first systematic attempt to construct a human protein subcellular localization dataset with comprehensive 3D information, fine-grained categorization of cell compartments, and cross-referenced localization labels with experiment-level annotations.
- We evaluate several SOTA baseline models on our proposed dataset CAPSUL, validating the positive influence of incorporating protein structural inputs.
- We investigate various training strategies to facilitate future exploration and enhance the interpretability for subcellular localization tasks by introducing the attention mechanism.

## 2 RELATED WORK

**Sequence-based protein representation learning.** Due to the relative ease of modeling protein amino acid sequences, early protein representation learning efforts typically relied solely on one-

dimensional sequence inputs. Examples include models based on CNN, LSTM, or ResNet architectures (Shanehsazzadeh et al., 2020; Rao et al., 2019). Subsequently, Transformer-based models have demonstrated strong performance, especially after large-scale pretraining, achieving impressive results across a range of downstream tasks (Rives et al., 2019; Lin et al., 2022; Madani et al., 2023). In parallel, various self-supervised approaches have further enhanced the model's ability to capture meaningful features from protein sequences without a vast number of annotations (Rives et al., 2019; Lin et al., 2023; Elnaggar et al., 2021; Lu et al., 2020; He et al., 2021). However, in the subcellular localization task, which is known to be closely linked to protein structure, sequence-only models fall short of capturing the full complexity of protein features. As a result, incorporating 3D structural information has become increasingly recognized as essential for achieving richer and more comprehensive protein representations.

**Structure-based protein representation learning.** Efforts to model protein structures have been explored from multiple perspectives, including representations at the protein surface level, residue level, and atomic level. The protein language model also starts to consider structural information as input to enhance its understanding of proteins (Hayes et al., 2025). These approaches have achieved impressive results in tasks such as protein design, structure generation, and function prediction (Gligorijević et al., 2021; Gainza et al., 2020; Hermosilla et al., 2020; Hsu et al., 2022). Among them, models based on Graph Convolutional Network (GCN) have demonstrated consistently strong performance across various downstream tasks, highlighting their ability to effectively capture and interpret structural information (Fan et al., 2022; Jing et al., 2020; Zhang et al., 2022). However, most of these models require atomic or residue-level coordinate inputs, which are often missing from current benchmark datasets. To address this gap, we aim to construct a dataset specifically for the task of subcellular localization that incorporates 3D structural information, facilitating both the application and evaluation of structure-based models.

**Subcellular localization dataset.** Although many prestigious and task-specific protein benchmarks exist (Rao et al., 2019; Kryshtafovych et al., 2023), their lack of subcellular localization annotations makes them inapplicable on this downstream task. To the best of our knowledge, the only well-known dataset for subcellular localization originates from the training data used in DeepLoc (Thumuluri et al., 2022). Building on this, the PEER framework established a benchmark to evaluate baseline models on that dataset (Xu et al., 2022). However, the absence of 3D structural information makes it impossible to assess the performance of structure-based models that have already shown significant promise. To address this gap, we aim to reorganize and enrich the existing dataset by incorporating high-quality 3D structural information alongside fine-grained subcellular localization annotations. We further evaluate a range of representative baseline models on this updated dataset, with the goal of establishing a leading benchmark for subcellular localization prediction.

## 3 CAPSUL DATASET

To construct the CAPSUL dataset that offers 1) diverse and accessible 3D structural information, and 2) both detailed and aggregated subcellular localization annotations, we follow a multi-step curation process, as illustrated in Figure 1.

### 3.1 PROCESSING OF PROTEIN SEQUENCE AND STRUCTURE DATA

**Collection and filter of protein data.** We first retrieve all predicted human protein structures from the AlphaFold2 database (Jumper et al., 2021; Varadi et al., 2024), totaling 20,504 unique proteins. To ensure data quality and relevance, we filter this set by retaining only proteins marked as active in the UniProt database (Consortium, 2019), one of the most comprehensive and authoritative protein databases with well-documented annotations, resulting in a refined set of 20,401 proteins.

**Removal of fragmented structure predictions.** Among the refined set, AlphaFold2 typically adopts a sliding-window strategy to long protein sequences that segments the sequence with overlapping fragments to predict protein structure. To avoid inconsistencies of predicted coordinates that may arise during the stitching of these fragmented protein structures, we exclude such proteins from the dataset. After this step, we obtain 20,181 proteins of high quality and good consistency.

**Extraction and preprocessing of protein features.** We preserve the full PDB files for each protein, the original files downloaded from AlphaFold, including the positions of backbone atoms, side

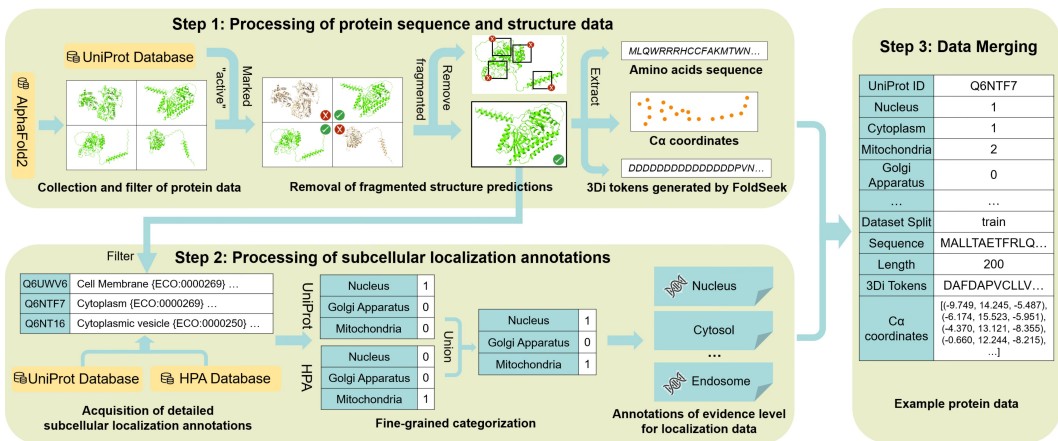

Figure 1: Procedures of CAPSUL dataset construction, including 3 key steps: Step 1 extracts and filters the sequence and structure data for each high-quality protein from AlphaFold2; Step 2 collects the annotations from UniProt and HPA for the resulting proteins in Step 1; Step 3 merges the structure data and the annotations for each protein, which consists of protein ID, localization annotations, amino acid sequence, sequence length, 3Di tokens, and C$\alpha$ coordinates, *etc.*

chains, and other relevant structural features essential for molecular modeling and analysis. The coordinates of C$\alpha$ atoms are extracted, which are important components for protein structure understanding. Furthermore, we employ the FoldSeek (Van Kempen et al., 2024) toolkit to tokenize the 3D structure of each amino acid. This provides a compact, informative structural representation that supports rapid, accurate modeling while reducing computational overhead, which has been empirically justified as effective and widely adopted in recent studies (Su et al., 2023).

Following the procedures above, we curate a dataset comprising 20,181 proteins, each labeled with amino acid sequence, C$\alpha$ coordinates, and 3Di tokens sequence. For the next step, we append localization annotations to each protein.

## 3.2 PROCESSING OF SUBCELLULAR LOCALIZATION ANNOTATIONS

**Acquisition of detailed subcellular localization annotations.** Based on the obtained proteins above, we collect the corresponding detailed subcellular localization annotations for human proteins from both the UniProt and HPA databases. This detailed dataset provides high-resolution localization annotations on widely accepted subcellular compartments, which is vital to facilitate research into the specific localization patterns within distinct organelles.

**Fine-grained categorization.** After that, we aggregate the dataset by adopting a refined categorization approach. Specifically, we consolidate the subcellular locations into 20 distinct categories inspired by DeepLoc's and HPA's subcellular localization classification scheme (Thumuluri et al., 2022; Thul et al., 2017), which is a fine-grained framework compared with DeepLoc's ten-class categorization. Then, the sublocations of 20 categories are specified separately, so the various terminologies in different databases can align with 20 unified categorizations. The entire procedure was conducted in accordance with a well-established cell biology textbook (Alberts et al., 2022) and further verified by domain experts, with detailed categorization information available in *Supp.* A.

**Annotations of evidence level for localization data.** To fulfill the various research demands for the reliability of localization labels, we further extract and consolidate annotations on the experimental evidence level. Specifically, for UniProt, each subcellular localization annotation is accompanied by an evidence code indicating the source of the localization label. Among them, the localization supported by experimental evidence (marked with the term *ECO:0000269*) is labeled as 1, indicating experimental validation. For the localization with other forms of evidence (*e.g.,* non-traceable author statement evidence), the label 2 is assigned. The label 0 is assigned to the localizations without evidence annotations. Moreover, since HPA primarily relies on experimental data obtained through immunofluorescence and confocal microscopy (Thul et al., 2017), we assign label 1 to all annotated

Table 1: Comparisons between existing datasets and CAPSUL.

| Dataset | Feartures | | Categoization | | Experimental Annotation |
|---|---|---|---|---|---|
| | Sequence | Structure | Aggregated | Detailed | |
| DeepLoc (Thumuluri et al., 2022) | ✓ | ✗ | ✓ | ✗ | ✗ |
| setHARD (Stärk et al., 2021) | ✓ | ✗ | ✓ | ✗ | ✗ |
| **CAPSUL** | ✓ | ✓ | ✓ | ✓ | ✓ |

Table 2: Statistics of CAPSUL.

| Number of Proteins | 20,181 | **Average Number of Annotations per Protein** | 2.51 | **Max Number of Annotations for Protein** | 14 | **Proportion of Experimental Annotations** | 0.857 |
|---|---|---|---|---|---|---|---|
| **Number of Annotations on:** | | | | | | | |
| Nucleus | 7,590 | Cytosol | 5,386 | Golgi Apparatus | 1,881 | Peroxisome | 110 |
| Nuclear Membrane | 452 | Cytoskeleton | 2,119 | Cell Membrane | 5,777 | Vesicle | 2,863 |
| Nucleoli | 1,641 | Centrosome | 1,000 | Endosome | 687 | Primary Cilium | 983 |
| Nucleoplasm | 6,786 | Mitochondria | 1,768 | Lipid Droplet | 94 | Secreted Proteins | 2,087 |
| Cytoplasm | 6,613 | Endoplasmic Reticulum | 1,710 | Lysosome/Vacuole | 453 | Sperm | 652 |

localizations and label 0 to the localizations without evidence annotations. During the union of UniProt and HPA datasets, we prioritize annotations with experimental evidence when available.

### 3.3 DATA MERGING

After the separate processing of protein sequence and structure data, along with the subcellular localization annotations, we merge the data to include complete information. In Figure 1, we present a sample record in CAPSUL, which consists of protein ID, localization annotations, amino acid sequence, sequence length, 3Di tokens, and C$\alpha$ coordinates, *etc.*

### 3.4 DATASET ANALYSIS

In summary, we construct a unified dataset comprising 20,181 proteins, each annotated with 20 sub-cellular localization labels. Our dataset CAPSUL provides a more comprehensive coverage compared to DeepLoc (Thumuluri et al., 2022) and setHARD (Stärk et al., 2021) in terms of involved features, localization categorization, and experimental annotations, which is shown in Table 1. The dataset is randomly split into training, validation, and test sets in a 70%:15%:15% ratio for training and evaluation. We present a statistical analysis of numerical features of our dataset in Table 2.

To ensure the **high quality** of our constructed CAPSUL dataset, we have incorporated three safeguards[1]: 1) **Reliable data sources**: reliable protein structures predicted by AlphaFold2 were utilized in CAPSUL, with high accuracy, strong consistency, and incorporation of available experimental data as templates in its prediction process (Jumper et al., 2021); the localization labels source UniProt, a world-leading database with the most comprehensive protein annotations from multiple resources, and HPA, a human-specific protein database offering high-resolution and experiment-validated data. 2) **Strict validation and filtering**: we perform a series of validation and filtering steps on human proteins to exclude fragmented AlphaFold structures, which could introduce inconsistent coordinate information, and to remove proteins annotated as inactive in UniProt, thereby ensuring the reliability of subcellular localization annotations; 3) **Evidence-level support**: we incorporate annotations indicating whether experimental validation exists for the localization labels, thereby enhancing their credibility and catering to diverse research needs.

## 4 EXPERIMENTS

### 4.1 BASELINE MODELS

To study how existing methods perform on our proposed dataset[2], we evaluate 1) **DeepLoc 2.1** (Ødum et al., 2024), one of the most well-known tools dedicated to subcellular localization. It leverages the pre-trained protein language model ESM-1b (Rives et al., 2021) and provides predictions across ten subcellular compartments. Besides, we evaluate existing representative protein

---

[1]For a detailed analysis of the data reliability in the dataset, please refer to *Supp.* B.

[2]The detailed descriptions and hyperparameter settings of all baseline models are provided in *Supp.* C.

representation methods for the subcellular localization task, including sequence-based and structure-based methods.

**Sequence-based models**. Since existing sequence-based works are not specifically designed for subcellular tasks, we extend the widely adopted pre-trained protein language model 2) **ESM-2** (650M parameters) (Lin et al., 2022) and its latest iteration, 3) **ESM-C** (600M parameters) (ESM Team, 2024). We adopt the sequence encoder module from existing methods to obtain protein representation, and extend it with a localization classifier, as detailed in the following.

- *Sequence Encoder*. For each protein, we have its amino acid sequence represented as $S = (s_1, s_2, \ldots, s_n) \in \mathbb{R}^{n \times 1}$, where $s_i$ denotes the $i$-th residue and $n$ is the length of the protein. We then apply the sequence encoder $f_{\text{seq}}(\cdot)$ of existing work to obtain contextual embeddings, $H = f_{\text{seq}}(S)$, where $H = (h_1, h_2, \ldots, h_n) \in \mathbb{R}^{n \times d}$, and $h$ is the per-residue embeddings of dimension $d$. To obtain a fixed-length representation for the entire protein, we apply mean pooling and generate a global representation $\bar{h} = \frac{1}{n} \sum_{i=1}^{n} h_i, \bar{h} \in \mathbb{R}^d$.

- *Localization Classifier*. To predict subcellular localization, we leverage an MLP classifier $\phi(\cdot)$ on top of sequence encoder, *i.e.*, $\hat{y} = \phi(\bar{h})$, where $\hat{y} \in \mathbb{R}^m$ is a multi-label prediction vector and $m$ denotes the total number of predicted subcellular compartments.

**Structure-based models**. We consider 4) **CDConv** (Fan et al., 2022) and 5) **GearNet-Edge** (Zhang et al., 2022), two representative GCN baselines in protein representation task. We adopt the GCN-based structure encoder and extend it with an additional Transformer encoder to enhance interpretability. We also evaluate 6) **FoldSeek** (Van Kempen et al., 2024), which leverages a pre-trained structure tokenizer to encode the 3D structural information of each residue into a sequence of structure tokens. The outputs of the above models are then averaged and processed through a localization classifier for prediction.

- *Structure Encoder*. We represent a protein's 3D structure as a graph $G = (V, E)$, where each node $v_i \in V$ corresponds to the $i$-th residue (typically using the $C\alpha$ atom position), and edges $(v_i, v_j) \in E$ are defined based on spatial or sequential adjacency. Each node $v_i$ is initialized with a feature vector $x_i \in \mathbb{R}^d$ including its positional information. Then we employ different graph encoders to capture higher-order topological relationships and produce updated representations $(h_1, \ldots, h_n)$. The protein-level embedding is then obtained via global pooling $\bar{h} = \frac{1}{n} \sum_{i=1}^{n} h_i$.

- *Localization Classifier*. We then obtain the final prediction $\hat{y} = \phi(\bar{h})$, as described above.

**Extension of structure-based models**. We also extend three novel methods, 7) **Graph Transformer** (Rampášek et al., 2022), 8) **Graph Mamba** (Gu & Dao, 2023), and 9) **Graph Diffusion** (Yang et al., 2023) to this task. The Graph Transformer employs attention mechanisms over graph-structured data, enabling the model to effectively capture both local and global dependencies among residues. Graph Mamba, on the other hand, incorporates selective state space models into graph learning, which facilitates long-range information propagation with improved efficiency. For Graph Diffusion, it leverages diffusion processes over graph-structured data to propagate information across nodes. To improve classification performance on minor categories, we further incorporate a contrastive loss mechanism into the CDConv model, *i.e.,* 10) **CDConv with Contrastive Loss**, aiming to enhance the similarity between representations of positive protein pairs and thereby encourage the learning of distinctive localization features. We also explore a fusion model that combines representative structure-based models with sequence-based pretrained protein language models, *i.e.,* 11) **ESM-C+CDConv Fusion Model**, investigating both early and late fusion strategies to integrate structural information into large-scale sequence models.

**Optimization**. To optimize the models, we adopt the Binary Cross Entropy (BCE) loss, defined as $\mathcal{L}_{\text{BCE}} = -\frac{1}{m} \sum_{i=1}^{m} [y_i \log(\hat{y}_i) + (1 - y_i) \log(1 - \hat{y}_i)]$, where $m$ is the number of classes, $y_i \in \{0, 1\}$ is the label for class $i$, and $\hat{y}_i \in (0, 1)$ is the predicted probability.

## 4.2 BENCHMARK OVERALL RESULTS AND DISCUSSION

Given the class imbalance in each location (*i.e.*, the proportion of proteins localized to each subcellular compartment is often small), we consider the widely used evaluation metrics in this task: Precision, Recall, and F1-score (Jiang et al., 2021; Thumuluri et al., 2022). In addition, we utilize micro-averaged and macro-averaged F1-score to evaluate the overall performance across different

Table 3: Overall performance of sequence-based, structure-based methods on CAPSUL.

| Subcellular Locations | Sequence-based Methods | | | | | | Structure-based Methods | | |
|---|---|---|---|---|---|---|---|---|---|
| | DeepLoc 2.1 | ESM-2 650M | ESM-2 650M$^f$ | ESM-C 600M | ESM-C 600M$^f$ | ESM-C 600M$^0$ | FoldSeek | CDConv$^t$ | GearNet-Edge$^t$ |
| **F1-score** | | | | | | | | | |
| Nucleus | 0.152 | - | 0.609 | **0.649** | 0.648 | 0.555 | 0.484 | 0.620 | 0.521 |
|    Nuclear Membrane | / | - | - | - | - | - | - | - | - |
|    Nucleoli | / | - | - | 0.091 | 0.039 | 0.024 | - | 0.147 | 0.121 |
|    Nucleoplasm | / | - | 0.562 | 0.621 | 0.623 | 0.500 | 0.433 | 0.583 | 0.515 |
| Cytoplasm | 0.154 | - | 0.248 | 0.536 | **0.551** | 0.438 | 0.174 | 0.483 | 0.495 |
|    Cytosol | / | - | - | 0.392 | 0.380 | 0.169 | 0.003 | 0.353 | 0.385 |
|    Cytoskeleton | / | - | 0.006 | 0.251 | 0.205 | 0.048 | 0.070 | 0.135 | 0.228 |
| Centrosome | / | - | - | 0.014 | - | - | - | - | 0.127 |
| Mitochondria | 0.120 | - | 0.317 | 0.562 | 0.544 | 0.099 | - | 0.476 | 0.318 |
| Endoplasmic Reticulum | 0.121 | - | - | 0.351 | 0.333 | 0.059 | - | 0.292 | 0.279 |
| Golgi Apparatus | 0.061 | - | - | 0.099 | 0.027 | - | - | 0.073 | 0.026 |
| Cell Membrane | 0.142 | - | 0.555 | 0.631 | 0.648 | 0.372 | 0.343 | 0.562 | 0.556 |
| Endosome | / | - | - | 0.018 | - | - | - | - | 0.067 |
| Lipid Droplet | / | - | - | - | - | - | - | - | - |
| Lysosome / Vacuole | **0.118** | - | - | - | - | - | - | - | 0.073 |
| Peroxisome | **0.131** | - | - | - | - | - | - | - | - |
| Vesicle | / | - | - | 0.009 | - | 0.005 | - | 0.027 | 0.068 |
| Primary Cilium | / | - | - | **0.164** | 0.112 | - | - | - | 0.147 |
| Secreted Proteins | 0.191 | - | 0.713 | **0.826** | 0.797 | 0.433 | 0.328 | 0.767 | 0.687 |
| Sperm | / | - | - | 0.052 | 0.070 | - | - | - | 0.086 |
| **Micro Avg F1-score** | / | - | 0.375 | **0.495** | 0.492 | 0.338 | 0.248 | 0.452 | 0.417 |
| **Macro Avg F1-score** | / | - | 0.150 | **0.263** | 0.249 | 0.135 | 0.092 | 0.226 | 0.235 |
| **Micro Avg Precision** | / | - | 0.647 | 0.690 | 0.693 | 0.598 | 0.605 | 0.632 | 0.546 |
| **Micro Avg Recall** | / | - | 0.264 | 0.386 | 0.382 | 0.236 | 0.156 | 0.352 | 0.337 |

| Subcellular Locations | Extension of Structure-based Methods | | | | | |
|---|---|---|---|---|---|---|
| | Graph Transformer | Graph Mamba | Graph Diffusion | CDConv$^t$ with Contrastive Loss | ESM-C+CDConv Early Fusion | ESM-C+CDConv Late Fusion |
| **F1-score** | | | | | | |
| Nucleus | 0.597 | 0.559 | 0.624 | 0.592 | 0.643 | 0.645 |
|    Nuclear Membrane | - | **0.037** | - | - | - | - |
|    Nucleoli | **0.203** | 0.168 | 0.047 | 0.140 | 0.125 | 0.153 |
|    Nucleoplasm | 0.552 | 0.502 | 0.578 | 0.556 | **0.643** | 0.617 |
| Cytoplasm | 0.393 | 0.418 | 0.503 | 0.480 | 0.455 | 0.515 |
|    Cytosol | 0.248 | **0.426** | 0.288 | 0.250 | 0.157 | 0.370 |
|    Cytoskeleton | 0.042 | 0.270 | 0.099 | 0.243 | 0.100 | **0.287** |
| Centrosome | - | **0.181** | 0.014 | - | - | 0.037 |
| Mitochondria | 0.475 | 0.341 | 0.303 | 0.468 | **0.563** | 0.557 |
| Endoplasmic Reticulum | 0.184 | 0.059 | 0.150 | 0.361 | **0.446** | - |
| Golgi Apparatus | 0.041 | 0.185 | - | 0.156 | **0.249** | - |
| Cell Membrane | 0.547 | 0.540 | 0.496 | 0.539 | 0.629 | **0.673** |
| Endosome | - | **0.100** | - | 0.034 | 0.054 | - |
| Lipid Droplet | - | - | - | - | - | - |
| Lysosome / Vacuole | - | - | - | - | 0.026 | - |
| Peroxisome | - | - | - | - | - | - |
| Vesicle | 0.044 | **0.135** | 0.018 | 0.027 | 0.067 | - |
| Primary Cilium | 0.012 | 0.088 | - | 0.036 | - | 0.115 |
| Secreted Proteins | 0.705 | 0.557 | 0.623 | 0.780 | 0.819 | 0.725 |
| Sperm | 0.018 | **0.130** | - | 0.018 | - | - |
| **Micro Avg F1-score** | 0.410 | 0.411 | 0.424 | 0.435 | 0.470 | 0.476 |
| **Macro Avg F1-score** | 0.203 | 0.235 | 0.187 | 0.234 | 0.249 | 0.235 |
| **Micro Avg Precision** | 0.637 | 0.414 | 0.596 | 0.650 | **0.710** | 0.634 |
| **Micro Avg Recall** | 0.302 | **0.408** | 0.329 | 0.326 | 0.351 | 0.381 |

$^f$We finetune the pre-trained protein language model. $^t$The original MLP is replaced by Transformer layers. $^0$The parameters of ESM-C are initialized randomly. "/" indicates that DeepLoc 2.1 does not support prediction for that location, and therefore, average metrics are not considered in this case. "–" indicates that no prediction is made for that location. **Bold** value indicates the best results.

Table 4: Ablation study of CDConv and GearNet-Edge to randomly sample C$\alpha$ coordinates.

| | CDConv$^t$ (random C$\alpha$ coordinates) | CDConv$^t$ | GearNet-Edge$^t$ (random C$\alpha$ coordinates) | GearNet-Edge$^t$ |
|---|---|---|---|---|
| Micro Avg F1-score | 0.329 | **0.452** | 0.348 | **0.417** |
| Micro Avg Precision | 0.586 | **0.632** | 0.450 | **0.546** |
| Micro Avg Recall | 0.229 | **0.352** | 0.283 | **0.337** |

$^t$The original MLP is replaced by Transformer layers. **Bold** value indicates the better result for each baseline.

categories. The overall performance[3] of all baselines on our proposed dataset is presented in Table 3, from which we have the following observations:

---

[3]The detailed results *w.r.t.* Precision and Recall, including other experimental results mentioned later in the main text, are provided in *Supp.* D.

**Large pre-training benefits sequence-based methods for subcellular location prediction**. Among all sequence-based methods, ESM-C generally obtains higher F1-scores than ESM-2. We believe this is attributed to the extensive data and training compute used in the ESM-C pre-training, which facilitates a better representation of the protein's sequence features. Similar observations are also seen in (Hayes et al., 2025). Besides, this hypothesis can be further confirmed by the significantly inferior performance of ESM-C 600M$^0$, *i.e.,* without pre-training, than the pre-trained ESM-C. On the other hand, it is expected that DeepLoc yields inferior performance due to its overlook of the fine-grained categorization during pre-training, which may result in its inability to sufficiently differentiate the representations of proteins in multi-label classification tasks (Hong et al., 2023). This further validates the necessity of detailed categorizations of subcellular locations in CAPSUL.

**The 3D structure is essential for subcellular localization task.** Despite that structure-based methods slightly fall behind the pre-trained ESM-C, both CDConv and GearNet-Edge outperform the ESM-C 600M$^0$ in most cases. Also, a group of ablation studies is conducted on CDConv and GearNet-Edge, with coordinates randomly sampled from each protein's spatial range. As shown in Table 4, randomly sampling the input of protein 3D structural data leads to a significant drop in model performance. These two results validate that structural information plays a decisive role in determining subcellular localization. Besides, CDConv demonstrates the strongest overall performance among the structure-based models, justifying the effectiveness of relative distance and the dynamic radius for convolution. Nevertheless, the inferior performance of FoldSeek may be due to the lack of sequence information and its coarse tokenization of structural information.

**The models generally demonstrate better performance on subcellular locations with larger localization sample sizes.** For classes with a large number of localization samples (*e.g.*, nucleus), most models tend to demonstrate relatively strong predictive performance. In contrast, for underrepresented classes (*e.g.*, lipid droplet), the prediction performance is generally poor, with some classes even failing to produce any correctly identified proteins. This is a common outcome in imbalanced multi-label classification tasks, as the standard BCE loss tends to neglect fewer-number labels. Additionally, potential conflicts among multiple optimization targets may further exacerbate this issue. To address these challenges, we conduct in-depth analysis in Section 4.3.1 and 4.3.2, exploring strategies such as reweighting and single-label classification to mitigate the effects of class imbalance and task conflict.

**Structure-based models showcase their potential to capture non-trivial patterns for subcellular locations with few samples**. Graph Mamba and GearNet-Edge tend to perform better on certain classes with smaller localization sample sizes compared with sequence-based models. We believe that this is because of the relational message passing layer adopted in them, which uniquely models different spatial interactions among residues. This demonstrates that structure-based models showcase potential to identify specific structural features that are indicative of localization to a particular organelle, thus achieving a notably good performance. Further investigation on the patterns with intuitive biological interpretability captured by the structure-based model can be found in Section 4.3.3.

**Contrastive learning and fusion strategies demonstrate strong potential on the baseline models.** We observe that the introduction of contrastive loss to CDConv improves performance on several minority classes (*i.e.,* the notable F1-score improvements on macro-average level and certain categories such as Endosome, Primary Cilium, and Golgi Apparatus compared to the original CDConv). We attribute this to the contrastive learning paradigm, which encourages the model to maximize embedding similarity for positive pairs and to capture shared characteristics within minority-class positive samples through the contrastive objective. Also, although the fusion model slightly underperforms ESM-C on average metrics, it achieves the best performance across multiple subcellular compartments among all baselines, highlighting the considerable potential of integrating protein structural information into sequence-based protein language models.

## 4.3 IN-DEPTH ANALYSIS

### 4.3.1 PROTEIN IMBALANCE MITIGATION VIA REWEIGHTING

**Reweighting Schemes**. In this task, for each subcellular location, the number of positive samples (*i.e.*, proteins localized to that compartment) is substantially smaller than the number of negative samples (*i.e.*, proteins not localized to that compartment). Reweighting is a widely used strategy to

Table 5: Performance of ESM-C 600M, CDConv, and GearNet-Edge with reweighting scheme.

| Subcellular Locations | ESM-C 600M | CDConv[t] | GearNet-Edge[t] | Subcellular Locations | ESM-C 600M | CDConv[t] | GearNet-Edge[t] |
|---|---|---|---|---|---|---|---|
| **F1-score** | | | | **F1-score** | | | |
| Nucleus | 0.630 | **0.625** | 0.618 | Endosome | - | **0.114** | **0.150** |
| Nuclear Membrane | - | **0.062** | 0.058 | Lipid Droplet | 0.235 | 0.023 | **0.111** |
| Nucleoli | - | **0.188** | 0.224 | Lysosome/Vacuole | - | **0.175** | 0.111 |
| Nucleoplasm | 0.576 | **0.607** | 0.574 | Peroxisome | 0.190 | 0.072 | 0.108 |
| Cytoplasm | 0.500 | **0.582** | 0.544 | Vesicle | - | **0.288** | 0.281 |
| Cytosol | **0.133** | 0.495 | 0.484 | Primary Cilium | 0.024 | **0.167** | **0.176** |
| Cytoskeleton | 0.083 | **0.292** | 0.294 | Secreted Proteins | 0.778 | 0.564 | 0.614 |
| Centrosome | - | **0.160** | 0.175 | Sperm | - | **0.120** | **0.125** |
| Mitochondria | 0.481 | 0.297 | 0.313 | | | | |
| Endoplasmic Reticulum | - | **0.308** | **0.345** | | | | |
| Golgi Apparatus | - | **0.246** | 0.238 | **Micro Avg F1-score** | 0.429 | 0.381 | **0.453** |
| Cell Membrane | 0.566 | 0.560 | 0.536 | **Macro Avg F1-score** | 0.210 | **0.297** | **0.304** |

[t]The original MLP is replaced by Transformer layers. "–" indicates that no prediction is made for that location. **Bold** value indicates that it improves compared with the result without reweighting.

Table 6: Performance of ESM-C 600M, CDConv, and GearNet-Edge with single-label classification.

| Subcellular Locations | ESM-C 600M | CDConv[t] | GearNet-Edge[t] | Subcellular Locations | ESM-C 600M | CDConv[t] | GearNet-Edge[t] |
|---|---|---|---|---|---|---|---|
| **F1-score** | | | | **F1-score** | | | |
| Nuclear Membrane | - | **0.052** | 0.042 | Lysosome/Vacuole | 0.115 | - | 0.162 |
| Nucleoli | 0.267 | 0.151 | 0.228 | Peroxisome | 0.054 | - | 0.023 |
| Centrosome | 0.184 | 0.089 | 0.167 | Vesicle | 0.068 | 0.230 | 0.268 |
| Golgi Apparatus | 0.280 | 0.114 | 0.210 | Primary Cilium | 0.253 | 0.097 | 0.171 |
| Endosome | 0.167 | 0.049 | 0.126 | Sperm | 0.159 | 0.068 | 0.117 |
| Lipid Droplet | 0.021 | - | 0.051 | | | | |

[t]The original MLP is replaced by Transformer layers. "–" indicates that no prediction is made for that location. **Bold** value indicates that it improves compared with the result from multi-label classification.

address class imbalance by reducing the bias toward majority classes. Inspired by previous work on class-level reweighting, we evaluate three reweighting schemes. 1) Inverse frequency reweighting (Cao et al., 2019), *i.e.,* $w_c = \frac{1}{f_c}$. 2) Log-inverse frequency reweighting (Cui et al., 2019), *i.e.,* $w_c = \frac{1}{\log(1+f_c)}$. 3) Focal loss (Lin et al., 2017), which is defined as

$$\mathcal{L}_c = -w_c \cdot \sum_i \left[ y_{ic} \cdot (1 - \hat{y}_{ic})^\gamma \log(\hat{y}_{ic}) + (1 - y_{ic}) \cdot \hat{y}_{ic}^\gamma \log(1 - \hat{y}_{ic}) \right],$$

where $f_c$ is the frequency of positive samples in class $c$, $w_c$ is the computed class-specific weight, $y_{ic} \in \{0, 1\}$ denotes the ground truth label for sample $i$ and class $c$, $\hat{y}_{ic} \in (0, 1)$ is the predicted probability, and $\gamma$ is the focusing parameter. It deserves attention that the $w_c$ in the focal loss strategy is chosen from either inverse or log-inverse frequency weight.

**Results**. We apply the three reweighting schemes on three competitive models (ESM-C, CDConv, and GearNet-Edge) and report the best results for each model in Table 5. From the results, we observe that the two structure-based baseline models exhibit substantial improvements under reweighting strategies, especially for the higher Precision across underrepresented categories. In particular, CDConv and GearNet-Edge successfully identify positive instances for every class. These findings highlight that reweighting can significantly enhance model performance on minority classes, especially for structure-based models.

### 4.3.2 SINGLE-LABEL CLASSIFICATION

To explore how different methods perform on each subcellular location respectively, we adopt the single-label setting, aiming to mitigate the potential conflict between optimization across different classes. In this setting, we train separate binary classifiers for each subcellular localization category with ESM-C, CDConv, and GearNet-Edge. We apply this single-label prediction framework specifically to those subcellular localization classes where the F1-score of at least one of the models (ESM-C, CDConv, or GearNet-Edge) is lower than 0.1. Our goal is to shift the model's attention toward underrepresented classes and improve the predictive influence of positive samples.

From the results in Table 6, we observe 1) notable improvements in the prediction performance of previously underperforming classes, particularly for GearNet-Edge. However, 2) ESM-C and CDConv still fail to generate any predictions for a few categories, primarily due to the extremely low proportion of positive samples (ranging from 0.5% to 3%). Given that such severe class imbalance is a common challenge in subcellular localization tasks, we consider the single-label prediction

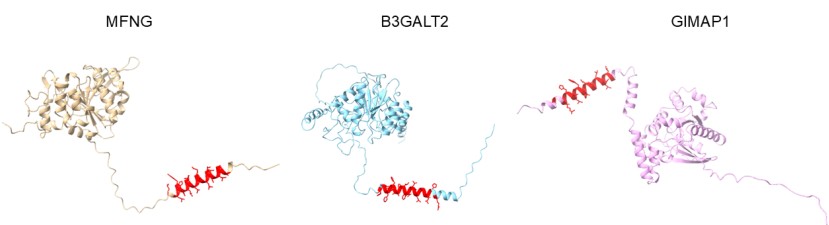

Figure 2: Visualization of the top 20 attention-scored residues of the three representative proteins.

strategy a promising and practical solution. Moreover, this approach lays the groundwork for future research focused on identifying localization patterns specific to individual subcellular compartments.

### 4.3.3 BIOLOGICAL INTERPRETABILITY

We analyze a CDConv model on Golgi apparatus prediction with an exceptional precision of 100%. Specifically, with our novel attempt of the Transformer module extended to the GCN-based models, we identify and visualize the tokens (*i.e.*, residues) that receive the 20 highest attention weights in Figure 2, offering insights into which structure the model considers most decisive for subcellular localization. We find that the model consistently highlights similar $\alpha$-helix spatial conformation, such as residues 8-27 of MFNG, residues 24-45 of B3GALT2, and residues 273-292 at the C-terminus of GIMAP1. Remarkably, these findings show strong concordance with prior experimental evidence (Paulson & Colley, 1989; Linstedt et al., 1995). It is highlighted that despite significant sequence divergence, the model specifically focuses on $\alpha$-helix transmembrane domains (20-30 amino acids in length) that maintain consistent topological orientations across all targets. Recent studies have demonstrated that the topological conformation of transmembrane domains can influence Golgi localization by regulating electrostatic potential gradients in transmembrane regions and lipid membrane anchoring efficiency (Cosson et al., 2013; Hanulova & Weiss, 2012; Bian et al., 2024). This evidence not only confirms the model's capability for structural pattern recognition beyond sequence similarity but also provides theoretical support for its structural identification mechanisms.

## 5 CONCLUSION AND FUTURE WORK

We pointed out the crucial importance of constructing a subcellular localization benchmark with protein 3D information to facilitate the investigation of structure-based models for the subcellular localization task. To achieve this, we constructed a benchmark called CAPSUL that contains comprehensive structural information and fine-grained annotations of 20 categories of subcellular compartments with biological experiment evidence labels. Based on CAPSUL, we evaluated SOTA sequence-based and structure-based models as well as their feasible optimization strategies, demonstrating the effectiveness of incorporating protein structural information. Moreover, a case study on Golgi apparatus validates the biology-aligned interpretability of structure-based models trained on a specific fine-grained subcellular location, supported by CAPSUL. This work proposes a comprehensive human protein benchmark with 3D information and fine-grained annotations for subcellular localization. Based on CAPSUL, we highlight several research directions that are worth future exploration: 1) To fully leverage structural information, aligning or disentangling the understanding across different dimensions (*i.e.,* amino acid sequence, C$\alpha$, and 3Di) specifically for subcellular localization is a promising direction. 2) Causal discovery on the relationship between 3D structure and subcellular localization is worthwhile to be explored on CAPSUL, with the goal of establishing direct links to underlying biological principles.

### ACKNOWLEDGMENT

Shulin Li was supported by China Postdoctoral Science Foundation (grant number BX20240186 and 2024M761616) and the Shuimu Tsinghua Scholar Program.

ETHICS STATEMENT

This research presents a dataset and benchmark for protein subcellular localization prediction using AI methods. We confirm that our work raises no ethical concerns as it involves only the analysis of publicly available protein data, with no human subjects, animal experiments, or biological interventions. We have fully considered the potential societal impacts and do not foresee any direct, immediate, or negative consequences. We are committed to the ethical dissemination of our findings and encourage their responsible use.

REPRODUCIBILITY STATEMENT

All the results in this work are reproducible. The access to the necessary code and complete dataset can be found in *Supp.* L. We discuss the experimental details in *Supp.* C, including implementation details such as the hyperparameters chosen for each experiment, to help reproduce our results. Additionally, further experimental results, detailed dataset interpretations, and usage guidelines are provided in *Supp.* E, F, and J to facilitate better understanding and utilization of our dataset.

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

# Supplementary Material for CAPSUL: A Comprehensive Human Protein Benchmark for Subcellular Localization

## A    DATASET CONSTRUCTION

### A.1    SUBCELLULAR LOCATION CATEGORIZATION AND TERMINOLOGY MAPPING

To facilitate model classification, we first categorize the detailed subcellular localizations of proteins. Existing datasets often use coarse-grained classifications (*e.g.,* DeepLoc categorizes subcellular locations into 10 broad classes). However, since each subcellular compartment typically follows distinct localization patterns, such coarse categorizations can hinder the model's ability to capture consistent intra-class features, ultimately leading to reduced prediction accuracy. Moreover, coarse-grained classification also hinders researchers from exploring localization mechanisms specific to finer subcellular compartments. Inspired by the subcellular location categories in HPA and DeepLoc, we propose a finer-grained classification scheme consisting of 20 subcellular categories. Notably, "Nucleus" and "Cytoplasm" categories serve as umbrella terms for several finer locations to ensure compatibility with DeepLoc during evaluation.

When aligning protein localization annotations from the UniProt and HPA databases to our refined categorization, we observe inconsistencies in terminology (*e.g.,* "Cell Membrane" in UniProt versus "Plasma Membrane" in HPA). To resolve such discrepancies, we refer to the prestigious textbook Molecular Biology of the Cell (7th Edition) (Alberts et al., 2022) and create a unified mapping, as shown in Table 7, which allows for consistent categorization across the two databases.

Domain experts were extensively engaged to ensure and validate the accuracy of the classification standards and data alignment procedures. We invited cell biologists from several prestigious universities and research institutes to review and revise the dataset, which ensures that CAPSUL is firmly grounded in cell biology. All of them have over eight years of research experience in their field. They are rigorously involved throughout the entire process, including 1) curating authoritative datasets, 2) determining primary subcellular localizations, and 3) validating the biological plausibility of localization assignments.

Through the above processes, we have established a fine-grained subcellular localization classification standard and successfully unified annotations from multiple databases under a unified labeling framework.

### A.2    DATASET SPLITS

To construct separate datasets for training, validating, and testing, we randomly split the original dataset into three subsets in a 70%: 15%: 15% ratio, each containing 14,126, 3,027, 3,028 proteins. The partitioning of different protein data used in our experiments is also available in the CAPSUL dataset. The number of labels for each subcellular location in three subsets is shown in Table 8. Although the data is randomly assigned to different subsets, we have verified the distribution characteristics among classes to maintain a similar proportional relationship, ensuring balance and representativeness across the subsets.

## B    DATASET RELIABILITY

### B.1    OVERVIEW OF DATA SOURCES

In Section 3, we provide a detailed description of the data preprocessing procedures implemented to ensure the high quality of CAPSUL. Here, we would like to emphasize that the data sources themselves are highly reliable. Specifically, the protein-related data used in this study were primarily obtained from the following databases:

**AlphaFold.** AlphaFold provides protein structural data in CAPSUL. 1) AlphaFold has already **incorporated experimentally resolved structures of proteins as templates** during its prediction process (Jumper et al., 2021). AlphaFold explicitly describes how its pipeline automatically searches

Table 7: Categorization of CAPSUL and terminology mapping between HPA and Uniprot.

| 20 fine-grained categories | HPA | UniProt |
|---|---|---|
| Nucleus | | |
|   Nuclear Membrane | Nuclear membrane | Nucleus membrane, Nucleus envelope, Nucleus inner membrane, Nucleus outer membrane |
|   Nucleoli | Nucleoli, Nucleoli fibrillar center, Nucleoli rim | Nucleolus |
|   Nucleoplasm | Kinetochore, Mitotic chromosome, Nuclear bodies, Nuclear speckles, Nucleoplasm | Nucleus matrix, Nucleus lamina, Chromosome, Nucleus speckle |
| Cytoplasm | | |
|   Cytosol | Aggresome, Cytoplasmic bodies, Cytosol, Rods Rings | Cytosol |
|   Cytoskeleton | Actin filaments, Cleavage furrow, Focal adhesion sites, Cytokinetic bridge, Microtubule ends, Microtubules, Midbody, Midbody ring, Mitotic spindle, Intermediate filaments | Cytoskeleton |
| Centrosome | Centriolar satellite, Centrosome | Centrosome |
| Mitochondria | Mitochondria | Mitochondrion, Mitochondrion envelop, Mitochondrion inner membrane, Mitochondrion outer membrane, Mitochondrion membrane, Mitochondrion matrix, Mitochondrion intermembrane space |
| Endoplasmic Reticulum | Endoplasmic reticulum | Endoplasmic reticulum, Endoplasmic reticulum membrane, Endoplasmic reticulum lumen, Microsome, Rough endoplasmic reticulum, Smooth endoplasmic reticulum, Sarcoplasmic reticulum |
| Golgi Apparatus | Golgi apparatus | Golgi apparatus, Golgi apparatus membrane, Golgi apparatus lumen |
| Cell Membrane | Cell Junctions, Plasma membrane | Cell membrane, Apical cell membrane, Apicolateral cell membrane, Basal cell membrane, Basolateral cell membrane, Lateral cell membrane, Cell projection |
| Endosome | Endosomes | Endosome |
| Lipid Droplet | Lipid droplets | Lipid droplet |
| Lysosome/Vacuole | Lysosomes | Lysosome, Vacuole, Vacuole lumen, Vacuole membrane, Lysosome lumen, Lysosome membrane |
| Peroxisome | Peroxisomes | Peroxisome, Peroxisome matrix, Peroxisome membrane |
| Vesicle | Vesicles | Vesicle |
| Primary Cilium | Basal body, Primary cilium, Primary cilium tip, Primary cilium transition zone | Cilium |
| Secreted Proteins | Secreted Proteins | Secreted |
| Sperm | Acrosome, Annulus, Calyx, Connecting piece, End piece, Equatorial segment, Flagellar centriole, Mid piece, Perinuclear theca, Principal piece | Acrosome, Calyx, Perinuclear theca |

Table 8: Label counts for training, validation, and test set of CAPSUL.

| Subcellular Locations | Counts | | | |
|---|---|---|---|---|
| | Training Set | Validation Set | Test Set | **Sum** |
| Nucleus | 5,312 | 1,128 | 1,150 | 7,590 |
|   Nuclear Membrane | 313 | 63 | 76 | 452 |
|   Nucleoli | 1,143 | 249 | 249 | 1,641 |
|   Nucleoplasm | 4,751 | 1,007 | 1,028 | 6,786 |
| Cytoplasm | 4,652 | 984 | 977 | 6,613 |
|   Cytosol | 3,787 | 811 | 788 | 5,386 |
|   Cytoskeleton | 1,499 | 302 | 318 | 2,119 |
| Centrosome | 713 | 140 | 147 | 1,000 |
| Mitochondria | 1,247 | 259 | 262 | 1,768 |
| Endoplasmic Reticulum | 1,146 | 275 | 289 | 1,710 |
| Golgi Apparatus | 1,323 | 271 | 287 | 1,811 |
| Cell Membrane | 4,022 | 863 | 892 | 5,777 |
| Endosome | 466 | 113 | 108 | 687 |
| Lipid Droplet | 63 | 16 | 15 | 94 |
| Lysosome/Vacuole | 313 | 65 | 75 | 453 |
| Peroxisome | 71 | 20 | 19 | 110 |
| Vesicle | 2,019 | 404 | 440 | 2,863 |
| Primary Cilium | 699 | 123 | 161 | 983 |
| Secreted Proteins | 1,477 | 317 | 293 | 2,087 |
| Sperm | 444 | 99 | 109 | 652 |

the PDB for experimentally resolved structures, selecting up to four structural templates, and maps atom coordinates from those templates to the target sequence during inference. These coordinates are used as template inputs alongside MSA-based evolutionary information, enabling AlphaFold to leverage high-quality experimental structural data in its predictions. 2) AlphaFold-predicted structures have been demonstrated to achieve exceptionally **high accuracy**, competitive with experimental data. AlphaFold was entered for CASP14, and shows that it achieves accuracy competitive with experiment in a majority of cases. Specifically, the median backbone accuracy of its predictions is 0.96 Å r.m.s.d.$_{95}$ (C$\alpha$ root-mean-square deviation at 95% residue coverage), which is often within the margin of error of experimental structures (Jumper et al., 2021). 3) AlphaFold provides full-length protein structures containing complete structural information, which minimizes the potential negative influence of structural variability caused by different versions of experimental protein data. This choice allows us to maintain **a high level of consistency** across the CAPSUL dataset.

**UniProt.** UniProt provides protein localization annotation and evidence-level annotations in CAPSUL. UniProt serves as one of the most authoritative and widely used protein knowledge bases, integrating sequence, functional, and localization information across a broad spectrum of species. In particular, the manually curated Swiss-Prot section is recognized for its rigorous curation standards, where annotations (including subcellular localization annotations) are derived from authoritative experimental studies and peer-reviewed literature, complemented by computational analyses and homology-based inferences. Each localization entry is systematically annotated with evidence codes that explicitly denote whether the information originates from direct experimental validation, literature reports, or computational prediction, thereby providing transparency and traceability of the data source. This evidence-based framework ensures that localization annotations are not only comprehensive but also of consistently high quality.

**Human Protein Atlas (HPA).** HPA provides protein localization annotation and subcellular categories reference in CAPSUL. HPA provides a unique and experimentally grounded resource for human protein subcellular localization. Its Subcellular Atlas is built upon systematic immunofluorescence imaging combined with antibody-based profiling in multiple well-characterized human cell lines. This approach allows direct visualization of protein distribution within distinct subcellular compartments, thereby offering cell-type-specific and high-resolution localization evidence. These measures substantially reduce the likelihood of false annotations and provide users with a clear indication of annotation confidence.

### B.2 AN ALTERNATIVE ATTEMPT FOR PROTEIN STRUCTURE INPUTS

In our study, we used protein structural data predicted by AlphaFold2, the most accurate and widely adopted source of protein structural information, to provide structure inputs for our structure-

Table 9: Partial evaluation on protein structure input from AlphaFold2 and Boltz-2.

| Subcellular Locations | CDConv[t] (1,223 protein structures from AlphaFold2) | | | CDConv[t] (1,223 protein structures from Boltz-2) | | |
|---|---|---|---|---|---|---|
| | Precision | Recall | F1-Score | Precision | Recall | F1-Score |
| Nucleus | 0.798 | 0.709 | 0.751 | 0.788 | 0.565 | 0.658 |
| Nuclear Membrane | - | - | - | - | - | - |
| Nucleoli | 0.500 | 0.064 | 0.113 | 0.267 | 0.025 | 0.047 |
| Nucleoplasm | 0.794 | 0.649 | 0.714 | 0.791 | 0.448 | 0.572 |
| Cytoplasm | 0.701 | 0.577 | 0.633 | 0.634 | 0.673 | 0.653 |
| Cytosol | 0.645 | 0.429 | 0.515 | 0.541 | 0.622 | 0.579 |
| Cytoskeleton | 0.548 | 0.085 | 0.147 | 0.467 | 0.035 | 0.065 |
| Centrosome | - | - | - | - | - | - |
| Mitochondria | 0.694 | 0.200 | 0.311 | 0.358 | 0.152 | 0.213 |
| Endoplasmic Reticulum | 0.630 | 0.168 | 0.266 | 0.500 | 0.149 | 0.229 |
| Golgi Apparatus | 0.500 | 0.023 | 0.044 | 0.333 | 0.008 | 0.015 |
| Cell Membrane | 0.823 | 0.382 | 0.522 | 0.826 | 0.269 | 0.406 |
| Endosome | - | - | - | - | - | - |
| Lipid Droplet | - | - | - | - | - | - |
| Lysosome/Vacuole | - | - | - | - | - | - |
| Peroxisome | - | - | - | - | - | - |
| Vesicle | 0.800 | 0.019 | 0.037 | 0.667 | 0.010 | 0.019 |
| Primary Cilium | - | - | - | - | - | - |
| Secreted Proteins | 0.724 | 0.375 | 0.494 | 0.655 | 0.339 | 0.447 |
| Sperm | - | - | - | - | - | - |
| **Micro Avg** | 0.745 | 0.407 | 0.527 | 0.668 | 0.371 | 0.477 |
| **Macro Avg** | 0.408 | 0.184 | 0.227 | 0.341 | 0.165 | 0.195 |

[t]The original MLP is replaced by Transformer layers. "–" indicates that no prediction is made for that location.

based models. We acknowledge that Boltz is an efficient implementation of the still-unreleased AlphaFold3, and therefore seek to examine the subcellular localization performance when using Boltz-predicted structures as input (Wohlwend et al., 2025; Passaro et al., 2025).

To the best of our knowledge, Boltz has not publicly released a complete set of inference results of human protein structures, compared with the AlphaFold2 dataset whose complete inference results can be downloaded publicly. Therefore, we locate a partial dataset of Boltz inference results, which includes structures of protein complexes predicted by Boltz-2 (Ille et al., 2025). From this work, we extracted the subset overlapping with our CAPSUL benchmark, yielding 1,223 proteins. For these 1,223 proteins, we compared different structure inputs (*i.e.*, structures predicted by AlphaFold2 and by Boltz-2) on our previously trained structure-based model CDConv for inference. We report the results in Table 9.

Across the overall metric and the majority of subcellular locations, **AlphaFold2-based structural inputs outperform Boltz-based inputs**. This observation further supports the rationale behind our use of AlphaFold2-predicted structures in constructing CAPSUL. As the most accurate and widely adopted source of protein structural information, AlphaFold2 provides high-quality structural inputs that lead to strong downstream performance in subcellular localization.

## C  EXPERIMENT DETAILS

### C.1  IMPLEMENTATION DETAILS

The experiments were performed utilizing NVIDIA RTX 3090, A40 and A100 GPUs. We employ an early stopping strategy to mitigate overfitting with a tolerance of 5 epochs. Hyperparameters such as learning rate, number of epochs, and batch size are explored separately for each model type, considering their distinct architectures.

### C.2  FURTHER DESCRIPTION OF STRUCTURE-BASED MODELS AND THEIR EXTENSIONS

### C.2.1  OVERVIEW OF GRAPH CONSTRUCTION

In the graphs constructed by our structure-based models, each node represents an amino acid, and edges encode the relationships between amino acids. Specifically, the edges are constructed as follows:

**Edge Criteria.** There are two types of adjacency to form the edges in the graphs. **Sequential adjacency** refers to the proximity of amino acids along the one-dimensional primary sequence of a protein (*e.g.,* if the sequential adjacency range is set to 3, then the amino acids from position [x-3, x+3] are considered sequential neighbors of the x-th residue). On the other hand, **spatial adjacency** captures the proximity of amino acids in the three-dimensional space of the protein (*e.g.,* if the spatial adjacency radius is set to 8 Å, all amino acids located within an 8 Åsphere centered at a given residue are considered its spatial neighbors). These adjacency relationships define the edges in the constructed protein graph.

**Edge Features.** For the GCN-based baselines CDConv and GearNet-Edge, we adopted their innovative edge feature implementation methods originally proposed in their respective frameworks, which can be found in the corresponding publications (Fan et al., 2022; Zhang et al., 2022). These methods incorporate and encode both relative orientation and Euclidean distance. For our extended models, Graph Transformer, Graph Mamba, and Graph Diffusion, the edge features are derived by processing the aforementioned different edge criteria through an embedding layer.

### C.2.2 EXPLANATION OF GRAPH ENCODER

Within the structure-based baseline models, the graph encoders vary in their approaches to processing the input feature vectors: A **GCN** updates node representations via neighborhood aggregation, *i.e.,* $\boldsymbol{m}_i^{(0)} = \boldsymbol{x}_i$, $\boldsymbol{m}_i^{(l+1)} = \sigma\left(\sum_{j \in \mathcal{N}(i)} \boldsymbol{W}^{(l)} \boldsymbol{m}_j^{(l)} + \boldsymbol{b}^{(l)}\right)$. $\boldsymbol{m}_i^{(l)}$ is the representation of node $i$ at layer $l$ (the first layer is initialized with node embeddings $\boldsymbol{x}_i$ various in different baselines), $\mathcal{N}(i)$ denotes the neighbors of node $i$, $W^{(l)}$ and $\boldsymbol{b}^{(l)}$ are trainable weights and bias, and $\sigma$ is a non-linear activation function (*e.g.*, ReLU). After $L$ layers of graph convolution, we obtain the final node representations $\{\boldsymbol{m}_i^{(L)}\}_{i=1}^n$. To enhance interpretability and capture global interactions among residues, we replace the traditional average pooling with a Transformer encoder $\mathcal{T}(\cdot)$ to obtain the residue representation, *i.e.,* $\boldsymbol{h} = (\boldsymbol{h}_1, \ldots, \boldsymbol{h}_n) = \mathcal{T}\left(\{\boldsymbol{m}_i^{(L)}\}_{i=1}^n\right)$, where $\boldsymbol{h} \in \mathbb{R}^{n \times d}$.

Similarly, **Graph Transformer** and **Graph Mamba** substitute the convolution-based encoder with their respective architectures, while adhering to the same overall procedure to obtain the global protein representation.

Given a protein structure represented as a graph, we introduce a **diffusion**-based refinement process in which node coordinates or geometric features are gradually perturbed with Gaussian noise and then denoised through a learned reverse process. The diffusion module serves as an auxiliary representation-learning stage designed to enhance geometric feature extraction prior to the downstream subcellular localization prediction. This allows the network to capture multi-scale spatial dependencies while remaining robust to structural noise.

### C.2.3 EXPLANATION OF CONTRASTIVE LEARNING AND FUSION MODELS

**CDConv with Contrastive Loss.** For each of the 20 subcellular compartments, we construct positive pairs (*i.e.,* pairing protein samples that localize to the same compartment), and positive–negative pairs (*i.e.,* pairing one protein that localizes to the compartment with another that does not). On top of the original loss function, we incorporate a contrastive loss to encourage higher embedding similarity for positive pairs while enforcing lower similarity for positive–negative pairs.

We provide a formal mathematical description of how the contrastive loss is incorporated. Let $\bar{\mathbf{h}}$ denote the protein-level representations from model (*i.e.,* average embedding obtained before MLP classifier). For each of the 20 independent classification tasks, we compute the cosine similarity between representations of positive–positive pairs and positive–negative pairs, and construct the contrastive loss accordingly. For class $c \in \{1, \ldots, 20\}$, let

- $\mathcal{P}_c = \{i \mid y_{i,c} = 1\}$ denote the set of positive samples,
- $\mathcal{N}_c = \{i \mid y_{i,c} = 0\}$ denote the negative samples,
- $\bar{\mathbf{h}}_i$ denotes the protein-level embedding of the sample $i$.

The positive–positive similarity matrix is

$$S_{ij}^{(+,+)} = \cos(\mathbf{z}_i, \mathbf{z}_j), \quad i, j \in \mathcal{P}_c.$$

We encourage positive samples to be close to each other by minimizing

$$\mathcal{L}_c^{(+,+)} = 1 - \frac{1}{|\mathcal{P}_c|^2} \sum_{i,j \in \mathcal{P}_c} S_{ij}^{(+,+)}.$$

The positive–negative similarity matrix is

$$S_{ij}^{(+,-)} = \cos(\mathbf{z}_i, \mathbf{z}_j), \quad i \in \mathcal{P}_c, j \in \mathcal{N}_c.$$

We encourage positive and negative samples to be dissimilar by minimizing

$$\mathcal{L}_c^{(+,-)} = \frac{1}{|\mathcal{P}_c||\mathcal{N}_c|} \sum_{i \in \mathcal{P}_c} \sum_{j \in \mathcal{N}_c} S_{ij}^{(+,-)}.$$

Thus, the contrastive loss for class $c$ is

$$\mathcal{L}_c^{\text{contrast}} = \mathcal{L}_c^{(+,+)} + \mathcal{L}_c^{(+,-)}.$$

Finally, the overall contrastive loss averaged over all classes is

$$\mathcal{L}^{\text{contrast}} = \frac{1}{C} \sum_{c=1}^{C} \mathcal{L}_c^{\text{contrast}}.$$

**ESM-C+CDConv Fusion Model.** In the **early fusion** model, the structural representations produced by CDConv (without the additional Transformer architecture introduced by our paper) for each amino acid are added to the initial protein embedding of ESM-C. The combined representation is then passed through the pretrained ESM-C Transformer for interaction, followed by mean pooling to obtain a protein-level representation for downstream classification. The **late fusion** setting is similar, except that the structural representations from CDConv are added to the final sequence representation produced by ESM-C before mean pooling for the downstream classification task.

## C.3 HYPERPARAMETER SETTINGS

For all the experiments, we choose the best hyperparameters according to the best micro F1-score on the test set.

For the main experiment, the best hyperparameter setting for each model is as follows: 1) **ESM-2 (650M)**, the MLP hidden layers are set to (512,256), and learning rate to $1 \times 10^{-4}$. 2) **ESM-C (600M)**, the MLP hidden layers are set to (512,256) (to (512) when finetuning), and learning rate to $5 \times 10^{-4}$. 3) **FoldSeek**, the embedding dimensions are set to 256, transformer layers to 2, transformer heads to 4, and learning rate to $1 \times 10^{-4}$. 4) **CDConv**, the kernel channels are set to 24, feature channels to (256,512), geometric adjacency to 4Åand 8Å(gradually increase with the convolutional layers), sequential adjacency to 5, sequential kernel size to 5, transformer layers to 3, transformer heads to 2, and learning rate to $5 \times 10^{-4}$. 5) **GearNet-Edge**, the max sequence length is set to 3,000, spatial adjacency set to [5Å,10Å], KNN adjacency set to [5,10], sequential adjacency set to 2, convolution hidden dimensions to (512,512,512), transformer layers to 2, transformer heads to 2, and learning rate to $1 \times 10^{-5}$. 6) **Graph Transformer**, the transformer layers are set to 10, geometric adjacency to 10Å, sequential adjacency to 5, node dimensions set to 256, positional embedding dimension set to 8, and learning rate set to $5 \times 10^{-5}$. 7) **Graph Mamba**, the Mamba layers are set to 5, geometric adjacency to 10Å, sequential adjacency to 5, node dimensions set to 256, and learning rate set to $1 \times 10^{-4}$. 8) **Graph Diffusion**, the node dimensions are set to 64, geometric adjacency to 10Å, sequential adjacency to 5, timesteps set to 200, variance of Gaussian noise set to $1 \times 10^{-4}$ in the beginning and 0.02 in the end, weight for diffusion loss to 0.1 compared with classification loss, and learning rate to $5 \times 10^{-4}$. 9) **CDConv with Contrastive Loss**, the weight for contrastive loss is set to 0.1 compared with classification loss, and other settings are the same as the originial CDConv. 10) **ESM-C+CDConv Fusion Model**, the learning rate is set to $5 \times 10^{-4}$ for early fusion and to $1 \times 10^{-4}$ for late fusion, and other settings are the same as the originial ESM-C and CDConv.

For the reweighting strategy, we inherit the optimal hyperparameter settings for ESM-C (600M), CDConv, and GearNet-Edge mentioned above. The best reweighting scheme for each model is as

follows: 1) **ESM-C (600M)**, focal loss with $\alpha$ set to the weights of log-inverse frequency, and $\gamma$ set to 1.0. 2) **CDConv**, focal loss with $\alpha$ set to the weights of log-inverse frequency, and $\gamma$ set to 3.0. 3) **GearNet-Edge**, inverse frequency reweighting.

For the single-label classification strategy, we inherit the optimal hyperparameter settings for ESM-C (600M), CDConv, and GearNet-Edge mentioned above. To address class imbalance, we undersample the negative class to achieve a 1:3 positive-to-negative sample ratio for ESM-C (600M), and a 1:1 positive-to-negative sample ratio for CDConv.

## D    DETAILED BASELINE RESULTS

Detailed experimental results of main experiments, reweighting strategy, and single-label classification strategy are provided in Tables 10, 11, and 12, respectively. They include evaluation metrics of precision, recall, and F1-score.

## E    ABLATION STUDY

Although it has been recognized in the biological community that many patterns of subcellular localization cannot be fully captured by simple sequence information, we aim to investigate the potential benefits of incorporating protein structural information as input for prediction. Therefore, we conduct an ablation study on two representative structure-based baselines to quantify the positive impact of 3D information incorporated.

Specifically, to preserve the integrity of the model input, we performed preprocessing on the protein structural data. For each protein, we obtained the boundary values of its 3D coordinates and uniformly sampled the $C\alpha$ coordinates at random within these boundaries to generate new protein structures. The 1D sequence data were kept unchanged, while the randomly sampled structures were used as the 3D structural input. Using the same hyperparameter settings as in the main experiments, we conducted an ablation study, with the detailed results shown in Table 13. We observed a significant performance drop in this setting, which further demonstrates the decisive role of accurate 3D structural input in enabling correct model predictions.

## F    EXPLANATION AND ILLUSTRATIVE EXAMPLES OF EVIDENCE-LEVEL ANNOTATIONS

### F.1    EXPLANATION OF EVIDENCE-LEVEL ANNOTATIONS

The evidence-level annotations design was originally intended to allow researchers to flexibly select annotations based on their specific use cases. For instance, when the task is rigorous and requires high precision such as Nucleolar retention motifs discovery, selecting annotations with high confidence (*i.e.,* choosing the experimentally validated annotations only) is more appropriate. Conversely, for large-scale protein localization prediction, using lower-confidence but more abundant annotations (*i.e.,* choosing both the non-experimentally validated and non-experimentally validated annotations) enriches the training data and leads to better model performance.

### F.2    ILLUSTRATIVE EXAMPLES OF THREE STRATEGIES TOWARDS NON-EXPERIMENTALLY VALIDATED ANNOTATIONS

In our main experiments, all non-experimentally validated annotations were treated as positive samples to enhance the models' performance in high-throughput prediction settings. Here we present two illustrative examples of the flexible usages of evidence-level annotations: 1) **weighting labels** (*i.e.,* treating non-experimentally validated annotations as positive samples, but assigning a weight of 0.7 to them relative to experimental ones, which reduces the weight of non-experimental data in influencing the model) and 2) **filtering labels** (*i.e.,* treating non-experimentally validated annotations as negative samples, which restricts models learning to experimental data with high reliability). The results are compared with the original one in our paper in Table 14.

Table 10: Detailed performance of sequence-based and structure-based methods on CAPSUL.

| Subcellular Locations | DeepLoc 2.1 | | | ESM-2 650M | | | ESM-2 650M[f] | | |
|---|---|---|---|---|---|---|---|---|---|
| | Precision | Recall | F1-Score | Precision | Recall | F1-Score | Precision | Recall | F1-Score |
| Nucleus | 0.675 | 0.086 | 0.152 | - | - | - | 0.633 | 0.586 | 0.609 |
| Nuclear Membrane | / | / | / | - | - | - | - | - | - |
| Nucleoli | / | / | / | - | - | - | - | - | - |
| Nucleoplasm | / | / | / | - | - | - | 0.592 | 0.535 | 0.562 |
| Cytoplasm | 0.510 | 0.100 | 0.167 | - | - | - | 0.598 | 0.157 | 0.248 |
| Cytosol | / | / | / | - | - | - | - | - | - |
| Cytoskeleton | / | / | / | - | - | - | 0.200 | 0.003 | 0.006 |
| Centrosome | / | / | / | - | - | - | - | - | - |
| Mitochondria | 0.799 | 0.065 | 0.120 | - | - | - | 0.850 | 0.195 | 0.317 |
| Endoplasmic Reticulum | 0.581 | 0.067 | 0.121 | - | - | - | - | - | - |
| Golgi Apparatus | 0.594 | 0.032 | 0.061 | - | - | - | - | - | - |
| Cell Membrane | 0.740 | 0.078 | 0.142 | - | - | - | 0.722 | 0.451 | 0.555 |
| Endosome | / | / | / | - | - | - | - | - | - |
| Lipid Droplet | / | / | / | - | - | - | - | - | - |
| Lysosome/Vacuole | 0.198 | 0.084 | 0.118 | - | - | - | - | - | - |
| Peroxisome | 0.667 | 0.073 | 0.131 | - | - | - | - | - | - |
| Vesicle | / | / | / | - | - | - | - | - | - |
| Primary Cilium | / | / | / | - | - | - | - | - | - |
| Secreted Proteins | 0.773 | 0.109 | 0.191 | - | - | - | 0.742 | 0.686 | 0.713 |
| Sperm | / | / | / | - | - | - | - | - | - |
| **Micro Avg** | / | / | / | - | - | - | 0.647 | 0.264 | 0.375 |
| **Macro Avg** | / | / | / | - | - | - | 0.217 | 0.131 | 0.150 |

| Subcellular Locations | ESM-C 600M | | | ESM-C 600M[f] | | | ESM-C 600M[0] | | |
|---|---|---|---|---|---|---|---|---|---|
| | Precision | Recall | F1-Score | Precision | Recall | F1-Score | Precision | Recall | F1-Score |
| Nucleus | 0.694 | 0.609 | 0.649 | 0.708 | 0.597 | 0.648 | 0.626 | 0.498 | 0.555 |
| Nuclear Membrane | - | - | - | - | - | - | - | - | - |
| Nucleoli | 0.800 | 0.048 | 0.091 | 1.000 | 0.020 | 0.039 | 1.000 | 0.012 | 0.024 |
| Nucleoplasm | 0.679 | 0.573 | 0.621 | 0.686 | 0.570 | 0.623 | 0.620 | 0.418 | 0.500 |
| Cytoplasm | 0.611 | 0.477 | 0.536 | 0.614 | 0.499 | 0.551 | 0.507 | 0.385 | 0.438 |
| Cytosol | 0.541 | 0.307 | 0.392 | 0.567 | 0.286 | 0.380 | 0.456 | 0.104 | 0.169 |
| Cytoskeleton | 0.681 | 0.154 | 0.251 | 0.629 | 0.123 | 0.205 | 0.471 | 0.025 | 0.048 |
| Centrosome | 1.000 | 0.007 | 0.014 | - | - | - | - | - | - |
| Mitochondria | 0.865 | 0.416 | 0.562 | 0.903 | 0.389 | 0.544 | 0.667 | 0.053 | 0.099 |
| Endoplasmic Reticulum | 0.687 | 0.235 | 0.351 | 0.674 | 0.221 | 0.333 | 0.500 | 0.031 | 0.059 |
| Golgi Apparatus | 0.938 | 0.052 | 0.099 | 1.000 | 0.014 | 0.027 | - | - | - |
| Cell Membrane | 0.777 | 0.531 | 0.631 | 0.753 | 0.568 | 0.648 | 0.786 | 0.243 | 0.372 |
| Endosome | 1.000 | 0.009 | 0.018 | - | - | - | - | - | - |
| Lipid Droplet | - | - | - | - | - | - | - | - | - |
| Lysosome/Vacuole | - | - | - | - | - | - | - | - | - |
| Peroxisome | - | - | - | - | - | - | - | - | - |
| Vesicle | 1.000 | 0.005 | 0.009 | - | - | - | 1.000 | 0.002 | 0.005 |
| Primary Cilium | 0.682 | 0.093 | 0.164 | 0.556 | 0.062 | 0.112 | - | - | - |
| Secreted Proteins | 0.903 | 0.761 | 0.826 | 0.877 | 0.730 | 0.797 | 0.604 | 0.338 | 0.433 |
| Sperm | 0.500 | 0.028 | 0.052 | 0.667 | 0.037 | 0.070 | - | - | - |
| **Micro Avg** | 0.690 | 0.386 | 0.495 | 0.693 | 0.382 | 0.492 | 0.598 | 0.236 | 0.338 |
| **Macro Avg** | 0.618 | 0.215 | 0.263 | 0.482 | 0.206 | 0.249 | 0.362 | 0.106 | 0.135 |

| Subcellular Locations | FoldSeek | | | CDConv[t] | | | GearNet-Edge[t] | | |
|---|---|---|---|---|---|---|---|---|---|
| | Precision | Recall | F1-Score | Precision | Recall | F1-Score | Precision | Recall | F1-Score |
| Nucleus | 0.616 | 0.398 | 0.484 | 0.651 | 0.592 | 0.620 | 0.619 | 0.450 | 0.521 |
| Nuclear Membrane | - | - | - | - | - | - | - | - | - |
| Nucleoli | - | - | - | 0.583 | 0.084 | 0.147 | 0.531 | 0.068 | 0.121 |
| Nucleoplasm | 0.591 | 0.341 | 0.433 | 0.633 | 0.541 | 0.583 | 0.613 | 0.444 | 0.515 |
| Cytoplasm | 0.581 | 0.102 | 0.174 | 0.580 | 0.414 | 0.483 | 0.498 | 0.491 | 0.495 |
| Cytosol | 0.500 | 0.001 | 0.003 | 0.489 | 0.277 | 0.353 | 0.417 | 0.358 | 0.385 |
| Cytoskeleton | 0.480 | 0.038 | 0.070 | 0.649 | 0.075 | 0.135 | 0.296 | 0.186 | 0.228 |
| Centrosome | - | - | - | - | - | - | 0.228 | 0.088 | 0.127 |
| Mitochondria | - | - | - | 0.707 | 0.359 | 0.476 | 0.470 | 0.240 | 0.318 |
| Endoplasmic Reticulum | - | - | - | 0.441 | 0.218 | 0.292 | 0.475 | 0.197 | 0.279 |
| Golgi Apparatus | - | - | - | 0.733 | 0.038 | 0.073 | 0.211 | 0.014 | 0.026 |
| Cell Membrane | 0.626 | 0.237 | 0.343 | 0.721 | 0.461 | 0.562 | 0.708 | 0.457 | 0.556 |
| Endosome | - | - | - | - | - | - | 0.364 | 0.037 | 0.067 |
| Lipid Droplet | - | - | - | - | - | - | - | - | - |
| Lysosome/Vacuole | - | - | - | - | - | - | 0.429 | 0.040 | 0.073 |
| Peroxisome | - | - | - | - | - | - | - | - | - |
| Vesicle | - | - | - | 0.667 | 0.014 | 0.027 | 0.270 | 0.039 | 0.068 |
| Primary Cilium | - | - | - | - | - | - | 0.467 | 0.087 | 0.147 |
| Secreted Proteins | 0.600 | 0.225 | 0.328 | 0.795 | 0.741 | 0.767 | 0.722 | 0.655 | 0.687 |
| Sperm | - | - | - | - | - | - | 0.714 | 0.046 | 0.086 |
| **Micro Avg** | 0.605 | 0.156 | 0.248 | 0.632 | 0.352 | 0.452 | 0.546 | 0.337 | 0.417 |
| **Macro Avg** | 0.200 | 0.067 | 0.092 | 0.382 | 0.191 | 0.226 | 0.402 | 0.195 | 0.235 |

Table 10: (Continued) Detailed performance of sequence-based and structure-based methods on CAPSUL.

| Subcellular Locations | Graph Transformer | | | Graph Mamba | | | Graph Diffusion | | |
|---|---|---|---|---|---|---|---|---|---|
| | Precision | Recall | F1-Score | Precision | Recall | F1-Score | Precision | Recall | F1-Score |
| Nucleus | 0.664 | 0.543 | 0.597 | 0.562 | 0.556 | 0.559 | 0.617 | 0.631 | 0.624 |
|   Nuclear Membrane | - | - | - | 0.061 | 0.026 | 0.037 | - | - | - |
|   Nucleoli | 0.554 | 0.124 | 0.203 | 0.433 | 0.104 | 0.168 | 0.667 | 0.024 | 0.047 |
|   Nucleoplasm | 0.642 | 0.483 | 0.552 | 0.526 | 0.481 | 0.502 | 0.590 | 0.566 | 0.578 |
| Cytoplasm | 0.552 | 0.305 | 0.393 | 0.476 | 0.373 | 0.418 | 0.542 | 0.469 | 0.503 |
|   Cytosol | 0.457 | 0.170 | 0.248 | 0.421 | 0.431 | 0.426 | 0.440 | 0.214 | 0.288 |
|   Cytoskeleton | 0.538 | 0.022 | 0.042 | 0.249 | 0.296 | 0.270 | 0.383 | 0.057 | 0.099 |
| Centrosome | - | - | - | 0.128 | 0.313 | 0.181 | 1.000 | 0.007 | 0.014 |
| Mitochondria | 0.688 | 0.363 | 0.475 | 0.407 | 0.294 | 0.341 | 0.680 | 0.195 | 0.303 |
| Endoplasmic Reticulum | 0.552 | 0.111 | 0.184 | 0.529 | 0.031 | 0.059 | 0.556 | 0.087 | 0.150 |
| Golgi Apparatus | 0.857 | 0.021 | 0.041 | 0.182 | 0.188 | 0.185 | - | - | - |
| Cell Membrane | 0.718 | 0.442 | 0.547 | 0.417 | 0.766 | 0.540 | 0.737 | 0.373 | 0.496 |
| Endosome | - | - | - | 0.125 | 0.083 | 0.100 | - | - | - |
| Lipid Droplet | - | - | - | - | - | - | - | - | - |
| Lysosome/Vacuole | - | - | - | - | - | - | - | - | - |
| Peroxisome | - | - | - | - | - | - | - | - | - |
| Vesicle | 0.526 | 0.023 | 0.044 | 0.306 | 0.086 | 0.135 | 0.667 | 0.009 | 0.018 |
| Primary Cilium | 0.500 | 0.006 | 0.012 | 0.205 | 0.056 | 0.088 | - | - | - |
| Secreted Proteins | 0.767 | 0.652 | 0.705 | 0.426 | 0.802 | 0.557 | 0.738 | 0.539 | 0.623 |
| Sperm | 1.000 | 0.009 | 0.018 | 0.116 | 0.147 | 0.130 | - | - | - |
| **Micro Avg** | 0.637 | 0.302 | 0.410 | 0.414 | 0.408 | 0.411 | 0.596 | 0.329 | 0.424 |
| **Macro Avg** | 0.451 | 0.164 | 0.203 | 0.279 | 0.252 | 0.235 | 0.381 | 0.159 | 0.187 |

| Subcellular Locations | CDConv with Contrastive Loss | | | ESM-C+CDConv Early Fusion | | | ESM-C+CDConv Late Fusion | | |
|---|---|---|---|---|---|---|---|---|---|
| | Precision | Recall | F1-Score | Precision | Recall | F1-Score | Precision | Recall | F1-Score |
| Nucleus | 0.681 | 0.524 | 0.592 | 0.712 | 0.587 | 0.643 | 0.701 | 0.598 | 0.645 |
|   Nuclear Membrane | - | - | - | - | - | - | - | - | - |
|   Nucleoli | 0.556 | 0.080 | 0.140 | 0.739 | 0.068 | 0.125 | 0.210 | 0.120 | 0.153 |
|   Nucleoplasm | 0.646 | 0.487 | 0.556 | 0.704 | 0.591 | 0.643 | 0.676 | 0.567 | 0.617 |
| Cytoplasm | 0.590 | 0.404 | 0.480 | 0.682 | 0.341 | 0.455 | 0.572 | 0.469 | 0.515 |
|   Cytosol | 0.516 | 0.165 | 0.250 | 0.487 | 0.094 | 0.157 | 0.467 | 0.306 | 0.370 |
|   Cytoskeleton | 0.473 | 0.164 | 0.243 | 0.773 | 0.053 | 0.100 | 0.412 | 0.220 | 0.287 |
| Centrosome | - | - | - | - | - | - | 0.231 | 0.020 | 0.037 |
| Mitochondria | 0.754 | 0.340 | 0.468 | 0.872 | 0.416 | 0.563 | 0.827 | 0.420 | 0.557 |
| Endoplasmic Reticulum | 0.519 | 0.277 | 0.361 | 0.595 | 0.356 | 0.446 | - | - | - |
| Golgi Apparatus | 0.553 | 0.091 | 0.156 | 0.462 | 0.171 | 0.249 | - | - | - |
| Cell Membrane | 0.746 | 0.422 | 0.539 | 0.774 | 0.530 | 0.629 | 0.732 | 0.622 | 0.673 |
| Endosome | 0.250 | 0.019 | 0.034 | 1.000 | 0.028 | 0.054 | - | - | - |
| Lipid Droplet | - | - | - | - | - | - | - | - | - |
| Lysosome/Vacuole | - | - | - | 1.000 | 0.013 | 0.026 | - | - | - |
| Peroxisome | - | - | - | - | - | - | - | - | - |
| Vesicle | 0.750 | 0.014 | 0.027 | 0.432 | 0.036 | 0.067 | - | - | - |
| Primary Cilium | 0.500 | 0.019 | 0.036 | - | - | - | 0.255 | 0.075 | 0.115 |
| Secreted Proteins | 0.797 | 0.765 | 0.780 | 0.905 | 0.747 | 0.819 | 0.908 | 0.604 | 0.725 |
| Sperm | 1.000 | 0.009 | 0.018 | - | - | - | - | - | - |
| **Micro Avg** | 0.650 | 0.326 | 0.435 | 0.710 | 0.351 | 0.470 | 0.634 | 0.381 | 0.476 |
| **Macro Avg** | 0.467 | 0.189 | 0.234 | 0.507 | 0.202 | 0.249 | 0.300 | 0.201 | 0.235 |

[f]We finetune the pre-trained protein language model. [t]The original MLP is replaced by Transformer layers. [0]The parameters of ESM-C is initialized randomly. "/" indicates that DeepLoc 2.1 does not support prediction for that location, and therefore, average metrics are not considered in this case. "–" indicates that no prediction is made for that location.

Table 11: Detailed performance of selected baselines with reweighting scheme.

| Subcellular Locations | ESM-C 600M | | | CDConv† | | | GearNet-Edge† | | |
|---|---|---|---|---|---|---|---|---|---|
| | Precision | Recall | F1-Score | Precision | Recall | F1-Score | Precision | Recall | F1-Score |
| Nucleus | 0.698 | 0.575 | 0.630 | 0.481 | 0.892 | 0.625 | 0.484 | 0.856 | 0.618 |
| Nuclear Membrane | - | - | - | 0.033 | 0.566 | 0.062 | 0.046 | 0.079 | 0.058 |
| Nucleoli | - | - | - | 0.105 | 0.916 | 0.188 | 0.153 | 0.418 | 0.224 |
| Nucleoplasm | 0.679 | 0.500 | 0.576 | 0.469 | 0.859 | 0.607 | 0.436 | 0.841 | 0.574 |
| Cytoplasm | 0.568 | 0.446 | 0.500 | 0.450 | 0.823 | 0.582 | 0.441 | 0.711 | 0.544 |
| Cytosol | 0.513 | 0.076 | 0.133 | 0.353 | 0.829 | 0.495 | 0.366 | 0.714 | 0.484 |
| Cytoskeleton | 0.778 | 0.044 | 0.083 | 0.184 | 0.698 | 0.292 | 0.218 | 0.450 | 0.294 |
| Centrosome | - | - | - | 0.089 | 0.776 | 0.160 | 0.134 | 0.252 | 0.175 |
| Mitochondria | 0.846 | 0.336 | 0.481 | 0.191 | 0.672 | 0.297 | 0.247 | 0.427 | 0.313 |
| Endoplasmic Reticulum | - | - | - | 0.195 | 0.737 | 0.308 | 0.276 | 0.460 | 0.345 |
| Golgi Apparatus | - | - | - | 0.152 | 0.648 | 0.246 | 0.177 | 0.366 | 0.238 |
| Cell Membrane | 0.723 | 0.465 | 0.566 | 0.462 | 0.709 | 0.560 | 0.398 | 0.820 | 0.536 |
| Endosome | - | - | - | 0.067 | 0.407 | 0.114 | 0.177 | 0.130 | 0.150 |
| Lipid Droplet | 1.000 | 0.133 | 0.235 | 0.014 | 0.067 | 0.023 | 0.333 | 0.067 | 0.111 |
| Lysosome/Vacuole | - | - | - | 0.117 | 0.347 | 0.175 | 0.116 | 0.107 | 0.111 |
| Peroxisome | 1.000 | 0.105 | 0.190 | 0.040 | 0.421 | 0.072 | 0.111 | 0.105 | 0.108 |
| Vesicle | - | - | - | 0.198 | 0.532 | 0.288 | 0.206 | 0.445 | 0.281 |
| Primary Cilium | 0.667 | 0.012 | 0.024 | 0.096 | 0.640 | 0.167 | 0.123 | 0.311 | 0.176 |
| Secreted Proteins | 0.833 | 0.730 | 0.778 | 0.413 | 0.891 | 0.564 | 0.509 | 0.775 | 0.614 |
| Sperm | - | - | - | 0.066 | 0.679 | 0.120 | 0.109 | 0.147 | 0.125 |
| **Micro Avg** | 0.679 | 0.313 | 0.429 | 0.253 | 0.772 | 0.381 | 0.348 | 0.650 | 0.453 |
| **Macro Avg** | 0.415 | 0.171 | 0.210 | 0.209 | 0.655 | 0.197 | 0.253 | 0.424 | 0.304 |

†The original MLP is replaced by Transformer layers. "–" indicates that no prediction is made for that location.

Table 12: Detailed performance of selected baselines with single-label classification strategy.

| Subcellular Locations | ESM-C 600M | | | CDConv† | | | GearNet-Edge† | | |
|---|---|---|---|---|---|---|---|---|---|
| | Precision | Recall | F1-Score | Precision | Recall | F1-Score | Precision | Recall | F1-Score |
| Nuclear Membrane | - | - | - | 0.027 | 0.711 | 0.052 | 0.026 | 0.118 | 0.042 |
| Nucleoli | 0.251 | 0.285 | 0.267 | 0.082 | 0.992 | 0.151 | 0.151 | 0.470 | 0.228 |
| Centrosome | 0.124 | 0.361 | 0.184 | 0.051 | 0.333 | 0.089 | 0.099 | 0.531 | 0.167 |
| Golgi Apparatus | 0.293 | 0.268 | 0.280 | 0.080 | 0.199 | 0.114 | 0.161 | 0.303 | 0.210 |
| Endosome | 0.111 | 0.333 | 0.167 | 0.029 | 0.176 | 0.049 | 0.082 | 0.278 | 0.126 |
| Lipid Droplet | 0.011 | 0.200 | 0.021 | - | - | - | 0.032 | 0.133 | 0.051 |
| Lysosome/Vacuole | 0.075 | 0.253 | 0.115 | - | - | - | 0.097 | 0.493 | 0.162 |
| Peroxisome | 0.029 | 0.526 | 0.054 | - | - | - | 0.013 | 0.158 | 0.023 |
| Vesicle | 0.270 | 0.039 | 0.068 | 0.141 | 0.625 | 0.230 | 0.207 | 0.380 | 0.268 |
| Primary Cilium | 0.175 | 0.460 | 0.253 | 0.055 | 0.379 | 0.097 | 0.104 | 0.472 | 0.171 |
| Sperm | 0.121 | 0.229 | 0.159 | 0.045 | 0.138 | 0.068 | 0.077 | 0.239 | 0.117 |

†The original MLP is replaced by Transformer layers. "–" indicates that no prediction is made for that location.

Table 13: Detailed performance comparison of CDConv and GearNet-Edge under random sampling of Cα coordinates.

| Subcellular Locations | CDConv† (ablation) | | | CDConv† | | | GearNet-Edge† (ablation) | | | GearNet-Edge† | | |
|---|---|---|---|---|---|---|---|---|---|---|---|---|
| | Precision | Recall | F1-Score | Precision | Recall | F1-Score | Precision | Recall | F1-Score | Precision | Recall | F1-Score |
| Nucleus | 0.595 | 0.512 | 0.550 | 0.651 | 0.592 | 0.620 | 0.515 | 0.459 | 0.485 | 0.619 | 0.450 | 0.521 |
| Nuclear Membrane | - | - | - | - | - | - | - | - | - | - | - | - |
| Nucleoli | 0.417 | 0.020 | 0.038 | 0.583 | 0.084 | 0.147 | 0.268 | 0.076 | 0.119 | 0.531 | 0.068 | 0.121 |
| Nucleoplasm | 0.578 | 0.419 | 0.486 | 0.633 | 0.541 | 0.583 | 0.479 | 0.428 | 0.452 | 0.613 | 0.444 | 0.515 |
| Cytoplasm | 0.535 | 0.214 | 0.306 | 0.580 | 0.414 | 0.483 | 0.432 | 0.414 | 0.422 | 0.498 | 0.491 | 0.495 |
| Cytosol | 0.478 | 0.069 | 0.120 | 0.489 | 0.277 | 0.353 | 0.394 | 0.279 | 0.327 | 0.417 | 0.358 | 0.385 |
| Cytoskeleton | - | - | - | 0.649 | 0.075 | 0.135 | 0.252 | 0.119 | 0.162 | 0.296 | 0.186 | 0.228 |
| Centrosome | - | - | - | - | - | - | 0.184 | 0.061 | 0.092 | 0.228 | 0.088 | 0.127 |
| Mitochondria | 0.537 | 0.084 | 0.145 | 0.707 | 0.359 | 0.476 | 0.283 | 0.065 | 0.106 | 0.470 | 0.240 | 0.318 |
| Endoplasmic Reticulum | 0.625 | 0.017 | 0.034 | 0.441 | 0.218 | 0.292 | 0.321 | 0.062 | 0.104 | 0.475 | 0.197 | 0.279 |
| Golgi Apparatus | - | - | - | 0.733 | 0.038 | 0.073 | 0.132 | 0.017 | 0.031 | 0.211 | 0.014 | 0.026 |
| Cell Membrane | 0.621 | 0.425 | 0.505 | 0.721 | 0.461 | 0.562 | 0.595 | 0.413 | 0.487 | 0.708 | 0.457 | 0.556 |
| Endosome | - | - | - | - | - | - | - | - | - | 0.364 | 0.037 | 0.067 |
| Lipid Droplet | - | - | - | - | - | - | - | - | - | - | - | - |
| Lysosome/Vacuole | - | - | - | - | - | - | - | - | - | 0.429 | 0.040 | 0.073 |
| Peroxisome | - | - | - | - | - | - | - | - | - | - | - | - |
| Vesicle | - | - | - | 0.667 | 0.014 | 0.027 | 0.185 | 0.077 | 0.109 | 0.270 | 0.039 | 0.068 |
| Primary Cilium | - | - | - | - | - | - | 0.368 | 0.043 | 0.078 | 0.467 | 0.087 | 0.147 |
| Secreted Proteins | 0.703 | 0.218 | 0.333 | 0.795 | 0.741 | 0.767 | 0.515 | 0.232 | 0.320 | 0.722 | 0.655 | 0.687 |
| Sperm | - | - | - | - | - | - | 0.125 | 0.009 | 0.017 | 0.714 | 0.046 | 0.086 |
| **Micro Avg** | 0.586 | 0.229 | 0.329 | 0.632 | 0.352 | 0.452 | 0.450 | 0.283 | 0.348 | 0.546 | 0.337 | 0.417 |
| **Macro Avg** | 0.254 | 0.099 | 0.126 | 0.382 | 0.191 | 0.226 | 0.252 | 0.138 | 0.166 | 0.402 | 0.195 | 0.235 |

†The original MLP is replaced by Transformer layers. "–" indicates that no prediction is made for that location.

Table 14: Detailed performance of two illustrative examples of evidence-level annotations: weighting labels and filtering labels.

| Subcellular Locations | ESM-C 600M | | | ESM-C 600M (weighting) | | | ESM-C 600M (filtering) | | |
|---|---|---|---|---|---|---|---|---|---|
| | Precision | Recall | F1-Score | Precision | Recall | F1-Score | Precision | Recall | F1-Score |
| Nucleus | 0.694 | 0.609 | 0.649 | 0.717 | 0.585 | 0.645 | 0.703 | 0.575 | 0.633 |
| Nuclear Membrane | - | - | - | - | - | - | - | - | - |
| Nucleoli | 0.800 | 0.048 | 0.091 | 0.786 | 0.088 | 0.159 | 0.867 | 0.055 | 0.104 |
| Nucleoplasm | 0.679 | 0.573 | 0.621 | 0.692 | 0.558 | 0.618 | 0.686 | 0.551 | 0.611 |
| Cytoplasm | 0.611 | 0.477 | 0.536 | 0.619 | 0.453 | 0.523 | 0.572 | 0.400 | 0.470 |
| Cytosol | 0.541 | 0.307 | 0.392 | 0.534 | 0.256 | 0.346 | 0.590 | 0.239 | 0.340 |
| Cytoskeleton | 0.681 | 0.154 | 0.251 | 0.649 | 0.116 | 0.197 | 0.381 | 0.034 | 0.062 |
| Centrosome | 1.000 | 0.007 | 0.014 | 1.000 | 0.007 | 0.014 | - | - | - |
| Mitochondria | 0.865 | 0.416 | 0.562 | 0.907 | 0.374 | 0.530 | 0.798 | 0.373 | 0.508 |
| Endoplasmic Reticulum | 0.687 | 0.235 | 0.351 | 0.726 | 0.156 | 0.256 | 0.500 | 0.039 | 0.072 |
| Golgi Apparatus | 0.938 | 0.052 | 0.099 | 1.000 | 0.010 | 0.021 | - | - | - |
| Cell Membrane | 0.777 | 0.531 | 0.631 | 0.757 | 0.570 | 0.650 | 0.661 | 0.254 | 0.367 |
| Endosome | 1.000 | 0.009 | 0.018 | - | - | - | - | - | - |
| Lipid Droplet | - | - | - | - | - | - | - | - | - |
| Lysosome/Vacuole | - | - | - | - | - | - | - | - | - |
| Peroxisome | - | - | - | - | - | - | - | - | - |
| Vesicle | 1.000 | 0.005 | 0.009 | 1.000 | 0.005 | 0.009 | - | - | - |
| Primary Cilium | 0.682 | 0.093 | 0.164 | 0.538 | 0.043 | 0.080 | 1.000 | 0.008 | 0.016 |
| Secreted Proteins | 0.903 | 0.761 | 0.826 | 0.920 | 0.669 | 0.775 | - | - | - |
| Sperm | 0.500 | 0.028 | 0.052 | 0.800 | 0.037 | 0.070 | 0.500 | 0.010 | 0.019 |
| **Micro Avg** | 0.690 | 0.386 | 0.495 | 0.700 | 0.366 | 0.481 | 0.657 | 0.306 | 0.418 |
| **Macro Avg** | 0.618 | 0.215 | 0.263 | 0.582 | 0.196 | 0.245 | 0.363 | 0.127 | 0.160 |

| Subcellular Locations | CDConv[t] | | | CDConv[t] (weighting) | | | CDConv[t] (filtering) | | |
|---|---|---|---|---|---|---|---|---|---|
| | Precision | Recall | F1-Score | Precision | Recall | F1-Score | Precision | Recall | F1-Score |
| Nucleus | 0.651 | 0.592 | 0.620 | 0.674 | 0.539 | 0.599 | 0.644 | 0.530 | 0.581 |
| Nuclear Membrane | - | - | - | - | - | - | - | - | - |
| Nucleoli | 0.583 | 0.084 | 0.147 | 0.556 | 0.020 | 0.039 | 0.484 | 0.640 | 0.112 |
| Nucleoplasm | 0.633 | 0.541 | 0.583 | 0.657 | 0.497 | 0.566 | 0.652 | 0.424 | 0.514 |
| Cytoplasm | 0.580 | 0.414 | 0.483 | 0.557 | 0.469 | 0.509 | 0.528 | 0.445 | 0.483 |
| Cytosol | 0.489 | 0.277 | 0.353 | 0.470 | 0.293 | 0.361 | 0.468 | 0.337 | 0.392 |
| Cytoskeleton | 0.649 | 0.075 | 0.135 | 0.714 | 0.063 | 0.116 | 0.400 | 0.008 | 0.016 |
| Centrosome | - | - | - | - | - | - | - | - | - |
| Mitochondria | 0.707 | 0.359 | 0.476 | 0.762 | 0.355 | 0.484 | 0.588 | 0.363 | 0.449 |
| Endoplasmic Reticulum | 0.441 | 0.218 | 0.292 | 0.561 | 0.159 | 0.248 | 0.278 | 0.024 | 0.045 |
| Golgi Apparatus | 0.733 | 0.038 | 0.073 | 0.615 | 0.028 | 0.053 | - | - | - |
| Cell Membrane | 0.721 | 0.461 | 0.562 | 0.723 | 0.447 | 0.553 | 0.562 | 0.194 | 0.288 |
| Endosome | - | - | - | - | - | - | - | - | - |
| Lipid Droplet | - | - | - | - | - | - | - | - | - |
| Lysosome/Vacuole | - | - | - | - | - | - | - | - | - |
| Peroxisome | - | - | - | - | - | - | - | - | - |
| Vesicle | 0.667 | 0.014 | 0.027 | - | - | - | - | - | - |
| Primary Cilium | - | - | - | 0.333 | 0.006 | 0.012 | - | - | - |
| Secreted Proteins | 0.795 | 0.741 | 0.767 | 0.857 | 0.573 | 0.687 | 0.400 | 0.044 | 0.079 |
| Sperm | - | - | - | - | - | - | - | - | - |
| **Micro Avg** | 0.632 | 0.352 | 0.452 | 0.637 | 0.333 | 0.438 | 0.577 | 0.291 | 0.386 |
| **Macro Avg** | 0.382 | 0.191 | 0.226 | 0.374 | 0.173 | 0.250 | 0.250 | 0.122 | 0.148 |

| Subcellular Locations | GearNet-Edge[t] | | | GearNet-Edge[t] (weighting) | | | GearNet-Edge[t] (filtering) | | |
|---|---|---|---|---|---|---|---|---|---|
| | Precision | Recall | F1-Score | Precision | Recall | F1-Score | Precision | Recall | F1-Score |
| Nucleus | 0.619 | 0.450 | 0.521 | 0.622 | 0.337 | 0.437 | 0.607 | 0.478 | 0.535 |
| Nuclear Membrane | - | - | - | - | - | - | - | - | - |
| Nucleoli | 0.531 | 0.068 | 0.121 | 0.421 | 0.032 | 0.060 | 0.333 | 0.064 | 0.107 |
| Nucleoplasm | 0.613 | 0.444 | 0.515 | 0.618 | 0.326 | 0.427 | 0.576 | 0.453 | 0.507 |
| Cytoplasm | 0.498 | 0.491 | 0.495 | 0.473 | 0.466 | 0.469 | 0.452 | 0.479 | 0.465 |
| Cytosol | 0.417 | 0.358 | 0.385 | 0.424 | 0.339 | 0.377 | 0.394 | 0.403 | 0.398 |
| Cytoskeleton | 0.296 | 0.186 | 0.228 | 0.342 | 0.167 | 0.224 | 0.284 | 0.105 | 0.153 |
| Centrosome | 0.228 | 0.088 | 0.127 | 0.213 | 0.068 | 0.103 | 0.200 | 0.054 | 0.085 |
| Mitochondria | 0.470 | 0.240 | 0.318 | 0.667 | 0.176 | 0.278 | 0.508 | 0.142 | 0.221 |
| Endoplasmic Reticulum | 0.475 | 0.197 | 0.279 | 0.435 | 0.128 | 0.198 | 0.268 | 0.073 | 0.115 |
| Golgi Apparatus | 0.211 | 0.014 | 0.026 | 0.211 | 0.014 | 0.026 | 0.250 | 0.004 | 0.008 |
| Cell Membrane | 0.708 | 0.457 | 0.556 | 0.629 | 0.392 | 0.483 | 0.463 | 0.170 | 0.248 |
| Endosome | 0.364 | 0.037 | 0.067 | 0.333 | 0.028 | 0.051 | 0.500 | 0.029 | 0.056 |
| Lipid Droplet | - | - | - | - | - | - | - | - | - |
| Lysosome/Vacuole | 0.429 | 0.040 | 0.073 | 0.125 | 0.013 | 0.024 | - | - | - |
| Peroxisome | - | - | - | - | - | - | - | - | - |
| Vesicle | 0.270 | 0.039 | 0.068 | 0.268 | 0.093 | 0.138 | 0.280 | 0.055 | 0.092 |
| Primary Cilium | 0.467 | 0.087 | 0.147 | 0.524 | 0.068 | 0.121 | 0.364 | 0.031 | 0.058 |
| Secreted Proteins | 0.722 | 0.655 | 0.687 | 0.826 | 0.519 | 0.637 | 0.273 | 0.066 | 0.106 |
| Sperm | 0.714 | 0.046 | 0.086 | 0.667 | 0.037 | 0.070 | 0.500 | 0.020 | 0.038 |
| **Micro Avg** | 0.546 | 0.337 | 0.417 | 0.529 | 0.282 | 0.368 | 0.485 | 0.300 | 0.371 |
| **Macro Avg** | 0.402 | 0.195 | 0.235 | 0.390 | 0.160 | 0.206 | 0.313 | 0.131 | 0.160 |

[t]The original MLP is replaced by Transformer layers. "–" indicates that no prediction is made for that location.

Table 15: Detailed performance of different weights for non-experimentally validated annotations on ESM-C.

| Subcellular Locations | ESM-C weighting 0.1 | | | ESM-C weighting 0.3 | | | ESM-C weighting 0.5 | | |
|---|---|---|---|---|---|---|---|---|---|
| | Precision | Recall | F1-Score | Precision | Recall | F1-Score | Precision | Recall | F1-Score |
| Nucleus | 0.733 | 0.549 | 0.628 | 0.73 | 0.553 | 0.629 | 0.728 | 0.564 | 0.636 |
|   Nuclear Membrane | - | - | - | - | - | - | - | - | - |
|   Nucleoli | 0.789 | 0.060 | 0.112 | 0.792 | 0.076 | 0.139 | 0.824 | 0.056 | 0.105 |
|   Nucleoplasm | 0.708 | 0.518 | 0.598 | 0.714 | 0.519 | 0.601 | 0.710 | 0.522 | 0.602 |
| Cytoplasm | 0.605 | 0.511 | 0.554 | 0.592 | 0.506 | 0.546 | 0.600 | 0.505 | 0.548 |
|   Cytosol | 0.530 | 0.350 | 0.422 | 0.518 | 0.336 | 0.408 | 0.511 | 0.382 | 0.437 |
|   Cytoskeleton | 0.636 | 0.110 | 0.188 | 0.587 | 0.116 | 0.194 | 0.618 | 0.107 | 0.182 |
| Centrosome | - | - | - | - | - | - | 1.000 | 0.007 | 0.014 |
| Mitochondria | 0.917 | 0.382 | 0.539 | 0.917 | 0.382 | 0.539 | 0.914 | 0.366 | 0.523 |
| Endoplasmic Reticulum | 0.750 | 0.114 | 0.198 | 0.698 | 0.128 | 0.216 | 0.696 | 0.166 | 0.268 |
| Golgi Apparatus | - | - | - | - | - | - | 1.000 | 0.003 | 0.007 |
| Cell Membrane | 0.771 | 0.367 | 0.497 | 0.768 | 0.367 | 0.496 | 0.791 | 0.442 | 0.567 |
| Endosome | - | - | - | - | - | - | - | - | - |
| Lipid Droplet | - | - | - | - | - | - | - | - | - |
| Lysosome/Vacuole | - | - | - | - | - | - | - | - | - |
| Peroxisome | - | - | - | - | - | - | - | - | - |
| Vesicle | - | - | - | - | - | - | 1.000 | 0.005 | 0.009 |
| Primary Cilium | 0.583 | 0.043 | 0.081 | 0.647 | 0.068 | 0.124 | 0.625 | 0.031 | 0.059 |
| Secreted Proteins | 0.933 | 0.474 | 0.629 | 0.936 | 0.447 | 0.605 | 0.958 | 0.471 | 0.632 |
| Sperm | 0.167 | 0.009 | 0.017 | 0.375 | 0.028 | 0.051 | 0.667 | 0.018 | 0.036 |
| **Micro Avg** | 0.687 | 0.338 | 0.453 | 0.682 | 0.338 | 0.452 | 0.685 | 0.353 | 0.466 |
| **Macro Avg** | 0.406 | 0.174 | 0.223 | 0.414 | 0.176 | 0.227 | 0.582 | 0.182 | 0.231 |

| Subcellular Locations | ESM-C weighting 0.7 | | | ESM-C weighting 0.9 | | |
|---|---|---|---|---|---|---|
| | Precision | Recall | F1-Score | Precision | Recall | F1-Score |
| Nucleus | 0.717 | 0.585 | 0.645 | 0.711 | 0.578 | 0.638 |
|   Nuclear Membrane | - | - | - | - | - | - |
|   Nucleoli | 0.786 | 0.088 | 0.159 | 0.833 | 0.080 | 0.147 |
|   Nucleoplasm | 0.692 | 0.558 | 0.618 | 0.699 | 0.548 | 0.614 |
| Cytoplasm | 0.619 | 0.453 | 0.523 | 0.596 | 0.517 | 0.554 |
|   Cytosol | 0.534 | 0.256 | 0.346 | 0.517 | 0.372 | 0.432 |
|   Cytoskeleton | 0.649 | 0.116 | 0.197 | 0.646 | 0.132 | 0.219 |
| Centrosome | 1.000 | 0.007 | 0.014 | 1.000 | 0.007 | 0.014 |
| Mitochondria | 0.907 | 0.374 | 0.530 | 0.899 | 0.374 | 0.528 |
| Endoplasmic Reticulum | 0.726 | 0.156 | 0.256 | 0.680 | 0.235 | 0.350 |
| Golgi Apparatus | 1.000 | 0.010 | 0.021 | 0.909 | 0.035 | 0.067 |
| Cell Membrane | 0.757 | 0.570 | 0.650 | 0.759 | 0.540 | 0.631 |
| Endosome | - | - | - | 0.667 | 0.019 | 0.036 |
| Lipid Droplet | - | - | - | - | - | - |
| Lysosome/Vacuole | - | - | - | - | - | - |
| Peroxisome | - | - | - | - | - | - |
| Vesicle | 1.000 | 0.005 | 0.009 | 1.000 | 0.007 | 0.014 |
| Primary Cilium | 0.538 | 0.043 | 0.080 | 0.571 | 0.050 | 0.091 |
| Secreted Proteins | 0.920 | 0.669 | 0.775 | 0.907 | 0.734 | 0.811 |
| Sperm | 0.800 | 0.037 | 0.070 | 0.750 | 0.028 | 0.053 |
| **Micro Avg** | 0.700 | 0.366 | 0.481 | 0.683 | 0.388 | 0.495 |
| **Macro Avg** | 0.582 | 0.196 | 0.245 | 0.607 | 0.213 | 0.260 |

"–" indicates that no prediction is made for that location.

As shown in the table, models that treat non-experimentally validated annotations as positive samples generally achieve the best overall performance. This may be because many non-validated annotations also originate from reliable sources (*e.g.,* the "ECO:0000303" code in the UniProt database indicates that the localization information is extracted from published literature); thus, they still hold relatively high credibility. This also demonstrates that for large-scale deep learning training, as our work does, including such annotations can increase sample diversity and improve data richness, and therefore, improve models' performance.

## F.3 ANALYSIS OF DIFFERENT WEIGHTS FOR NON-EXPERIMENTALLY VALIDATED ANNOTATIONS

To have a comprehensive analysis of the effect of evidence level, we further extend the weighting-label strategy by setting different weights to samples with non-experimentally validated annotations. We report the results in Table 15 and 16.

From the results, we have the following observation:

Table 16: Detailed performance of different weights for non-experimentally validated annotations on CDConv.

| Subcellular Locations | CDConv[t] weighting 0.1 | | | CDConv[t] weighting 0.3 | | | CDConv[t] weighting 0.5 | | |
|---|---|---|---|---|---|---|---|---|---|
| | Precision | Recall | F1-Score | Precision | Recall | F1-Score | Precision | Recall | F1-Score |
| Nucleus | 0.640 | 0.646 | 0.643 | 0.616 | 0.662 | 0.638 | 0.671 | 0.543 | 0.600 |
| Nuclear Membrane | - | - | - | - | - | - | - | - | - |
| Nucleoli | 0.652 | 0.060 | 0.110 | 0.471 | 0.096 | 0.160 | 0.667 | 0.016 | 0.031 |
| Nucleoplasm | 0.614 | 0.533 | 0.571 | 0.596 | 0.600 | 0.598 | 0.640 | 0.509 | 0.567 |
| Cytoplasm | 0.594 | 0.450 | 0.512 | 0.592 | 0.398 | 0.476 | 0.558 | 0.514 | 0.535 |
| Cytosol | 0.493 | 0.322 | 0.390 | 0.478 | 0.278 | 0.352 | 0.468 | 0.348 | 0.399 |
| Cytoskeleton | 0.750 | 0.009 | 0.019 | 0.818 | 0.028 | 0.055 | 0.619 | 0.041 | 0.077 |
| Centrosome | - | - | - | - | - | - | - | - | - |
| Mitochondria | 0.728 | 0.347 | 0.470 | 0.708 | 0.370 | 0.486 | 0.703 | 0.344 | 0.462 |
| Endoplasmic Reticulum | 0.559 | 0.131 | 0.213 | 0.475 | 0.131 | 0.206 | 0.540 | 0.163 | 0.250 |
| Golgi Apparatus | - | - | - | - | - | - | 0.750 | 0.010 | 0.021 |
| Cell Membrane | 0.762 | 0.333 | 0.463 | 0.732 | 0.413 | 0.528 | 0.753 | 0.445 | 0.560 |
| Endosome | - | - | - | - | - | - | - | - | - |
| Lipid Droplet | - | - | - | - | - | - | - | - | - |
| Lysosome/Vacuole | - | - | - | - | - | - | - | - | - |
| Peroxisome | - | - | - | - | - | - | - | - | - |
| Vesicle | - | - | - | - | - | - | - | - | - |
| Primary Cilium | 1.000 | 0.006 | 0.012 | 1.000 | 0.006 | 0.012 | - | - | - |
| Secreted Proteins | 0.823 | 0.618 | 0.706 | 0.835 | 0.672 | 0.745 | 0.868 | 0.539 | 0.665 |
| Sperm | - | - | - | - | - | - | - | - | - |
| **Micro Avg** | 0.631 | 0.340 | 0.442 | 0.617 | 0.354 | 0.450 | 0.629 | 0.343 | 0.444 |
| **Macro Avg** | 0.381 | 0.173 | 0.205 | 0.366 | 0.183 | 0.213 | 0.362 | 0.174 | 0.208 |

| Subcellular Locations | CDConv[t] weighting 0.7 | | | CDConv[t] weighting 0.9 | | |
|---|---|---|---|---|---|---|
| | Precision | Recall | F1-Score | Precision | Recall | F1-Score |
| Nucleus | 0.674 | 0.539 | 0.599 | 0.682 | 0.523 | 0.592 |
| Nuclear Membrane | - | - | - | - | - | - |
| Nucleoli | 0.556 | 0.020 | 0.039 | 0.750 | 0.012 | 0.024 |
| Nucleoplasm | 0.657 | 0.497 | 0.566 | 0.669 | 0.479 | 0.558 |
| Cytoplasm | 0.557 | 0.469 | 0.509 | 0.551 | 0.571 | 0.561 |
| Cytosol | 0.470 | 0.293 | 0.361 | 0.470 | 0.398 | 0.431 |
| Cytoskeleton | 0.714 | 0.063 | 0.116 | 0.480 | 0.075 | 0.130 |
| Centrosome | - | - | - | - | - | - |
| Mitochondria | 0.762 | 0.355 | 0.484 | 0.707 | 0.332 | 0.452 |
| Endoplasmic Reticulum | 0.561 | 0.159 | 0.248 | 0.481 | 0.173 | 0.254 |
| Golgi Apparatus | 0.615 | 0.028 | 0.053 | 0.588 | 0.035 | 0.066 |
| Cell Membrane | 0.723 | 0.447 | 0.553 | 0.725 | 0.493 | 0.587 |
| Endosome | - | - | - | - | - | - |
| Lipid Droplet | - | - | - | - | - | - |
| Lysosome/Vacuole | - | - | - | - | - | - |
| Peroxisome | - | - | - | - | - | - |
| Vesicle | - | - | - | - | - | - |
| Primary Cilium | 0.333 | 0.006 | 0.012 | 0.333 | 0.006 | 0.012 |
| Secreted Proteins | 0.857 | 0.573 | 0.687 | 0.845 | 0.597 | 0.700 |
| Sperm | - | - | - | - | - | - |
| **Micro Avg** | 0.637 | 0.333 | 0.438 | 0.625 | 0.359 | 0.456 |
| **Macro Avg** | 0.374 | 0.173 | 0.211 | 0.364 | 0.185 | 0.218 |

[t]The original MLP is replaced by Transformer layers. "–" indicates that no prediction is made for that location.

Table 17: Analysis of precision and recall on Nucleus on ESM-C.

| ESM-C weighting | 0.1 | 0.3 | 0.5 | 0.7 | 0.9 | Treat as Positive |
|---|---|---|---|---|---|---|
| Precision for Nucleus | 0.733 | 0.730 | 0.728 | 0.717 | 0.711 | 0.694 |
| Recall for Nucleus | 0.549 | 0.553 | 0.564 | 0.585 | 0.578 | 0.609 |

1) For certain subcellular compartments, baselines trained exclusively on experimentally validated annotations or assigned low positive weights to non-experimentally validated annotations generally **achieve higher precision**, as shown in Table 17. Actually, precision and recall often represent a trade-off in modeling strategies. That is, adopting a more conservative prediction strategy typically increases precision but reduces the number of correctly recalled samples, and vice versa. Therefore, selecting high-confidence evidence levels can be seen as **a method of enforcing a more conservative prediction approach, helping to reduce the likelihood of false-positive predictions**. This highlights the novelty of evidence-level annotations: **using experimentally validated data is considered a strategy to ensure high precision and reliability.**

2) However, for overall results and most subcellular compartments, the results among the three strategies **show no significant differences**. This indirectly supports the notion we mentioned above that even **non-validated annotations in CAPSUL still possess relatively high reliability**. This

Table 18: Detailed performance of hierarchical classifiers on CDConv.

| | | | CDConv[t] Hierarchical Classifiers | | | |
|---|---|---|---|---|---|---|
| Subcellular Locations | Precision | Recall | F1-Score | Subcellular Locations | Precision | Recall | F1-Score |
| **Nucleus's classifier** | | | | **Cytoplasm's classifier** | | | |
| Nuclear Membrane | - | - | - | Cytosol | 0.260 | 0.980 | 0.411 |
| Nucleoli | - | - | - | Cytoskeleton | 0.331 | 0.126 | 0.182 |
| Nucleoplasm | 0.339 | 1.000 | 0.507 | | | | |

[t]The original MLP is replaced by Transformer layers. "–" indicates that no prediction is made for that location.

Table 19: Analysis of performance of three pLDDT groups on CDConv.

| Subcellular Locations | High pLDDT Group | | | Medium pLDDT Group | | | Low pLDDT Group | | |
|---|---|---|---|---|---|---|---|---|---|
| | Precision | Recall | F1-Score | Precision | Recall | F1-Score | Precision | Recall | F1-Score |
| Nucleus | 0.561 | 0.393 | 0.462 | 0.636 | 0.570 | 0.601 | 0.700 | 0.742 | 0.720 |
| Nuclear Membrane | - | - | - | - | - | - | - | - | - |
| Nucleoli | 0.667 | 0.082 | 0.146 | 0.632 | 0.136 | 0.224 | 0.375 | 0.034 | 0.062 |
| Nucleoplasm | 0.512 | 0.310 | 0.387 | 0.575 | 0.497 | 0.533 | 0.713 | 0.717 | 0.715 |
| Cytoplasm | 0.601 | 0.513 | 0.553 | 0.578 | 0.422 | 0.488 | 0.546 | 0.298 | 0.385 |
| Cytosol | 0.520 | 0.451 | 0.483 | 0.467 | 0.241 | 0.318 | 0.414 | 0.112 | 0.177 |
| Cytoskeleton | 0.833 | 0.060 | 0.112 | 0.577 | 0.128 | 0.210 | 0.800 | 0.034 | 0.065 |
| Centrosome | - | - | - | - | - | - | - | - | - |
| Mitochondria | 0.789 | 0.475 | 0.593 | 0.644 | 0.322 | 0.430 | 0.529 | 0.167 | 0.254 |
| Endoplasmic Reticulum | 0.506 | 0.344 | 0.409 | 0.410 | 0.152 | 0.222 | 0.176 | 0.054 | 0.082 |
| Golgi Apparatus | 0.857 | 0.057 | 0.107 | 0.714 | 0.048 | 0.089 | - | - | - |
| Cell Membrane | 0.806 | 0.463 | 0.588 | 0.743 | 0.581 | 0.653 | 0.530 | 0.272 | 0.360 |
| Endosome | - | - | - | - | - | - | - | - | - |
| Lipid Droplet | - | - | - | - | - | - | - | - | - |
| Lysosome/Vacuole | - | - | - | - | - | - | - | - | - |
| Peroxisome | - | - | - | 0.333 | 0.006 | 0.012 | - | - | - |
| Vesicle | 0.833 | 0.035 | 0.068 | - | - | - | - | - | - |
| Primary Cilium | - | - | - | 0.856 | 0.720 | 0.782 | - | - | - |
| Secreted Proteins | 0.746 | 0.775 | 0.760 | - | - | - | 0.816 | 0.702 | 0.755 |
| Sperm | - | - | - | | | | - | - | - |
| **Micro Avg** | 0.617 | 0.345 | 0.443 | 0.628 | 0.345 | 0.446 | 0.651 | 0.367 | 0.469 |
| **Macro Avg** | 0.412 | 0.198 | 0.233 | 0.538 | 0.191 | 0.228 | 0.280 | 0.157 | 0.179 |

"–" indicates that no prediction is made for that location.

ensures that, when CAPSUL is used for downstream tasks, the inclusion of non-experimentally validated annotations does not introduce substantial bias.

# G  ATTEMPT TOWARD HIERARCHICAL CLASSIFIERS FOR NESTED CATEGORIES

In order to make precise predictions of nested subcategories, we attempt to train a hierarchical classifier for the two nested categories, Nucleus and Cytoplasm. These classifiers were built upon CDConv, using positive samples of the corresponding parent categories as the training set. We report the experimental results in Table 18.

We have the following observations:

1) For the three subcategories within the nucleus, the hierarchical classifier actually leads to decreased performance compared with the original CDConv. We attribute this to the reduced number of training samples when training this hierarchical classifier, which, combined with the already severe class imbalance in these subcategories, likely exacerbated the issue. Considering the results discussed in Section 4.3 of the main text, we find that for such highly imbalanced subcellular localization tasks, directly applying a single-label classification strategy, while adjusting the corresponding reweighting and classification thresholds, yields more significant improvements.

2) For the two subcategories within the cytoplasm, classification performance greatly improves, demonstrating that the hierarchical classification strategy can be beneficial for certain subcellular compartments. This result provides an additional strategy, hierarchical classifiers, to address the minority class issue apart from what we have discussed in Section 4.3 of the main text.

## H  ANALYSIS OF THE pLDDT SCORES OF PROTEIN STRUCTURE INPUT

In AlphaFold-predicted structures, regions with low pLDDT often indicate intrinsically disordered segments, which naturally lack stable tertiary structure and are therefore harder for any predictor to model with high confidence. We conduct an analysis on the representative structure-based model, CDConv, examining **the relationship between residue-level pLDDT and model performance** across the test set.

Specifically, we divide all 3,028 proteins in the test set into three groups of equal size based on their protein-level mean pLDDT values. The highest and lowest mean pLDDT values within each group are [83.56, 99.39], [70.40, 83.54], and [28.11, 70.40]. We then compare the performance of these groups to determine whether substantial performance discrepancies exist, which would suggest that the model is sensitive to variations in structural confidence. We report the results of three groups in Table 19.

Upon further examination of specific subcellular categories, we find that:

1) For some subcellular locations (*e.g.,* Nucleus, Cytoplasm, Cytosol, and Cell Membrane), the performance differences among the three groups may appear larger. This is probably attributable to the uneven distribution of positive samples across the groups. For example, in the Nucleus category, the high-pLDDT group contains only 318 positive samples, whereas the low-pLDDT group contains 476 positive samples, which contributes to this noticeable discrepancy.

2) For other subcellular locations (*e.g.,* Cytoskeleton and Secreted Proteins), the results across the three groups do not exhibit pronounced differences. This observation is consistent with the trend shown by the overall evaluation metrics above, indicating that proteins with different average pLDDT levels have not greatly influenced the prediction of these subcellular localizations.

In summary, **the low-pLDDT group does not exhibit worse predictions systematically**. This indicates that the CAPSUL is robust to fluctuations in pLDDT and does not rely disproportionately on regions of high structural confidence. In other words, **CAPSUL provides stable and reliable structural inputs for downstream tasks even when proteins contain disordered or low-confidence regions, demonstrating the quality and suitability of our structural dataset for subcellular localization prediction.** However, results of particular subcellular compartments highlight the need for further exploration of developing more informative and robust structural representations to better capture the determinants of subcellular localization in future work.

## I  SAMPLE EFFICIENCY

A sample efficiency curve would explicitly reflect a possible strategy to choose the proper size of the training set, which strikes a balance between model performance and computational costs. We randomly select training subsets of varying sizes and evaluate their sample efficiency curve on CDConv in Figure 3.

We observed that the smaller dataset, compared with the original CAPSUL, exhibits noticeably poorer performance, reflected in both lower micro F1-scores and fewer categories for which any positive samples were successfully predicted. This indicates that, in order to achieve satisfactory overall performance as well as decent performance on minority classes, the current size of training samples is relatively appropriate.

## J  INTERPRETABILITY WITH ATTENTION SCORE

In Transformer architectures, the attention mechanism allows each token to compute a weighted representation of all other tokens in the sequence. Specifically, for a given token, a set of attention weights is derived via scaled dot-product operations between its query vector and the key vectors of all tokens, followed by a softmax normalization. These attention weights reflect how much information the token attends to from each of its peers. To assess the relative importance of each token within the sequence, we aggregated the attention it receives from all other tokens, *i.e.,* summing over the attention scores directed toward that token across the entire sequence. This provides a global measure of how influential a token is in shaping the contextual representations learned by

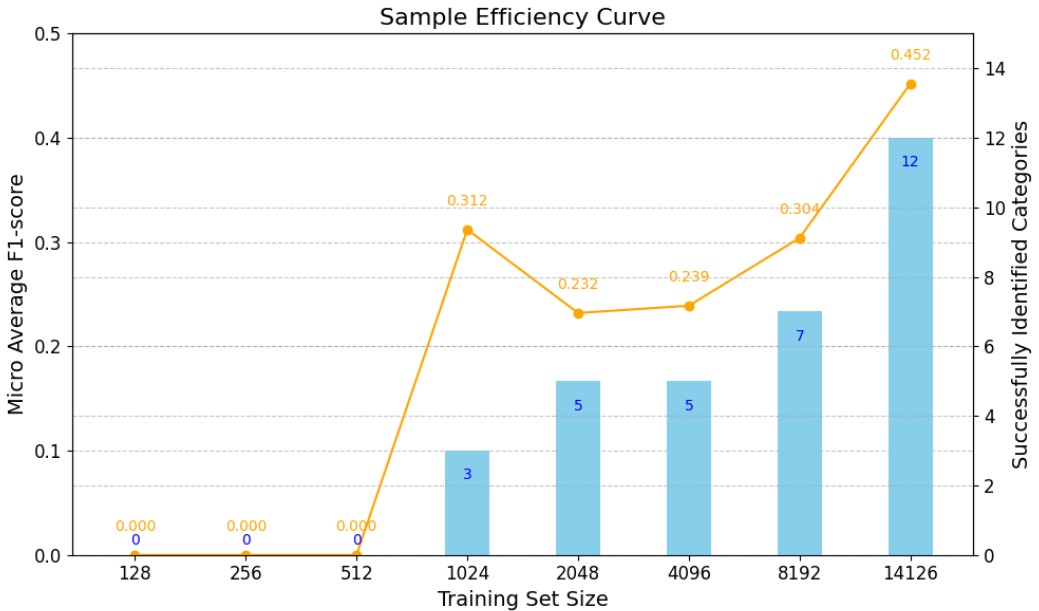

Figure 3: Sample efficiency curve on CDConv.

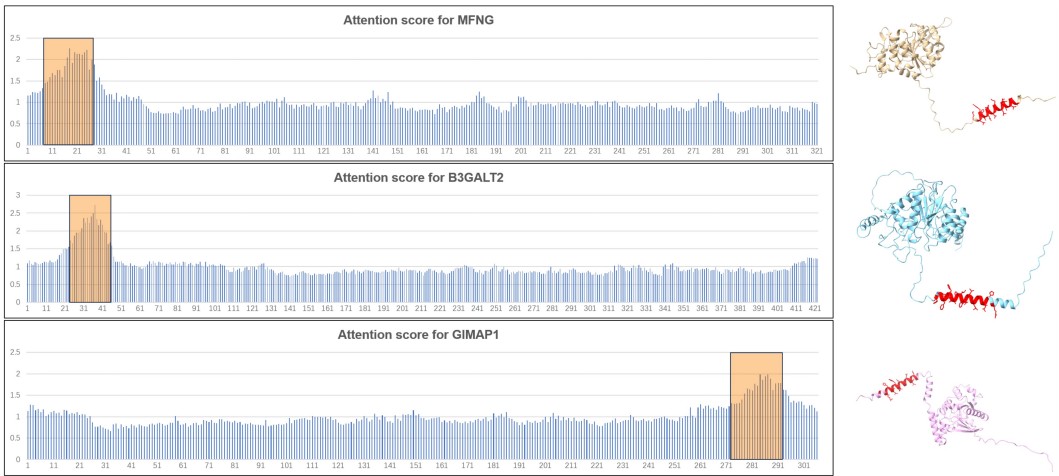

Figure 4: Visualization of full attention scores and structures of proteins MFNG, B3GALT2, and GIMAP1, where the residues of known pattern $\alpha$-helix are highlighted.

the model. We interpret this aggregated attention as a proxy for biological interpretability, where highly attended residues may correspond to structurally or functionally important positions within the protein.

In Section 4.3.3 of the main text, we introduce a CDConv model for predicting Golgi apparatus localization. By analyzing the attention score within the model's Transformer architecture, we identify a localization pattern associated with an $\alpha$-helix, which is consistent with existing biological findings. Here, we visualize the full attention score of the three example proteins discussed in the main text (*i.e.,* MFNG, B3GALT2, and GIMAP1), as shown in Figure 4. The residues of known localization patterns $\alpha$-helix are highlighted in orange for clear comparison. Notably, the 20 residues with the highest attention scores exhibit a 90% overlap with the ground truth, further highlighting the CDConv model's precision in identifying localization patterns.

**More interpretability results.** There are two more potential localization patterns identified on CAPSUL: 1) **"W-pair" for Golgi apparatus** (*i.e.,* two spatially adjacent Tryptophan residues), which aligns with existing studies suggesting that Tryptophan can influence Golgi targeting (Ashlin et al., 2021), and 2) **a flexible region at the N-terminus of the protein for Mitochondria**, which aligns with existing studies suggesting that this area can influence multiple compartments targeting (Sohn et al., 2009). However, these newly identified patterns still require further experimental validation. Nevertheless, these findings together demonstrate that both our curated dataset and the selected structure-based baseline model are capable of capturing meaningful subcellular localization signals, offering promising insights and directions for future research in cell biology.

## K    GENERALIZATION ABILITY

**The CAPSUL workflow is readily applicable to other species.** While our current study exclusively uses human protein data, we do not assert that CAPSUL's pipeline is intrinsically limited to human proteins. Our dataset and benchmark construction pipeline, including 1) the acquisition of one-dimensional and three-dimensional protein information, 2) the collection of localization annotations, and 3) the comparison of various baselines, is entirely species-agnostic. Once the CAPSUL pipeline becomes robust and well-received, there can be attempts to extend the same workflow to develop subcellular localization datasets and benchmarks across multiple species in parallel. This extension will not only enable broader biological investigations but also provide valuable resources for studying the evolutionary conservation and divergence of protein localization mechanisms across phylogenetic lineages. By comparing subcellular patterns across species, researchers can gain insights into the selective pressures shaping cellular organization, the emergence of organelle-specific functions, and the molecular adaptations that underpin evolutionary innovation.

**The CAPSUL workflow is applicable to context-dependent localization if sufficient related data is accessible.** With regard to the context-dependent dynamic localization, current protein databases lack extensive data on protein structures across diverse biological contexts and their corresponding variations in subcellular localization. If CAPSUL were augmented with dynamic data capturing protein structural changes (*e.g.,* conformational shifts induced by stressors or ligands) from updated protein databases in the future, the baselines can be retrained on CAPSUL to model context-dependent localization. This would enable the models to make precise predictions about subcellular localization under specific cellular conditions or in response to external environmental states, moving beyond static localization to capture functional biological dynamics.

**The CAPSUL is readily extensible to additional protein data and baseline methods in the future.** Our highly standardized pipeline ensures that any newly released protein data can be updated to CAPSUL. Also, CAPSUL embraces a broader range of other possible protein structural inputs (*e.g.,* which can be used in the construction of graph nodes and edges) to enrich both the local and global representations of proteins. Moreover, the results of more advanced protein representation methods can be included to perform this downstream task if their input data is supported by CAPSUL, which enables fair and informative comparison across various baseline methods.

## L    AVAILABILITY OF DATASET AND CODE

The complete dataset, including localization labels, extracted protein structures, *etc.* can be accessed at https://huggingface.co/datasets/getbetterhyccc/CAPSUL. Our implementation is publicly available at https://github.com/getbetter-hyccc/CAPSUL. For some baseline models, we adopt publicly released implementations, including Graph Transformer at https://github.com/pyg-team/pytorch_geometric/tree/master and Graph Mamba at https://github.com/alxndrTL/mamba.py.

## M    THE USE OF LARGE LANGUAGE MODELS

In this study, Large Language Models (LLMs) were employed solely for linguistic refinement, such as polishing the clarity, grammar, and fluency of the manuscript. Importantly, all conceptual advances, methodological innovations, experimental designs, and primary contributions presented in this work were independently conceived, developed, and validated by the authors. The role of LLMs

was thus limited to improving readability and ensuring the precision of academic writing, without influencing the scientific content or originality of the research.

