# OpenReview forum: "CAPSUL: A Comprehensive Human Protein Benchmark for Subcellular Localization"
_ICLR.cc/2026/Conference — ICLR 2026 Poster_

### Official Review · Reviewer_Xxcu · 2025-10-28

**Soundness:** 3
**Presentation:** 3
**Contribution:** 3
**Rating:** 4
**Confidence:** 3

**Summary:**

This paper proposes a subcellular localization dataset called CAPSUL that offers comprehensive 3D structural information with detailed subcellular localization annotated by domain experts. A variety of state-of-the-art methods are included to test the benchmark to demonstrate the importance of introducing structural information to identify the subcellular localization. A case study is also introduced to showcase the powerful interpretability of structure-based methods in cell biology.

**Strengths:**

1. The challenges and motivation behind the paper are well clarified and clearly demonstrated.

2. Extensive experiments and various tasks are conducted to prove the effectiveness of the proposed benchmarks. Both the quantitative and qualitative results are provided to showcase the importance and contribution of the proposed benchmarks in paving the way of cell biology.

3. The paper is well written and organized. Benchmarks and code are also available.

**Weaknesses:**

1. The proposed benchmark uses AlphaFold2 to extract the structural information for each protein. However, AlphaFold2 also has limitations on structural information. How will this affect the benchmarks and the following evaluations? When more accurate models emerge, will the benchmarks become outdated? And if the downstream tasks models are more accurate than AlphaFold2, such as AlphaFold3, will this benchmark downstream tasks evaluation be invalid?

2. The baselines in the paper have covered sequence and structure-based models. But there are more recent state-of-the-art models, for instance, OpenFold[1], Boltz[2], that should also be considered in the baselines.

[1] Gustav Ahdritz, Nazim Bouatta, Cristian Floristean, et al. Openfold: retraining alphafold2 yields new insights into its learning mechanisms and capacity for generalization. Nature Methods, 21: 1514–1524, 2024. doi: 10.1038/s41592-024-02272-z
[2] Jeremy Wohlwend, Gabriele Corso, Saro Passaro, Noah Getz, Mateo Reveiz, Ken Leidal, Wojtek Swiderski, Liam Atkinson, Tally Portnoi, Itamar Chinn, Jacob Silterra, Tommi Jaakkola, and Regina Barzilay. Boltz-1: Democratizing biomolecular interaction modeling. bioRxiv, 2024. doi: 10.1101/2024.11.19.624167.

**Questions:**

Please refer to the weaknesses.

---

> ### Author Response · Authors · 2025-11-22
> **Response to Reviewer Xxcu (Part 1/3)**
>
> Dear Reviewer `Xxcu`,
>
> Thanks for your inspiring comments! We sincerely appreciate your great effort to review and your positive recognition of our work. As you noted, we have conducted extensive experiments across a wide range of diverse baseline methods and performed in-depth qualitative and quantitative analyses based on the experimental results, ensuring excellent reproducibility.
>
> Following your comments and suggestions, we have provided detailed explanations and additional experimental results to address your concerns. Please feel free to leave further comments or questions if there is any misunderstanding, and we will reply soon.
>
>
>
> >Q1. Possible impacts of using structural information from AlphaFold.
>
> **Reply:** We fully understand your concerns regarding the use of AlphaFold data. We would like to justify its necessity and explain how we have minimized potential negative impacts from the following two perspectives:
>
> - **AlphaFold ensures both accessibility and consistency of large-scale protein structural information.** The remarkable success of recent deep learning methods in protein structure analysis can be largely attributed to the highly accurate data provided by AlphaFold, which 1) incorporated experimentally resolved structures as templates during its prediction process [1]. The median backbone accuracy of its predictions is 0.96 Å r.m.s.d. at 95% residue coverage, which often falls within the margin of error of experimental structures [1]. Currently, no alternative data source can provide training datasets of comparable scale and accuracy for comparative analysis. Only by utilizing large-scale data can we establish a solid foundation for downstream tasks such as protein classification and generation.
>
>   Furthermore, even for proteins with experimentally determined structures, current protein databases often store them in fragments. In contrast, AlphaFold generates full-length predicted structures, thereby preventing the oversight of critical regions and ensuring high consistency in input data.
>
>   In the following, we also provide a case study using structural data from Boltz, as you suggested, to further mitigate your concerns regarding our use of AlphaFold2-predicted structures.
>
> - **We have implemented specific strategies to mitigate potential adverse effects.** As mentioned in Section 3.1, we have taken several measures to minimize possible drawbacks of using AlphaFold data (*e.g.*, we filtered out long proteins predicted by AlphaFold2 with overlapping fragments to avoid incomplete or inconsistent inputs). Our benchmark and evaluation framework is designed to establish a unified assessment system, determining which types of methods can achieve high accuracy and biologically meaningful interpretability.
>
> [1] Jumper J, Evans R, Pritzel A, et al. Highly accurate protein structure prediction with AlphaFold[J]. nature, 2021, 596(7873): 583-589.
>
>
>
> > Q2. The consistent validity of CAPSUL with updated input data and baselines.
>
> **Reply:** We totally agree that the timeliness of our benchmark is indeed crucial. We would like to alleviate your concerns from two perspectives:
>
> - **The dataset construction and baseline evaluation pipeline of CAPSUL are highly reproducible.** Our dataset and benchmark pipeline, including 1) the acquisition of one-dimensional (*e.g.*, amino acid sequences) and three-dimensional protein information (*e.g.*, atoms coordinates) from updated protein structure databases such as AlphaFold2, 2) the collection of localization annotations from open and updated protein databases such as UniProt, and 3) the comparison of various baselines with the input supported by CAPSUL, is entirely reproducible. Such a highly standardized pipeline ensures that any newly released protein data (*e.g.*, once AlphaFold3 becomes fully open-sourced, as you mentioned) or future baseline models can be incorporated in a timely manner and compared fairly against the current benchmark results.
> - **We also wish to emphasize our strong commitment to continuously maintaining and improving CAPSUL.** We have observed a growing number of biological studies contributing new data to the subcellular localization task, which go beyond traditional databases. For example, Hein *et al*. have updated experimentally validated localization data for additional proteins [2]. In future work, we plan to expand the dataset by incorporating diverse sources, including high-throughput proteomics sequencing and low-throughput organelle purification experiments, thereby enhancing its comprehensiveness, reliability, and biological relevance.
>
> [2] Hein M Y, Peng D, Todorova V, et al. Global organelle profiling reveals subcellular localization and remodeling at proteome scale[J]. Cell, 2025, 188(4): 1137-1155. e20.

---

> ### Author Response · Authors · 2025-11-22
> **Response to Reviewer Xxcu (Part 2/3)**
>
> > Q3. Additional baselines of OpenFold and Boltz.
>
> Thank you very much for bringing these two papers to our attention! Firstly, we would like to clarify that in our study, AlphaFold2 is used solely as the source of protein structural data, which are then fed into all baseline models evaluated in our paper (*e.g.*, pretrained protein language models, GCN-based models, and Transformer-based models) for the downstream subcellular localization classification task.
>
> After carefully reading the two works you recommended, we found that they both propose **alternatives for generating protein structure inputs**:
>
> - **OpenFold** provides a fully trainable and efficient re-implementation of AlphaFold2 due to the absence of training code and data in the original AlphaFold2 release. Although OpenFold introduces new training procedures and achieves faster inference, its core architecture is still based on AlphaFold2.
>
> - **Boltz** is a new open-source model built upon principles of AlphaFold3, providing new training strategies and data, and achieving performance comparable to AlphaFold3.
>
> To the best of our knowledge, neither OpenFold nor Boltz has publicly released a complete set of inference results of human protein structures, compared with the AlphaFold2 dataset whose complete inference results can be downloaded publicly. Since protein structure inference is computationally extremely expensive (typically 0.5 to 5 hours per protein, or even longer, on an NVIDIA A100 GPU depending on sequence length), reproducing the full dataset would not be feasible within a short period of time in the absence of publicly available predictions.
>
> Fortunately, we were able to locate a partial dataset of Boltz inference results (it includes structures of protein complexes predicted by Boltz) [3]. From this work, we extracted the subset overlapping with our CAPSUL benchmark, yielding 1,223 proteins. For these 1,223 proteins, we compared different structure inputs (*i.e.*, structures predicted by AlphaFold2 and by Boltz) on our previously trained structure-based model CDConv for inference. We report the following results (F1-score):
>
> || CDConv (1,223 protein structures from AlphaFold2) | CDConv (1,223 protein structures from Boltz) |
> | ----------------------- | ------------------------------------------------- | -------------------------------------------- |
> | Nucleus| 0.751| 0.658 |
> | Nuclear Membrane| - | - |
> | Nucleoli| 0.113| 0.047|
> | Nucleoplasm| 0.714| 0.572|
> | Cytoplasm| 0.633| 0.653|
> | Cytosol| 0.515| 0.579|
> | Cytoskeleton| 0.147| 0.065|
> | Centrosome| -| -|
> | Mitochondria| 0.311| 0.213|
> | Endoplasmic Reticulum| 0.266| 0.229|
> | Golgi Apparatus| 0.044 | 0.015|
> | Cell Membrane| 0.522| 0.406|
> | Endosome| -| -|
> | Lipid Droplet| -| -|
> | Lysosome/Vacuole| -| -|
> | Peroxisome| -| -|
> | Vesicle| 0.037| 0.019|
> | Primary Cilium| -| -|
> | Secreted Proteins| 0.494| 0.447|
> | Sperm| -| -|
> | **Micro Avg F1-score**  | 0.527| 0.477|
> | **Macro Avg F1-score**  | 0.227| 0.195|
> | **Micro Avg Precision** | 0.745| 0.668|
> | **Micro Avg Recall**    | 0.407| 0.371|
>
> Across the overall metric and the majority of subcellular locations, **AlphaFold2-based structural inputs outperform Boltz-based inputs**. This observation further supports the rationale behind our use of AlphaFold2-predicted structures in constructing CAPSUL. As the most accurate and widely adopted source of protein structural information, AlphaFold2 provides high-quality structural inputs that lead to strong downstream performance in subcellular localization.
>
> [3] Ille A M, et al. Human protein interactome structure prediction at scale with Boltz-2[J]. bioRxiv, 2025: 2025.07. 03.663068.

---

> ### Author Response · Authors · 2025-11-22
> **Response to Reviewer Xxcu (Part 3/3)**
>
> > Q3 (continued). Additional baselines of OpenFold and Boltz.
>
> For more downstream baselines which go beyond traditional sequence- and structure-based models as you mentioned, we have supplemented **1) the graph diffusion model, 2) the CDConv + contrastive learning model, and 3) the ESMC + CDConv fusion model** during the rebuttal period. You may refer to their results, which will help enrich the evaluation across more model types and different modalities:
>
> || Graph Diffusion Model | CDConv (incorporate contrastive loss) | ESMC+CDConv (early fusion) | ESMC+CDConv (late fusion) |
> | ----------------------- | --------------------- | ------------------------------------- | -------------------------- | ------------------------- |
> | Nucleus| 0.624 | 0.592| 0.643| 0.645|
> | Nuclear Membrane| -| -| -| -|
> | Nucleoli| 0.047| 0.140| 0.125| 0.153|
> | Nucleoplasm| 0.578| 0.556| 0.643| 0.617|
> | Cytoplasm| 0.503| 0.480| 0.455| 0.515|
> | Cytosol| 0.288| 0.250| 0.157| 0.370|
> | Cytoskeleton| 0.099| 0.243| 0.100| 0.287|
> | Centrosome| 0.014| -| -| 0.037|
> | Mitochondria| 0.303| 0.468| 0.563| 0.557|
> | Endoplasmic Reticulum| 0.150| 0.361| 0.446| -|
> | Golgi Apparatus| -| 0.156| 0.249| -|
> | Cell Membrane| 0.496| 0.539| 0.629| 0.673|
> | Endosome| -| 0.034| 0.054| -|
> | Lipid Droplet| -| -| -| -|
> | Lysosome/Vacuole| -| -| 0.026| -|
> | Peroxisome| -| -| -| -|
> | Vesicle| 0.018| 0.027| 0.067| -|
> | Primary Cilium| -| 0.036| -| 0.115|
> | Secreted Proteins| 0.623| 0.780| 0.819| 0.725|
> | Sperm| -| 0.018| -| -|
> | **Micro Avg F1-score**  | 0.424| 0.435| 0.470| 0.476|
> | **Macro Avg F1-score**  | 0.187| 0.234| 0.249| 0.235|
> | **Micro Avg Precision** | 0.596| 0.650| 0.710| 0.634|
> | **Micro Avg Recall**    | 0.329| 0.326| 0.351| 0.381|
>
>
>
> We sincerely appreciate your insightful comments, which have helped us improve the clarity of the manuscript and strengthen our evaluation.

---

> ### Comment · Reviewer_Xxcu · 2025-11-25
>
> Thanks for the detailed rebuttal. The responses address my concerns. I'd like to retain my score.

---

> ### Author Response · Authors · 2025-11-26
> **Follow-up on rebuttal discussion**
>
> Dear Reviewer `Xxcu`,
>
> We have just updated the PDF manuscript that includes our responses during the rebuttal period. With regard to your valuable feedback, we have updated: 1) data reliability of AlphaFold2 in *Supp.* Section B.1; 2) ongoing efforts to improve CAPSUL in *Supp.* Section K; 3) results and analysis of using Boltz-predicted protein structure data in *Supp.* Section B.2; 4) results and analysis of more baselines in the main text Section 4, along with its detailed implementation in *Supp.* Section C. **All revisions have been highlighted in blue for your convenience to locate and review them quickly.**
>
> We are delighted to notice that your concerns are fully addressed, so we would be truly grateful if you would consider reflecting that in your scoring! Thank you for your time and constructive feedback!

---

### Official Review · Reviewer_68FM · 2025-10-28

**Soundness:** 2
**Presentation:** 3
**Contribution:** 2
**Rating:** 4
**Confidence:** 1

**Summary:**

The paper presents CAPSUL, a benchmark of 20,181 human proteins with amino-acid sequences, Cα coordinates, and 3Di structural tokens, paired with fine-grained (20-category) subcellular localization labels aggregated from UniProt and HPA, along with evidence levels. It evaluates both sequence- and structure-based models (ESM-2/ESM-C, FoldSeek, GCN variants, Graph Transformer, Graph Mamba), explores reweighting and single-label strategies, and provides an interpretability case study (Golgi apparatus) via attention.

**Strengths:**

1.Unified access to structure (Cα, 3Di tokens) plus fine-grained labels and evidence levels; a clear advance beyond existing sequence-only/coarse-label datasets.
2.Broad coverage of representative sequence and structure baselines; reasonable class-imbalance mitigations (reweighting, focal, single-label) and a “randomized structure” ablation that is logically sound.

**Weaknesses:**

1.Evidence-level integration may introduce bias: treating non-experimental annotations as positives could inflate text biases.
2.Missing graph-construction details, i.e., edge criteria (kNN/sequence adjacencies), edge features (relative orientation/distance encodings), normalization, length truncation.
3.You are suggested that provide failure-case analyses (e.g., low pLDDT regions, disordered segments).
4.You should fix minor typos and keep notation consistent.
5.It is better to add fusion baselines (sequence+structure early/late fusion) to probe complementarity.

**Questions:**

None

---

> ### Author Response · Authors · 2025-11-22
> **Response to Reviewer 68FM (Part 1/4)**
>
> Dear Reviewer `68FM`,
>
> Thanks for your inspiring comments! We sincerely appreciate your great effort to review and your positive recognition of our work. As you noted, our work represents a substantial advance beyond existing protein sequence-only and coarse-grained subcellular localization datasets. We establish a fair and consistent comparison across different baselines, and we further conduct ablation studies on the role of structural inputs as well as explore multiple feasible training strategies.
>
> Following your comments and suggestions, we have provided detailed explanations and additional experimental results to address your concerns. Please feel free to leave further comments or questions if there is any misunderstanding, and we will reply soon.
>
>
>
> >Q1. Possible bias caused by evidence-level annotations.
>
> **Reply:** We fully understand your concerns about "treating non-experimental annotations as positives" as stated in our paper. We have already provided a detailed discussion and additional experiments regarding the evidence-level annotation in *Supp.* Section F. In addition to that, we further conduct additional experiments for comprehensive analysis. Here, we offer further clarification and interpretation:
>
> - **The original intent behind distinguishing experimentally validated annotations was to accommodate the diverse needs of different researchers.** For studies requiring highly rigorous data, one may exclude all non–experimentally validated annotations to ensure high-confidence labels. However, in practice, many non-validated annotations also originate from reliable sources (*e.g.*, the `ECO:0000303` code in the UniProt database indicates that the localization information is extracted from published literature). Therefore, for large-scale deep learning training as our work does, including such annotations can increase sample diversity and improve data richness, and therefore, improve models' performance.
> - **We have already included three strategies for handling evidence-level annotations in the supplementary materials.** In *Supp.* Section F, we examine three additional strategies for handling these annotations beyond the approach used in the main text: 1) Following the same setting as in the main text, *i.e.*, treating non–experimentally validated annotations as positive samples; 2) Weighting labels, *i.e.*, treating non–experimentally validated annotations as positive samples but assigning them a weight of 0.7 relative to experimentally validated ones, thereby reducing their influence on model learning; and 3) Filtering labels, *i.e.*, treating non–experimentally validated annotations as negative samples, which restricts the model to learning solely from high-reliability experimental data.
>
> During our rebuttal, to have a comprehensive analysis of the effect of evidence level, we further extended the weighting-label strategy by setting different weights to samples with non-experimentally validated annotations, and obtained the following results (F1-score) under different weighting settings:
>
> | CDConv (weight)     | 0.1   | 0.3   | 0.5   | 0.7   | 0.9   |
> | ------------------- | ----- | ----- | ----- | ----- | ----- |
> | Micro Avg F1-score  | 0.442 | 0.450 | 0.444 | 0.438 | 0.456 |
> | Macro Avg F1-score  | 0.205 | 0.213 | 0.208 | 0.211 | 0.218 |
> | Micro Avg Precision | 0.631 | 0.617 | 0.629 | 0.637 | 0.625 |
> | Micro Avg Recall    | 0.340 | 0.354 | 0.343 | 0.333 | 0.364 |
>
> | ESMC (weight)       | 0.1   | 0.3   | 0.5   | 0.7   | 0.9   |
> | ------------------- | ----- | ----- | ----- | ----- | ----- |
> | Micro Avg F1-score  | 0.453 | 0.452 | 0.466 | 0.481 | 0.495 |
> | Macro Avg F1-score  | 0.223 | 0.227 | 0.231 | 0.245 | 0.260 |
> | Micro Avg Precision | 0.687 | 0.682 | 0.685 | 0.700 | 0.683 |
> | Micro Avg Recall    | 0.338 | 0.338 | 0.353 | 0.366 | 0.388 |

---

> ### Author Response · Authors · 2025-11-22
> **Response to Reviewer 68FM (Part 2/4)**
>
> >Q1 (continued). Possible bias caused by evidence-level annotations.
>
> From the results above and the *Supp.*, we have the following observation:
>
> - For certain subcellular compartments, baselines trained exclusively on experimentally validated annotations or assigned low positive weights to non-experimentally validated annotations generally **achieve higher precision**. Actually, precision and recall often represent a trade-off in modeling strategies. That is, adopting a more conservative prediction strategy typically increases precision but reduces the number of correctly recalled samples, and vice versa. Therefore, selecting high-confidence evidence levels can be seen as **a method of enforcing a more conservative prediction approach, helping to reduce the likelihood of false-positive predictions**. This highlights the novelty of evidence-level annotations: **using experimentally validated data is considered a strategy to ensure high precision and reliability**.
>
>   | ESMC (weight)         | 0.1   | 0.3   | 0.5   | 0.7   | 0.9   | treat as positive |
>   | --------------------- | ----- | ----- | ----- | ----- | ----- | ----------------- |
>   | Precision for Nucleus | 0.733 | 0.730 | 0.728 | 0.717 | 0.711 | 0.694             |
>   | Recall for Nucleus    | 0.549 | 0.553 | 0.564 | 0.585 | 0.578 | 0.609             |
>
> - However, for overall results and most subcellular compartments, the results among the three strategies **show no significant differences**. This indirectly supports the notion we mentioned above that even **non-validated annotations in CAPSUL still possess relatively high reliability**. This ensures that, when CAPSUL is used for downstream tasks, the inclusion of non-experimentally validated annotations does not introduce substantial bias.
>
> We will update the detailed results in the manuscript after addressing all reviewers' comments. Thanks for your advice again!
>
>
>
> > Q2. Missing graph-construction details.
>
> **Reply:** Thanks for your valuable advice! We agree with you that detailed implementations for the graph constructions should be included for better understanding. We will supplement this information uniformly in the manuscript after addressing all reviewers' comments. Here, we provide further clarification on several important task-specific hyperparameters you mentioned:
>
> - **Edge Criteria**. **Sequential adjacency** refers to the proximity of amino acids along the one-dimensional primary sequence of a protein (*e.g.*, if the sequential adjacency range is set to 3, then the amino acids from position [x−3, x+3] are considered sequential neighbors of the x-th residue). On the other hand, **spatial adjacency** captures the proximity of amino acids in the three-dimensional space of the protein (*e.g.*, if the spatial adjacency radius is set to 8 Å, all amino acids located within an 8 Å sphere centered at a given residue are considered its spatial neighbors). These adjacency relationships define the edges in the constructed protein graph.
>
> - **Edge Features**. For the GCN-based baselines CDConv and GearNet-Edge, we adopted their innovative edge feature implementation methods originally proposed in their respective frameworks, which can be found in the corresponding publications [1, 2]. These methods incorporate and encode both relative orientation and Euclidean distance. For our extended models, Graph Transformer and Graph Mamba, the edge features are derived by processing the aforementioned edge criteria through an embedding layer.
>
> [1] Hehe Fan, et al. Continuous-discrete convolution for geometry-sequence modeling in proteins. In The Eleventh International Conference on Learning Representations, 2022
>
> [2] Zuobai Zhang, et al. Protein representation learning by geometric structure pretraining. arXiv preprint arXiv:2203.06125, 2022.

---

> ### Author Response · Authors · 2025-11-22
> **Response to Reviewer 68FM (Part 3/4)**
>
> > Q3. Failure-case analysis.
>
> **Reply:** We fully understand your concern that low-pLDDT regions may lead the model to make incorrect predictions due to reduced structural accuracy. Here, we further elaborate on pLDDT and disordered segments raised in your comments, and we provide an additional analysis to demonstrate that CAPSUL remains unaffected by this factor in most cases.
>
> In AlphaFold-predicted structures, regions with low pLDDT often indicate intrinsically disordered segments, which naturally lack stable tertiary structure and are therefore harder for any predictor to model with high confidence. To alleviate your concern, **we conducted an analysis on the representative structure-based model, CDConv, examining the relationship between residue-level pLDDT and model performance across the test set.** Specifically, we divided all 3,028 proteins in the test set into three groups of equal size based on their protein-level mean pLDDT values. The highest and lowest mean pLDDT values within each group are [83.56, 99.39], [70.40, 83.54], and [28.11, 70.40]. We then compared the performance of these groups to determine whether substantial performance discrepancies exist, which would suggest that the model is sensitive to variations in structural confidence. We report the results (F1-score) of three groups respectively:
>
> || high pLDDT test set | medium pLDDT test set | low pLDDT test set |
> | ----------------------- | ------------------- | --------------------- | ------------------ |
> | **Micro Avg F1-score**  | 0.443| 0.446| 0.469|
> | **Macro Avg F1-score**  | 0.233| 0.228| 0.179|
> | **Micro Avg Precision** | 0.617| 0.628| 0.651|
> | **Micro Avg Recall**    | 0.345| 0.345| 0.367|
>
> Our analysis shows that from an overall perspective, **there is no significant performance difference between the high-pLDDT and low-pLDDT groups.** Upon further examination of specific subcellular categories, we find that:
>
> - For some subcellular locations, the performance differences among the three groups may appear larger. This is probably attributable to the uneven distribution of positive samples across the groups. For example, in the Nucleus category, the high-pLDDT group contains only 318 positive samples, whereas the low-pLDDT group contains 476 positive samples, which contributes to this noticeable discrepancy:
>
>   || high pLDDT test set | medium pLDDT test set | low pLDDT test set |
>   | ------------- | ------------------- | --------------------- | ------------------ |
>   | Nucleus| 0.462| 0.601| 0.720|
>   | Cytoplasm| 0.553| 0.488| 0.385|
>   | Cytosol| 0.483| 0.318| 0.177|
>   | Cell Membrane | 0.588| 0.653| 0.360|
>
> - For other subcellular locations, the results across the three groups do not exhibit pronounced differences. This observation is consistent with the trend shown by the overall evaluation metrics above, indicating that proteins with different average pLDDT levels have not greatly influenced the prediction of these subcellular localizations:
>
>   || high pLDDT test set | medium pLDDT test set | low pLDDT test set |
>   | ----------------- | ------------------- | --------------------- | ------------------ |
>   | Cytoskeleton| 0.112| 0.210| 0.065|
>   | Secreted Proteins | 0.760| 0.782| 0.755|
>
> In summary, **the low-pLDDT group does not exhibit systematically worse predictions**. This indicates that the CAPSUL is robust to fluctuations in pLDDT and does not rely disproportionately on regions of high structural confidence. In other words, **CAPSUL provides stable and reliable structural inputs for downstream tasks even when proteins contain disordered or low-confidence regions, demonstrating the quality and suitability of our structural dataset for subcellular localization prediction.** However, results of particular subcellular compartments highlight the need for further exploration of developing more informative and robust structural representations to better capture the determinants of subcellular localization in future work.
>
>
>
> > Q4. Minor typos and consistent notation.
>
> **Reply:** Thank you very much for your valuable suggestions. We have re-examined the entire manuscript and identified the following typos and instances of inconsistent notation:
>
> - In Table 3, the subcategories under "Nucleus" and "Cytoplasm" are not properly indented.
> - The third-person verb form in line 50 is incorrect.
> - The letter following the comma should not be capitalized in line 481.
> - The formulas in lines 871 and 877 contain variables that are not explicitly defined or explained.
>
> We will revise collectively after addressing all reviewers’ comments. If we have overlooked any issues, please feel free to point them out—we will correct them immediately. We sincerely appreciate your careful and thoughtful review again.

---

> ### Author Response · Authors · 2025-11-22
> **Response to Reviewer 68FM (Part 4/4)**
>
> > Q5. Sequence+structure fusion models.
>
> **Reply:** Thank you very much for your valuable suggestions. Firstly, we would like to clarify that **structure-based models also take one-dimensional protein sequences as input**, using an embedding layer to initialize node representations in the graph. They are called "structure-based" because their primary innovation lies in performing graph propagation over 3D adjacency structures. We fail to explicitly mention that, which may lead to potential misunderstanding, and we will revise it in the updated manuscript.
>
> But we fully agree that the simple embedding layers typically used by structure-based models may insufficiently capture rich protein sequential features, whereas pretrained protein language models can effectively address this limitation. **Following your suggestion, we have added early and late fusion experiments combining representative sequence-based models (ESMC) and structure-based models (CDConv).**
>
> The models are defined as follows: 1) in **early fusion**, the structural representations produced by CDConv (without the additional Transformer architecture introduced by our paper) for each amino acid are added to the initial protein embedding of ESMC. The combined representation is then passed through the pretrained ESMC Transformer for interaction, followed by mean pooling to obtain a protein-level representation for downstream classification. 2) The **late fusion** setting is similar, except that the structural representations from CDConv are added to the final sequence representation produced by ESMC before mean pooling for the downstream classification task. We report the following results (F1-score):
>
> || ESMC+CDConv (early fusion) | ESMC+CDConv (late fusion) |
> | ----------------------- | -------------------------- | ------------------------- |
> | Nucleus| 0.643| 0.645|
> | Nuclear Membrane| -| -|
> | Nucleoli| 0.125| 0.153|
> | Nucleoplasm| 0.643| 0.617|
> | Cytoplasm| 0.455| 0.515|
> | Cytosol| 0.157| 0.370|
> | Cytoskeleton| 0.100| 0.287|
> | Centrosome| -| 0.037|
> | Mitochondria| 0.563| 0.557|
> | Endoplasmic Reticulum| 0.446| -|
> | Golgi Apparatus| 0.249| -|
> | Cell Membrane| 0.629| 0.673|
> | Endosome| 0.054| -|
> | Lipid Droplet| -| -|
> | Lysosome/Vacuole| 0.026| -|
> | Peroxisome| -| -|
> | Vesicle| 0.067| -|
> | Primary Cilium| -| 0.115|
> | Secreted Proteins| 0.819| 0.725|
> | Sperm| -| -|
> | **Micro Avg F1-score**  | 0.470| 0.476|
> | **Macro Avg F1-score**  | 0.249| 0.235|
> | **Micro Avg Precision** | 0.710| 0.634|
> | **Micro Avg Recall**    | 0.351| 0.381|
>
> From the results above, we observe that **for certain subcellular compartments (***e.g.***, Nucleoplasm, Cytoskeleton, Mitochondria, ER, and Cell Membrane), the fusion models achieve the best F1-score across all baseline models.** However, the overall performance of the two fusion models lies between that of the best sequence-based model (ESMC, micro-average F1-score of 0.495) and the best structure-based model (CDConv, micro-average F1-score of 0.452). We believe that the absence of a substantial performance improvement from combining the two models may be attributed to the following reasons:
>
> - **There exists an inherent gap between one-dimensional and three-dimensional representations.** Sequence-based models primarily capture evolutionary and biochemical signals encoded in amino acid orders, whereas structure-based models encode geometric and spatial dependencies that are not directly recoverable from linear sequences. This mismatch in representational space can make fusion challenging, as the two modalities emphasize different types of information. Therefore, **the fusion strategy we adopted is relatively trivial**, and more sophisticated techniques may be required to effectively integrate the embeddings from the two modalities.
> - **Using the same backbone to construct a unified model may exert negative influences**, given the inherent gap between the two modalities. If we directly perform sequence-level interactions within a single backbone (*e.g.*, transformer-based ESM), the high-order connections and interactions encoded in the 3D structure may instead be diluted or lost. Furthermore, this modality gap implies that enforcing interaction within the same backbone may lead the two information sources to function as mutual noise.
>
> Given these observations, we leave the exploration of more advanced fusion strategies to future work, aiming to maximally leverage both one-dimensional and three-dimensional protein information. For instance, it can be considered to employ cross-attention mechanisms or tailored alignment strategies, which are promising directions for the future development of the subcellular localization task.
>
> We will update the manuscript to include the detailed experimental setting and the detailed results after addressing all reviewers’ comments.
>
>
>
> We sincerely appreciate your insightful comments, which have helped us enrich our evaluation and improve the clarity of the manuscript.

---

> ### Author Response · Authors · 2025-11-26
> **Follow-up on rebuttal discussion**
>
> Dear Reviewer `68FM`,
>
> We have just updated the PDF manuscript that includes our responses during the rebuttal period. With regard to your valuable feedback, we have updated: 1) further explanation and analysis on evidence-level annotations in *Supp.* Section F; 2) graph-construction details in *Supp.* Section C; 3) analysis of disordered segments and pLDDT's impacts in *Supp.* Section H; 4) correction of minor typos and inconsistent notations as we figured out in rebuttal response; 5) results and analysis of sequence+structure fusion model in the main text Section 4. **All revisions have been highlighted in blue for your convenience to locate and review them quickly.**
>
> As the discussion deadline is approaching, we would like to kindly check whether our response has addressed all your concerns satisfactorily. Your insights are valuable to enhance our work, so please let us know if there are any remaining points that are still unclear.
>
> If your concerns are addressed, we would be truly grateful if you would consider reflecting that in your scoring! Thank you for your time and constructive feedback!

---

### Official Review · Reviewer_axjf · 2025-10-31

**Soundness:** 3
**Presentation:** 3
**Contribution:** 3
**Rating:** 8
**Confidence:** 4

**Summary:**

This paper introduces CAPSUL, a new large-scale human protein benchmark for subcellular localization that integrates comprehensive 3D structural information (Cα coordinates and FoldSeek 3Di tokens) with fine-grained, expert-curated annotations across 20 subcellular compartments. The dataset unifies information from UniProt and the Human Protein Atlas and includes evidence-level labels for experimental support. The authors benchmark a diverse range of state-of-the-art sequence-based and structure-based models, analyse class imbalance, and explore interpretability via Transformer attention. Results demonstrate that explicit 3D geometry provides predictive power comparable to massive sequence pre-training, and that fine-grained compartmental labels uncover biologically meaningful hierarchical structure in localization processes. The paper thus establishes a strong and reproducible foundation for evaluating future multimodal protein models.

**Strengths:**

**Originality and significance**

* CAPSUL fills a clear and impactful gap in current bio-ML resources by providing the first benchmark where **structural and sequence modalities can be directly compared** for subcellular localization.
* The study yields a **quantitative insight into the trade-off between pre-training and structure**, showing that a modestly sized structure-aware model can match the performance of billion-parameter sequence-only models trained on hundreds of millions of proteins. This constitutes a valuable empirical reference for the community.
* The dataset’s **fine-grained, hierarchical labeling** reveals that localization is a multi-stage process, aligning with known biological transport hierarchies (e.g., generic targeting → organelle entry → sub-organelle retention). This suggests natural directions for **hierarchical or multi-task learning architectures**.

**Technical quality**

* The benchmark suite is broad and technically sound: eight representative models spanning sequence Transformers, geometric GNNs, and graph Transformers.
* Ablations on randomised coordinates convincingly isolate the contribution of geometric structure.
* The exploration of reweighting and single-label training provides concrete, actionable strategies for imbalance mitigation.
* Interpretability analyses identify α-helix transmembrane motifs consistent with established Golgi localization mechanisms—an impressive validation of biological fidelity.

**Clarity and reproducibility**

* The paper is well organised and carefully written; data processing steps are fully documented with evidence codes and validation statistics.
* Supplementary material provides sufficient implementation detail for replication.

**Impact**

* CAPSUL is likely to become a **standard benchmark** for structure-aware protein models. Its explicit link between geometry, hierarchy, and localization creates opportunities for causal and interpretable modeling in computational biology.

**Weaknesses:**

* The **structure–sequence trade-off** is not yet quantified in a controlled architectural setting. While the results suggest that explicit structure compensates for the absence of large-scale pre-training, a **single unified model trained in both modalities** (e.g., ESM backbone + structural tokens) would enable direct estimation of their relative contributions.
* The paper stops short of leveraging the **hierarchical organization of the labels**. Implementing or evaluating hierarchical loss functions or coarse-to-fine classifiers would make the biological interpretation stronger and could address imbalance more effectively.
* Although the case study on the Golgi is compelling, additional interpretability analyses across other organelles would reinforce the claim that structure-based attention consistently yields mechanistically meaningful motifs.
* The benchmark could report **sample efficiency curves** (performance vs number of training samples) to further substantiate the “structure as information equivalent to pre-training” argument.

**Questions:**

1. **Quantifying modality equivalence** – Could the authors train a unified model that accepts both sequence and structure to explicitly measure how much performance gain is attributable to each modality? This would clarify whether “one structured protein ≈ 10⁴–10⁵ sequences” holds empirically.
2. **Hierarchical classification** – Given that many compartments (e.g., nucleus → nucleolus, nucleoplasm) are nested, have the authors considered a hierarchical target formulation or conditional prediction pipeline? Such models might reflect biological transport stages and improve minority-class recall.
3. **Causal alignment** – Can the authors comment on whether CAPSUL could enable causal studies linking specific structural motifs (e.g., transmembrane helix length or charge distribution) to localization outcomes?
4. **Fusion and transfer learning** – Have experiments been attempted where pretrained sequence embeddings are fused with structural graphs? This could help quantify complementarity and guide future multimodal protein models.
5. **Dataset generality** – While CAPSUL focuses on human proteins, do the authors foresee extending it to other organisms or to dynamic (context-dependent) localization? Such extensions could test generalisation and evolutionary transfer.

---

> ### Author Response · Authors · 2025-11-22
> **Response to Reviewer axjf (Part 1/4)**
>
> Dear Reviewer `axjf`,
>
> Thanks for your inspiring comments! We sincerely appreciate your great effort to review and your positive recognition of our work! As you noted, our work fills a clear and impactful gap in incorporating protein structures into the subcellular localization task. Our carefully designed protein data and annotations open avenues for hierarchical and interpretable modeling. CAPSUL is technically robust, biologically validated, and fully reproducible, with extensive ablations and analyses. We anticipate that CAPSUL will become a standard resource for computational biology and pave the way for interpretable cell biology research with deep learning assistance.
>
> Following your comments and suggestions, we have provided detailed explanations and additional experimental results to address your concerns. Please feel free to leave further comments or questions if there is any misunderstanding, and we will reply soon.
>
>
>
> >Q1 (corresponding to Weakness 1 and Question 1&4). Unified fusion models.
>
> **Reply:** Thank you very much for your constructive suggestions. **Following your suggestion, we have added early and late fusion experiments combining representative sequence-based models (ESMC) and structure-based models (CDConv).** Your advice of "ESM backbone + structural tokens" gives us a new perspective on evaluating the respective contribution of sequential and structural information.
>
> The models are defined as follows: 1) in **early fusion**, the structural representations produced by CDConv (without the additional Transformer architecture introduced by our paper) for each amino acid are added to the initial protein embedding of ESMC. The combined representation is then passed through the pretrained ESMC Transformer for interaction, followed by mean pooling to obtain a protein-level representation for downstream classification. 2) The **late fusion** setting is similar, except that the structural representations from CDConv are added to the final sequence representation produced by ESMC before mean pooling for the downstream classification task. We report the following results (F1-score):
>
> || ESMC+CDConv (early fusion) | ESMC+CDConv (late fusion) |
> | ----------------------- | -------------------------- | ------------------------- |
> | Nucleus| 0.643| 0.645|
> | Nuclear Membrane| -| -|
> | Nucleoli| 0.125| 0.153|
> | Nucleoplasm| 0.643| 0.617|
> | Cytoplasm| 0.455| 0.515|
> | Cytosol| 0.157| 0.370|
> | Cytoskeleton| 0.100| 0.287|
> | Centrosome| -| 0.037|
> | Mitochondria| 0.563| 0.557|
> | Endoplasmic Reticulum| 0.446| -|
> | Golgi Apparatus| 0.249| -|
> | Cell Membrane| 0.629| 0.673|
> | Endosome| 0.054| -|
> | Lipid Droplet| -| -|
> | Lysosome/Vacuole| 0.026| -|
> | Peroxisome| -| -|
> | Vesicle| 0.067| -|
> | Primary Cilium| -| 0.115|
> | Secreted Proteins| 0.819| 0.725|
> | Sperm| -| -|
> | **Micro Avg F1-score**  | 0.470| 0.476|
> | **Macro Avg F1-score**  | 0.249| 0.235|
> | **Micro Avg Precision** | 0.710| 0.634|
> | **Micro Avg Recall**    | 0.351| 0.381|

---

> ### Author Response · Authors · 2025-11-22
> **Response to Reviewer axjf (Part 2/4)**
>
> >Q1 (corresponding to Weakness 1 and Question 1&4) (continued). Unified fusion models.
>
> From the results above, we observe that **for certain subcellular compartments (***e.g.***, Nucleoplasm, Cytoskeleton, Mitochondria, ER, and Cell Membrane), the fusion models achieve the best F1-score across all baseline models.** It showcases that the simple embedding layers typically used by structure-based models may insufficiently capture rich protein sequential features, whereas pretrained protein language models can effectively address this limitation. Therefore, incorporating sequence-based models' embeddings into structure-based models will greatly benefit their performance.
>
> However, the overall performance of the two fusion models lies between that of the best sequence-based model (ESMC, micro-average F1-score of 0.495) and the best structure-based model (CDConv, micro-average F1-score of 0.452). We believe that the absence of a substantial performance improvement from combining the two models may be attributed to the following reasons:
>
> - **There exists an inherent gap between one-dimensional and three-dimensional representations.** Sequence-based models primarily capture evolutionary and biochemical signals encoded in amino acid orders, whereas structure-based models encode geometric and spatial dependencies that are not directly recoverable from linear sequences. This mismatch in representational space can make fusion challenging, as the two modalities emphasize different types of information. Therefore, **the fusion strategy we adopted is relatively trivial**, and more sophisticated techniques may be required to effectively integrate the embeddings from the two modalities.
> - **Using the same backbone to construct a unified model may exert negative influences**, given the inherent gap between the two modalities. If we directly perform sequence-level interactions within a single backbone (*e.g.*, transformer-based ESM), the high-order connections and interactions encoded in the 3D structure may instead be diluted or lost. Furthermore, this modality gap implies that enforcing interaction within the same backbone may lead the two information sources to function as mutual noise.
>
> Given these observations, we leave the exploration of more advanced fusion strategies to future work, aiming to maximally leverage both one-dimensional and three-dimensional protein information. For instance, it can be considered to employ cross-attention mechanisms or tailored alignment strategies, which are promising directions for the future development of the subcellular localization task.
>
> We will update the manuscript to include the detailed experimental setting and the detailed results after addressing all reviewers' comments. Thanks again for your constructive review!
>
>
>
> > Q2 (corresponding to Weakness 2 and Question 2). Further attempts at hierarchical classification.
>
> **Reply:** We sincerely appreciate your valuable suggestions. Following your advice, we attempted to train a hierarchical classifier for the two nested categories, **Nucleus** and **Cytoplasm**, in CAPSUL. These classifiers were built upon CDConv, using positive samples of the corresponding parent categories as the training set. We report the following experimental results (F1-score):
>
> || CDConv | CDConv (hierarchical classifier) |
> | ----------------------------- | ------ | -------------------------------- |
> | **Nucleus's subcategories**   |||
> | Nuclear Membrane| -| -|
> | Nucleoli| 0.147  | -|
> | Nucleoplasm| 0.583  | 0.507|
> ||||
> | **Cytoplasm's subcategories** |||
> | Cytosol| 0.353  | 0.411|
> | Cytoskeleton| 0.135  | 0.182|
>
> From the results above, there are two observations:
>
> - For three subcategories within the nucleus, the hierarchical classifier actually leads to decreased performance. We attribute this to the reduced number of training samples when training this hierarchical classifier, which, combined with the already severe class imbalance in these subcategories, likely exacerbated the issue. Considering the results discussed in Section 4.3 of the main text, we find that for such highly imbalanced subcellular localization tasks, directly applying a single-label classification strategy, while adjusting the corresponding reweighting and classification thresholds, yields more significant improvements.
> - For the two subcategories within the cytoplasm, classification performance greatly improves, demonstrating that the hierarchical classification strategy can be beneficial for certain subcellular compartments. This result provides an additional strategy, hierarchical classifiers, to address the minority class issue apart from what we have discussed in Section 4.3 of the main text.

---

> ### Author Response · Authors · 2025-11-22
> **Response to Reviewer axjf (Part 3/4)**
>
> > Q3 (corresponding to Weakness 3). More additional interpretability analysis.
>
> **Reply:** Thanks for your valuable comment! In our analysis of the model's ability to recognize structural localization patterns, we identified two more potential rules: 1) **"W-pair"** for Golgi apparatus (*i.e.*, two spatially adjacent Tryptophan residues), which aligns with existing studies suggesting that Tryptophan can influence Golgi targeting [1], and 2) **a flexible region at the N-terminus** of the protein for Mitochondria, which aligns with existing studies suggesting that this area can influence multiple compartments targeting [2]. However, these newly identified patterns still require further experimental validation. Nevertheless, these findings together demonstrate that both our curated dataset and the selected structure-based baseline model are capable of capturing meaningful subcellular localization signals, offering promising insights and directions for future research in cell biology.
>
> [1] Ashlin T G, Blunsom N J, Cockcroft S. Courier service for phosphatidylinositol: PITPs deliver on demand[J]. Biochimica et Biophysica Acta (BBA)-Molecular and Cell Biology of Lipids, 2021, 1866(9): 158985.
>
> [2] Sohn S, Joe M K, Kim T E, et al. Dual localization of wild-type myocilin in the endoplasmic reticulum and extracellular compartment likely occurs due to its incomplete secretion[J]. Molecular vision, 2009, 15: 545.
>
>
>
> > Q4 (corresponding to Weakness 4). Sample efficiency curves.
>
> **Reply:** We fully agree that a sample efficiency curve would explicitly reflect a possible strategy to choose the proper size of training set, which strikes a balance between model performance and computational costs. We randomly select training subsets of varying sizes and evaluate their sample efficiency curves on CDConv (here presented in a tabular format, and more intuitive line plots will be added to the manuscript after addressing all reviewers' comments):
>
> | CDConv (training set size)| 128  | 256  | 512  | 1,024 | 2,048 | 4,096 | 8,192 | 14,126 (original CAPSUL) |
> | ---------------------------------- | ---- | ---- | ---- | ----- | ----- | ----- | ----- | ------------------------ |
> | Successfully identified categories | 0| 0| 0| 3| 5| 5| 7| 12|
> | Micro Avg F1-score| -| -| -| 0.312 | 0.232 | 0.239 | 0.304 | 0.452|
> | Macro Avg F1-score| -| -| -| 0.065 | 0.064 | 0.066 | 0.090 | 0.226|
> | Micro Avg Precision| -| -| -| 0.377 | 0.426 | 0.400 | 0.481 | 0.632|
> | Micro Avg Recall| -| -| -| 0.266 | 0.160 | 0.171 | 0.222 | 0.352|
>
> We observed that the smaller dataset, compared with the original CAPSUL, exhibits noticeably poorer performance, reflected in both lower micro F1-scores and fewer categories for which any positive samples were successfully predicted. This indicates that, in order to achieve satisfactory overall performance as well as decent performance on minority classes, the current size of training samples is relatively appropriate.
>
>
>
> > Q5 (corresponding to Question 3). Potential of causal alignment.
>
> **Reply:** Thank you for your valuable suggestions regarding future directions for CAPSUL! We believe that CAPSUL is well-suited for causal alignment studies. There are some possibly optimistic directions for further exploration:
>
> - We could introduce a protein segment annotation mechanism into the dataset, categorized by properties such as charge distribution and hydrophobicity. This will be combined with analytical experiments, including ablation and contrastive studies, to systematically evaluate the impact of specific structural motifs on subcellular localization.
> - Additionally, we could incorporate structural semantic disentanglement into the model architecture, explicitly separating representations of localized structural features (*e.g.*, transmembrane helices, signal peptides) from global sequence context. This will allow us to perform controlled interventions in the latent space, further strengthening causal inference between individual protein motifs and localization outcomes.
> - Furthermore, techniques based on influence functions can be employed to identify which specific structural components of a protein contribute most to its predicted subcellular localization. By tracing how perturbations to individual structural elements affect the model's output, influence functions provide a principled framework for assessing structure–prediction sensitivity [3].
>
> [3] Koh P W, Liang P. Understanding black-box predictions via influence functions[C]//International conference on machine learning. PMLR, 2017: 1885-1894.

---

> ### Author Response · Authors · 2025-11-22
> **Response to Reviewer axjf (Part 4/4)**
>
> > Q6 (corresponding to Question 5). Dataset generality to other species and dynamic localization.
>
> **Reply:** We fully agree that generality to other species or to dynamic situations is crucial for paving the way for more universal and detailed research. In designing the CAPSUL pipeline, we adopted a vertically deep approach focused specifically on the human species for the following reasons:
>
> - As mentioned in Section 1 of our paper, subcellular localization of proteins is of great importance in research areas such as drug targeting, making studies on human proteins more urgent and practically valuable.
> - Conducting the study within a single species ensures consistency in both the dataset and evaluation criteria (*e.g.*, chloroplasts are only present in plant cells, so mixing human and plant proteins would be inappropriate).
> - Proteins show higher conservation (*i.e.*, structurally and functionally stable) within the same species (*e.g.*, the hepatitis C virus receptor is expressed in human cells but absent in mice). Therefore, focusing on a single species allows for more interpretable and species-specific conclusions.
>
> **The CAPSUL workflow is readily applicable to other species**, and we are committed to developing parallel versions across multiple species to gain insights into cell biology at a broader and more fundamental level:
>
> - **Clarification of CAPSUL's generalization to other species.** While our current study exclusively uses human protein data, we do not assert that CAPSUL's pipeline is intrinsically limited to human proteins. Our dataset and benchmark construction pipeline, including 1) the acquisition of one-dimensional and three-dimensional protein information, 2) the collection of localization annotations, and 3) the comparison of various baselines, is entirely **species-agnostic**.
> - **Future exploration of multi-species benchmark.** Once the CAPSUL pipeline becomes robust and well-received, in our future work, we would extend the same workflow to develop subcellular localization datasets and benchmarks across multiple species in parallel. This extension will not only enable broader biological investigations but also provide valuable resources for studying the evolutionary conservation and divergence of protein localization mechanisms across phylogenetic lineages. By comparing subcellular patterns across species, researchers can gain insights into the selective pressures shaping cellular organization, the emergence of organelle-specific functions, and the molecular adaptations that underpin evolutionary innovation.
>
> Also, with regard to the dynamic localization you mentioned, current protein databases lack extensive data on protein structures across diverse biological contexts and their corresponding variations in subcellular localization. If our dataset were augmented with dynamic data capturing protein structural changes (*e.g.*, conformational shifts induced by stressors or ligands) from updated protein databases in the future, **we could retrain CAPSUL to model context-dependent localization**. This would enable the model to make precise predictions about subcellular localization under specific cellular conditions or in response to external environmental states, moving beyond static localization to capture functional biological dynamics. We greatly appreciate your constructive suggestion, and we will add a discussion section clarifying potential generalization in our revised manuscript.
>
>
>
> We sincerely appreciate your insightful comments, which have helped us enrich our evaluation and improve the clarity of the manuscript.

---

> ### Author Response · Authors · 2025-11-27
> **Follow-up on rebuttal discussion**
>
> Dear Reviewer `axjf`,
>
> We have just updated the PDF manuscript that includes our responses during the rebuttal period. With regard to your valuable feedback, we have updated: 1) results and analysis of sequence+structure fusion model in the main text Section 4; 2) results and analysis of hierarchical classification in *Supp.* Section G; 3) more potential interpretability results in *Supp.* Section J; 4) sample efficiency curve in *Supp.* Section I; 5) discussion on CAPSUL's generalization ability to other species and dynamic localization in *Supp.* Section K. **All revisions have been highlighted in blue for your convenience to locate and review them quickly.**
>
> As the discussion deadline is approaching, we would like to kindly check whether our response has addressed all your concerns satisfactorily. Your insights are valuable to enhance our work, so please let us know if there are any remaining points that are still unclear. Thank you for your time and constructive feedback!

---

### Official Review · Reviewer_NKK9 · 2025-11-01

**Soundness:** 2
**Presentation:** 2
**Contribution:** 3
**Rating:** 4
**Confidence:** 3

**Summary:**

Protein subcellular localization is a fundamental aspect of cell biology, closely related to protein function and drug target identification. Although previous studies have shown that 3D protein structure plays a critical role in determining localization patterns, existing datasets provide sequence-level information, lacking structural data. This limits the development and evaluation of structure-aware models. This paper introduces CAPSUL, a comprehensive human protein benchmark for subcellular localization that integrates 3D structural information with fine-grained localization annotations.

**Strengths:**

1. CAPSUL combines 3D structural data with detailed subcellular localization annotations, enabling the development of structure-aware models.
2. The dataset includes 20 subcellular compartments, verified by domain experts, ensuring biological accuracy and interpretability.
3. The authors benchmark good structure-based models and propose reweighting and single-label classification strategies to mitigate class imbalance, showing improvements in underrepresented classes.

**Weaknesses:**

1. The current evaluation is limited to supervised multi-label classification. There is no attempt to leverage structural self-supervised learning or contrastive learning, which are promising directions for structure-aware protein modeling.
2. The structure encoders used are standard graph-based models. More advanced geometric deep learning methods (e.g., SE(3)-equivariant networks, structural diffusion models) are not explored, potentially limiting the upper bound of structural understanding.
3. Although evidence codes are provided, in the main experiments, all annotations are treated as positive, including non-experimental ones, which may introduce label noise. While ablations are provided, a more systematic analysis of how evidence levels affect model robustness is lacking.
4. Despite reweighting and single-label strategies, macro-averaged F1-scores remain low for rare classes, indicating that severe imbalance is still a fundamental challenge.

**Questions:**

Refer to weaknesses

---

> ### Author Response · Authors · 2025-11-22
> **Response to Reviewer NKK9 (Part 1/4)**
>
> Dear Reviewer `NKK9`,
>
> Thanks for your inspiring comments! We sincerely appreciate your great effort to review and your positive recognition of our work. As you noted, our work represents a substantial advance beyond existing protein sequence-only localization datasets, which enables the development of structure-aware models. We establish a fair and consistent benchmark across different baselines verified by domain experts, and we further explore multiple feasible training strategies and showcase strong biological interpretability through a Golgi case study.
>
> Following your comments and suggestions, we have provided detailed explanations and additional experimental results to address your concerns. Please feel free to leave further comments or questions if there is any misunderstanding, and we will reply soon.
>
>
>
> >Q1. Limitation of supervised multi-label classification.
>
> **Reply:** We appreciate your valuable suggestions! We further explore incorporating **a contrastive learning mechanism** into the structure-based model CDConv.
>
> Specifically, for each of the 20 subcellular compartments, we construct positive pairs (*i.e.*, pairing protein samples that localize to the same compartment), and positive–negative pairs (*i.e.*, pairing one protein that localizes to the compartment with another that does not). On top of the original loss function, we incorporate a contrastive loss to encourage higher embedding similarity for positive pairs while enforcing lower similarity for positive–negative pairs. We then report the performance (F1-score) of CDConv after incorporating the contrastive loss mechanism:
>
> || CDConv | CDConv (incorporate contrastive loss) |
> | ----------------------- | ------ | ------------------------------------- |
> | Nucleus| 0.620  | 0.592|
> | Nuclear Membrane| -| -|
> | Nucleoli| 0.147  | 0.140|
> | Nucleoplasm| 0.583  | 0.556|
> | Cytoplasm| 0.483  | 0.480|
> | Cytosol| 0.353  | 0.250|
> | Cytoskeleton| 0.135  | 0.243|
> | Centrosome| -| -|
> | Mitochondria| 0.476  | 0.468|
> | Endoplasmic Reticulum   | 0.292  | 0.361|
> | Golgi Apparatus| 0.073  | 0.156|
> | Cell Membrane| 0.562  | 0.539|
> | Endosome| -| 0.034|
> | Lipid Droplet| -| -|
> | Lysosome/Vacuole| -| -|
> | Peroxisome| -| -|
> | Vesicle| 0.027  | 0.027|
> | Primary Cilium| -| 0.036|
> | Secreted Proteins| 0.767  | 0.780|
> | Sperm| -| 0.018|
> | **Micro Avg F1-score**  | 0.452  | 0.435|
> | **Macro Avg F1-score**  | 0.226  | 0.234|
> | **Micro Avg Precision** | 0.632  | 0.650|
> | **Micro Avg Recall**    | 0.352  | 0.326|
>
> We observe that although the introduction of contrastive loss leads to a slight decrease in the micro average F1-score, **it improves performance on several minority classes** (*e.g.*, (1) the macro average F1-score improves, (2) it suceeds in identifying positive samples in Endosome, Primary Cilium and Sperm compared to original CDConv, and (3) the F1-score greatly improves in minority classes such as Golgi Apparatus). We attribute this to the contrastive learning paradigm, which enforces higher similarity among positive samples. This encourages the model to capture shared characteristics within minority-class positive samples through the contrastive objective.

---

> ### Author Response · Authors · 2025-11-22
> **Response to Reviewer NKK9 (Part 2/4)**
>
> >Q1 (continued). Limitation of supervised multi-label classification.
>
> To give a more detailed implementation, we next provide a formal mathematical description of how the contrastive loss is incorporated. Let $\mathbf{\bar{h}}$ denotes the protein-level representations from model (*i.e.*, average embedding obtained before MLP classifier). For each of the 20 independent classification tasks, we compute the cosine similarity between representations of positive–positive pairs and positive–negative pairs, and construct the contrastive loss accordingly.
>
> For class $c \in \\{ 1, \dots, 20 \\}$, let
>
> - $ \mathcal{P}_c = \\{ i \mid y\_{i,c} = 1 \\} $ denote the set of positive samples,
> - $ \mathcal{N}_c = \\{ i \mid y\_{i,c} = 0 \\} $ denote the set of negative samples,
> - $\mathbf{\bar{h}}_i $ denotes the protein-level embedding of the sample $i$.
>
> The positive–positive similarity matrix is
> $$
> S^{(+,+)}_{ij} = \cos(\mathbf{z}_i, \mathbf{z}_j), \quad i,j \in \mathcal{P}_c.
> $$
> We encourage positive samples to be close to each other by minimizing
> $$
> \mathcal{L}^{(+,+)}\_c = 1 - \frac{1}{|\mathcal{P}\_c|^2} \sum\_{i,j \in \mathcal{P}\_c} S^{(+,+)}\_{ij}.
> $$
>
> The positive–negative similarity matrix is
> $$
> S^{(+,-)}\_{ij} = \cos(\mathbf{z}\_i, \mathbf{z}\_j), \quad i \in \mathcal{P}\_c, j \in \mathcal{N}\_c.
> $$
>
> We encourage positive and negative samples to be dissimilar by minimizing
> $$
> \mathcal{L}^{(+,-)}\_c = \frac{1}{|\mathcal{P}\_c||\mathcal{N}\_c|}
>  \sum\_{i \in \mathcal{P}\_c} \sum\_{j \in \mathcal{N}\_c} S^{(+,-)}\_{ij}.
> $$
>
> Thus, the contrastive loss for class $c$ is
> $$
> \mathcal{L}^{\text{contrast}}\_c
>  = \mathcal{L}^{(+,+)}\_c + \mathcal{L}^{(+,-)}\_c.
> $$
>
> Finally, the overall contrastive loss averaged over all classes is
> $$
> \mathcal{L}^{\text{contrast}}
>  = \frac{1}{C} \sum\_{c=1}^{C} \mathcal{L}^{\text{contrast}}\_c.
> $$
>
>
> In summary, this supplementary experiment thus provides a foundation and demonstrates potential for uncovering biological patterns in imbalanced subcellular localization data. We will include the detailed hyperparameter settings and the detailed experimental results in the manuscript after addressing all reviewers’ comments. Once again, we sincerely appreciate your insightful suggestion!
>
>
> > Q2. More advanced geometric deep learning methods.
>
> **Reply:** We sincerely appreciate your valuable suggestions. Regarding your comment on SE(3)-equivariant models, we note that CDConv already incorporates that design, enabling it to maintain invariance under global rotations and translations of the input structure. With respect to your suggestion on structural diffusion models, we agree that this is a highly promising direction for further investigation.
>
> Following your recommendation, **we implement a graph diffusion model.** More specifically, given a protein structure represented as a graph, we introduce a diffusion-based refinement process in which node coordinates or geometric features are gradually perturbed with Gaussian noise and then denoised through a learned reverse process. The diffusion module serves as an auxiliary representation-learning stage designed to enhance geometric feature extraction prior to the downstream subcellular localization prediction. This allows the network to capture multi-scale spatial dependencies while remaining robust to structural noise. We report the corresponding experimental results (F1-score) below:
>
> || Graph Diffusion Model |
> | ----------------------- | --------------------- |
> | Nucleus| 0.624|
> | Nuclear Membrane| -|
> | Nucleoli| 0.047|
> | Nucleoplasm| 0.578|
> | Cytoplasm| 0.503|
> | Cytosol| 0.288|
> | Cytoskeleton| 0.099|
> | Centrosome| 0.014|
> | Mitochondria| 0.303                |
> | Endoplasmic Reticulum   | 0.150|
> | Golgi Apparatus| -|
> | Cell Membrane| 0.496|
> | Endosome| -|
> | Lipid Droplet| -|
> | Lysosome/Vacuole| -|
> | Peroxisome| -|
> | Vesicle| 0.018|
> | Primary Cilium| -|
> | Secreted Proteins| 0.623|
> | Sperm| -|
> | **Micro Avg F1-score**  | 0.424|
> | **Macro Avg F1-score**  | 0.187|
> | **Micro Avg Precision** | 0.596|
> | **Micro Avg Recall**    | 0.329|
>
> The results indicate that the diffusion-based model achieves performance comparable to most structure-based baselines. This suggests that diffusion models hold substantial potential for protein subcellular localization, as their iterative refinement mechanism naturally aligns with the hierarchical and noisy nature of protein structural information. Given that our current diffusion model is still relatively trivial and could be further optimized, we leave the exploration of fully exploiting the capacity of diffusion models for protein representation learning to future work.
>
> We will include the detailed hyperparameter settings and the detailed experimental results in the manuscript after addressing all reviewers’ comments. Thanks for your insightful advice again!

---

> ### Author Response · Authors · 2025-11-22
> **Response to Reviewer NKK9 (Part 3/4)**
>
> > Q3. Possible bias caused by evidence-level annotations.
>
> **Reply:** We fully understand your concerns about "treating non-experimental annotations as positives" as stated in our paper. We have already provided a detailed discussion and additional experiments regarding the evidence-level annotation in *Supp.* Section F. In addition to that, we further conduct additional experiments for comprehensive analysis. Here, we offer further clarification and interpretation:
>
> - **The original intent behind distinguishing experimentally validated annotations was to accommodate the diverse needs of different researchers.** For studies requiring highly rigorous data, one may exclude all non–experimentally validated annotations to ensure high-confidence labels. However, in practice, many non-validated annotations also originate from reliable sources (*e.g.*, the `ECO:0000303` code in the UniProt database indicates that the localization information is extracted from published literature). Therefore, for large-scale deep learning training as our work does, including such annotations can increase sample diversity and improve data richness, and therefore, improve models' performance.
> - **We have already included three strategies for handling evidence-level annotations in the supplementary materials.** In *Supp.* Section F, we examine three additional strategies for handling these annotations beyond the approach used in the main text: 1) Following the same setting as in the main text, *i.e.*, treating non–experimentally validated annotations as positive samples; 2) Weighting labels, *i.e.*, treating non–experimentally validated annotations as positive samples but assigning them a weight of 0.7 relative to experimentally validated ones, thereby reducing their influence on model learning; and 3) Filtering labels, *i.e.*, treating non–experimentally validated annotations as negative samples, which restricts the model to learning solely from high-reliability experimental data.
>
> During our rebuttal, to have a comprehensive analysis of the effect of evidence level, we further extended the weighting-label strategy by setting different weights to samples with non-experimentally validated annotations, and obtained the following results (F1-score) under different weighting settings:
>
> | CDConv (weight)     | 0.1   | 0.3   | 0.5   | 0.7   | 0.9   |
> | ------------------- | ----- | ----- | ----- | ----- | ----- |
> | Micro Avg F1-score  | 0.442 | 0.450 | 0.444 | 0.438 | 0.456 |
> | Macro Avg F1-score  | 0.205 | 0.213 | 0.208 | 0.211 | 0.218 |
> | Micro Avg Precision | 0.631 | 0.617 | 0.629 | 0.637 | 0.625 |
> | Micro Avg Recall    | 0.340 | 0.354 | 0.343 | 0.333 | 0.364 |
>
> | ESMC (weight)       | 0.1   | 0.3   | 0.5   | 0.7   | 0.9   |
> | ------------------- | ----- | ----- | ----- | ----- | ----- |
> | Micro Avg F1-score  | 0.453 | 0.452 | 0.466 | 0.481 | 0.495 |
> | Macro Avg F1-score  | 0.223 | 0.227 | 0.231 | 0.245 | 0.260 |
> | Micro Avg Precision | 0.687 | 0.682 | 0.685 | 0.700 | 0.683 |
> | Micro Avg Recall    | 0.338 | 0.338 | 0.353 | 0.366 | 0.388 |
>
> From the results above and the *Supp.*, we have the following observation:
>
> - For certain subcellular compartments, baselines trained exclusively on experimentally validated annotations or assigned low positive weights to non-experimentally validated annotations generally **achieve higher precision**. Actually, precision and recall often represent a trade-off in modeling strategies. That is, adopting a more conservative prediction strategy typically increases precision but reduces the number of correctly recalled samples, and vice versa. Therefore, selecting high-confidence evidence levels can be seen as **a method of enforcing a more conservative prediction approach, helping to reduce the likelihood of false-positive predictions**. This highlights the novelty of evidence-level annotations: **using experimentally validated data is considered a strategy to ensure high precision and reliability**.
>
>   | ESMC (weight)         | 0.1   | 0.3   | 0.5   | 0.7   | 0.9   | treat as positive |
>   | --------------------- | ----- | ----- | ----- | ----- | ----- | ----------------- |
>   | Precision for Nucleus | 0.733 | 0.730 | 0.728 | 0.717 | 0.711 | 0.694|
>   | Recall for Nucleus    | 0.549 | 0.553 | 0.564 | 0.585 | 0.578 | 0.609|
>
> - However, for overall results and most subcellular compartments, the results among the three strategies **show no significant differences**. This indirectly supports the notion we mentioned above that even **non-validated annotations in CAPSUL still possess relatively high reliability**. This ensures that, when CAPSUL is used for downstream tasks, the inclusion of non-experimentally validated annotations does not introduce substantial bias.
>
> We will update the detailed results in the manuscript after addressing all reviewers’ comments. Thanks for your advice again!

---

> ### Author Response · Authors · 2025-11-22
> **Response to Reviewer NKK9 (Part 4/4)**
>
> > Q4. Dataset imbalance and comparatively low performance on rare classes.
>
> **Reply:** Thanks for your valuable question. We fully understand your concern regarding the imbalanced dataset and comparatively low performance on rare classes. However, we would like to emphasize the importance of **preserving the original data distribution**, as well as our **ongoing commitment** to improving underrepresented classes' performance in future work.
>
> - **Importance of preserving the original data distribution.** We acknowledge that the model's performance may be affected by the presence of minority classes, but deliberate preprocessing procedures (*e.g.*, repeated sampling or fragment-based sampling) could distort the **natural distribution** (*i.e.*, a long-tail pattern) of protein data. It potentially causes the model to overfit to specific proteins within the minority class, rather than learning generalizable features. Also, it is extremely difficult to ensure the **biological validity** if we introduce or delete samples in the dataset, which requires extensive prior knowledge. For these reasons, we choose to preserve the original data distribution and focus on training strategies to mitigate the dataset imbalance issue.
>
> - **Ongoing commitment to improving underrepresented classes' performance.** We also wish to emphasize our strong commitment to continuously maintaining and improving CAPSUL. We have observed a growing number of biological studies contributing new and reliable data sources to the subcellular localization task, which go beyond traditional databases. For example, Hein *et al*. have updated experimentally validated localization data for additional proteins [1]. In future work, we plan to expand the dataset by incorporating diverse sources, including high-throughput proteomics sequencing and low-throughput organelle purification experiments. We also look forward to seeing more baseline models employ innovative processing techniques or training strategies to enhance performance under the dataset imbalance issue. We will continue to update the baseline experimental results to provide the community with the latest progress and future inspiration. We hope this will help address your reasonable concerns regarding the imbalanced dataset.
>
> [1] Hein M Y, Peng D, Todorova V, et al. Global organelle profiling reveals subcellular localization and remodeling at proteome scale[J]. Cell, 2025, 188(4): 1137-1155. e20.
>
>
>
> We sincerely appreciate your insightful comments, which have helped us enrich our evaluation and clarify some crucial concepts in our paper.

---

> ### Author Response · Authors · 2025-11-26
> **Follow-up on rebuttal discussion**
>
> Dear Reviewer `NKK9`,
>
> We have just updated the PDF manuscript that includes our responses during the rebuttal period. With regard to your valuable feedback, we have updated: 1) results and analysis of contrastive learning in the main text Section 4, along with its detailed implementation in *Supp.* Section C.2.3; 2) results of graph diffusion model in the main text Section 4; 3) further explanation and analysis on evidence-level annotations in *Supp.* Section F; 4) ongoing efforts to improve CAPSUL in *Supp.* Section K. **All revisions have been highlighted in blue for your convenience to locate and review them quickly.**
>
> As the discussion deadline is approaching, we would like to kindly check whether our response has addressed all your concerns satisfactorily. Your insights are valuable to enhance our work, so please let us know if there are any remaining points that are still unclear.
>
> If your concerns are addressed, we would be truly grateful if you would consider reflecting that in your scoring! Thank you for your time and constructive feedback!

---

### Author Response · Authors · 2025-12-03
**Brief Summary of Rebuttal and Discussion**

We sincerely appreciate all reviewers for their valuable comments. We are very pleased that our work CAPSUL has been positively recognized for several key contributions, including:

1. **A pioneering effort to incorporate protein 3D input (Reviewer `68FM`, `axjf`, `NKK9`):** Existing datasets and benchmarks on subcellular localization have been limited to sequence inputs, which hinders 3D structure—a well-recognized important signal—to benefit this task. To bridge this gap, CAPSUL represents the first systematic attempt to incorporate 3D structural information into the subcellular localization task.
2. **Comprehensive and informative labels (Reviewer `68FM`, `axjf`, `NKK9`):** CAPSUL curates 20 fine-grained, hierarchical compartment categories verified by domain experts, and provides evidence-level annotations indicating label credibility.
3. **Thorough evaluations on baselines and training strategies (Reviewer `Xxcu`, `68FM`, `axjf`, `NKK9`):** CAPSUL evaluates representative protein representation methods, including sequence- and structure-based ones. We validate the effectiveness of including structural inputs through ablation studies, and explore diverse training strategies in addressing the class imbalance issue.
4. **Significant biological interpretability (Reviewer `Xxcu`, `axjf`, `NKK9`):** CAPSUL showcases tremendous potential for new discovery in cell biology, which is validated by the $\alpha$-helix localization pattern that CAPSUL identifies for the Golgi apparatus.

We also appreciate the detailed feedback regarding potential weaknesses in our work, which provides an opportunity to further improve CAPSUL. We provide additional clarifications and experiments to address the concerns raised by the reviewers. Major concerns include:

1. **More evaluation on potential solutions with advanced structure-based models (Reviewer `Xxcu`, `68FM`, `axjf`, `NKK9`):** We have additionally evaluated three promising models: 1) **Graph Diffusion** deepens the model's comprehension of 3D information by iteratively denoising on protein structure; 2) **CDConv + Contrastive Learning** showcases great potential in minority categories; 3) **Unified model for sequential and structural modeling** leverages a unified pretrained protein language model to represent both sequence and 3D structure information.
2. **Possible bias from evidence-level annotations (Reviewer `68FM`, `NKK9`):** We clarify that 1) the incorporation of evidence-level annotation aims to accommodate different research needs (*e.g.*, only selecting experimentally validated annotations in highly rigorous task such as discovering retention motifs), and 2) introducing labels of different level enriches training data and benefits model learning, supported by our comprehensive label-weighting experiments.
3. **More dataset reliability analysis (Reviewer `Xxcu`, `68FM`):** We validate that AlphaFold's predicted structures possess high credibility through an analysis of pLDDT scores. Also, we conduct experiments on an alternative structure source from Boltz, which further demonstrates the reliability of AlphaFold.
4. **Generalization of our benchmark (Reviewer `Xxcu`, `axjf`, `NKK9`):** We clarify that our benchmark can be easily extended to updated data sources, advanced baselines, and extensive tasks (*e.g.*, localization in other species or dynamic contexts) under the reproducible data construction pipeline of CAPSUL. We are also committed to performing future maintenance to ensure the timeliness of CAPSUL.

We have updated the manuscript, with major revisions highlighted in blue for your convenience.

We regret that some reviewers were unable to provide their feedback before the incident occurred. Nevertheless, **we have carefully provided crucial clarification and additional results to address all the concerns raised by the reviewers.** And we are pleased to have received affirmative acknowledgment from Reviewer `Xxcu` that our responses fully address his/her concerns. Overall, we extend our genuine thanks to the reviewers for their hard work during the review phase. We also convey our appreciation to our AC for their professionalism and dedication. Your constructive comments are inspirational for enhancing our work.

---

### Meta-Review · Area_Chair_LvRe · 2026-01-08

**Summary:**

Reviewers’ decision-relevant concerns centered on whether CAPSUL is (i) a reliable, well-specified benchmark and (ii) whether its evaluation convincingly demonstrates the value of 3D structure beyond sequence-only baselines. Multiple reviewers worried that the initial baseline suite might be missing stronger structure-aware approaches (e.g., more advanced geometric models, diffusion/contrastive/self-supervised ideas), potentially underestimating the “upper bound” for structure; that evidence-level annotations (treating non-experimental labels as positives) could inject label noise/bias unless analyzed systematically; and that reliance on AlphaFold-predicted structures raises questions about data reliability (e.g., low-confidence pLDDT/disordered regions) and future-proofing as structure predictors improve. Across reviews, the persistent long-tail class imbalance—reflected in low macro-F1 and weak performance on rare compartments—was flagged as a fundamental limitation that could reduce practical utility unless mitigated or at least transparently characterized. Finally, some reviewers asked for clearer implementation details (graph construction/edge features) and broader failure-case/interpretability analyses to support claims of biological insight.

**Reviewer Concerns:**

Addressed by the rebuttal (substantially)

- Need stronger / more modern structure-aware baselines (NKK9, Xxcu, 68FM): Authors added additional model families/strategies (e.g., diffusion-style model, CDConv+contrastive learning, and sequence+structure fusion) and reported their results, directly engaging the “upper bound may be underestimated” concern.

- Evidence-level label noise / bias (NKK9, 68FM): They expanded the analysis of evidence codes via weighting/handling experiments and clarified the intended use (filter/weight for high-rigor settings vs broader training), which addresses the “treating all positives may inflate noise” criticism.

- AlphaFold reliability / robustness to low-confidence regions (Xxcu, 68FM): They added pLDDT-based analyses and a comparison using an alternative structure source (Boltz subset), plus discussion of maintainability/extendability as structure predictors evolve.

- Missing implementation details for graph construction (68FM): They provided clarifications on edge criteria (sequence/spatial adjacency) and edge feature handling, reducing reproducibility ambiguity.

- Requested ablations that validate “structure matters” (general): The randomized-coordinate ablation already supports that models use geometry, and the rebuttal further reinforced this narrative.


Still outstanding (partially unresolved / remains a limitation)

- Severe class imbalance and weak rare-class performance (NKK9, 68FM): Even with reweighting/single-label/fusion/contrastive attempts, macro-F1 and many minority compartments remain difficult; this is an inherent challenge the benchmark exposes, but it remains a practical limitation.

- Clear “ceiling” for structure+sequence fusion and advanced modeling (axjf, NKK9): The added fusion/diffusion/contrastive experiments are helpful but don’t yet establish a strong new SOTA or a clean quantification of complementarity; results are mixed and the best sequence model remains competitive overall.

- Breadth/depth of interpretability beyond the Golgi case study (axjf, 68FM): They mention additional motif hypotheses, but broader, consistently validated interpretability across organelles is still limited.

**Reviewer Scores:**

axjf (score 8, accept): likely stays at 8. Their feedback was mostly “nice-to-have” strengthening (fusion, hierarchy, sample efficiency, broader interpretability), and the rebuttal/updated draft appears to add several of these, so they’d remain strongly positive.

Xxcu (score 4, marginally below): likely stays at 4. They explicitly commented that the rebuttal addressed concerns but they wanted to retain the score, which suggests limited upward movement even with discussion.

68FM (score 4, low confidence): likely 4 or 6. Their main asks (graph construction details, fusion baselines, pLDDT/failure-case analysis, evidence-level handling, typos/notation) were directly addressed; with discussion and increased confidence, a modest bump is plausible.

NKK9 (score 4, marginally below): likely 4 (maybe 6 if particularly satisfied). The rebuttal directly targets their key methodological gaps (contrastive learning, diffusion-style model, evidence-level robustness analysis). Remaining long-tail difficulty likely prevents a large jump.

---

### Decision · Program_Chairs · 2026-01-26

Accept (Poster)